# MambaSL: Exploring Single-Layer Mamba for Time Series Classification

**Yoo-Min Jung & Leekyung Kim**
Department of Industrial Engineering, Seoul National University
pamela7384@gmail.com, klk97@snu.ac.kr

## Abstract

Despite recent advances in state space models (SSMs) such as Mamba across various sequence domains, research on their standalone capacity for time series classification (TSC) has remained limited. We propose MambaSL, a framework that minimally redesigns the selective SSM and projection layers of a *single-layer Mamba*, guided by four TSC-specific hypotheses. To address benchmarking limitations—restricted configurations, partial University of East Anglia (UEA) dataset coverage, and insufficiently reproducible setups—we re-evaluate 20 strong baselines across all 30 UEA datasets under a unified protocol. As a result, MambaSL achieves state-of-the-art performance with statistically significant average improvements, while ensuring reproducibility via public checkpoints for all evaluated models. Together with visualizations, these results demonstrate the potential of Mamba-based architectures as a TSC backbone.

## 1 Introduction

State space models (SSMs) have emerged as a strong alternative to Transformers (Vaswani et al., 2017), with Mamba (Gu & Dao, 2024; Dao & Gu, 2024) marking a milestone in sequence domains such as language and video (Lenz et al., 2025; Li et al., 2024). In time series, however, convolutional- and Transformer-based architectures still dominate, with the former excelling in time series classification (TSC) and the latter in time series forecasting (TSF) (Wang et al., 2024b). While early works such as TimeMachine (Ahamed & Cheng, 2024) and S-Mamba (Wang et al., 2025b) showed Mamba's promise in TSF, its role in TSC remains underexplored.

Two gaps motivate our study. First, the standalone capacity of Mamba for TSC has seen little investigation. Wang et al. (2024b) ranked Mamba as the weakest TSC backbone, likely due to a lack of research rather than inherent architectural limitations, as only the vanilla variant was evaluated. To the best of our knowledge, TSCMamba (Ahamed & Cheng, 2025) is the only subsequent benchmarking attempt for TSC. However, its integration of feature engineering techniques such as ROCKET (Dempster et al., 2020) and CWT (Mallat, 1999) obscures Mamba's intrinsic contribution.

As a second gap, current TSC benchmarking suffers from three critical issues related to coverage, fairness, and reproducibility. First, evaluations are often restricted to only a fraction of the University of East Anglia (UEA) archive (Bagnall et al., 2018), leaving out challenging datasets with long sequences or high input dimensionality. Second, non-TSC models are frequently adopted without proper re-tuning, which risks underestimating their capacity; for example, TSF models like DLinear (Zeng et al., 2023) and PatchTST (Nie et al., 2023) have been reported in TSC contexts using default settings (Wu et al., 2023; Luo & Wang, 2024). Finally, reproducibility remains a concern—e.g., re-evaluations of TS2Vec (Yue et al., 2022) and GPT4TS (Zhou et al., 2023) by Eldele et al. (2024) revealed average accuracy drops exceeding 9%p compared to their original reports. These limitations undermine the reliability of comparative conclusions in the literature.

Motivated by these gaps, we revisit Mamba as a TSC backbone from two perspectives. **(i) Architecture:** We propose four TSC-specific hypotheses (H1–H4), which guide the redesign of selective SSM components and the input and output projection layers. **(ii) Evaluation protocol:** We establish a consistent benchmarking setup and re-evaluate strong TSC baselines across all 30 multivariate UEA datasets with extensive hyperparameter sweeps. Under the protocol, our *single-layer Mamba* TSC framework, MambaSL, achieves state-of-the-art performance. Our key contributions are:

- **TSC-specific hypotheses and architectural refinements.** We propose the following four hypotheses (H1–H4) that explain why naively applying Mamba within the existing TSC pipeline does not fully realize its potential, and validate them through corresponding design adjustments:

  **(H1) Scale input projection.** Since Mamba's output is modulated by a gating unit (Hua et al., 2022; Mehta et al., 2023a), insufficient input context can bottleneck performance, motivating a larger input projection receptive field for densely sampled time series.

  **(H2) Modularize time (in)variance.** As time series often exhibit near linear time-invariant behavior, we decouple time variance of Mamba as a hyperparameter. Simpler configurations often perform better, contradicting the ablation results from Gu & Dao (2024).

  **(H3) Remove skip connection.** In shallow networks, skip connections yield minimal performance gains (He et al., 2016; Ismail Fawaz et al., 2020). Given Mamba's strong long-range memory, we remove skip connections and construct logits solely from hidden states.

  **(H4) Aggregate via adaptive pooling.** Time series classification spans both global and event-driven patterns, which conventional pooling cannot accommodate. We therefore propose a multi-head adaptive pooling that weights temporal features in a dataset-specific manner.

- **Comprehensive and reproducible UEA benchmarking.** We evaluate 20 models on all 30 UEA datasets, covering various sequence lengths (8–17,984), input dimensions (2–1,345), and sample sizes (12–25,000). For each model, we explore approximately 200 hyperparameter combinations to select the optimal configuration. Notably, TSF models previously tested on the UEA datasets showed an average accuracy improvement of 3.04%p with hyperparameter tuning alone.

- **Empirical validation of a single-layer Mamba.** Our MambaSL framework achieves the state-of-the-art performance across the UEA benchmark, outperforming the second-best method by 1.41%p. We demonstrate the inherent potential of Mamba by validating our hypotheses through ablation studies and highlighting backbone- and dataset-specific features through visualizations.

The remainder of the paper is organized as follows. Section 2 reviews related work for TSC and Mamba, section 3 details our hypotheses and architectural refinements, section 4 describes our experimental setup and implementation details, section 5 presents results and analysis, and section 6 concludes the paper.

## 2 RELATED WORK

Recent advances in deep learning (DL) for TSC can be grouped into three streams. (i) **TSC-specific models** such as TS2Vec tailor encoders and readouts directly for classification (Yue et al., 2022; Eldele et al., 2024; Wen et al., 2025). (ii) **Foundation models** aim to adapt a single model across multiple time-series tasks, including TSC (Wu et al., 2023; Zhou et al., 2023; Luo & Wang, 2024; Wang et al., 2025a). (iii) **TSF-origin models** are frequently repurposed as baselines for TSC-specific and foundation models (Zeng et al., 2023; Zhang & Yan, 2023; Nie et al., 2023). Although evaluations typically rely on the UEA benchmark, inconsistent use of dataset subsets and single-setting evaluations for TSF-origin models have led to underestimated baselines. This highlights the need for standardized and consistent benchmarking in TSC.

Within SSMs, Mamba has been explored primarily for TSF. Variants such as TimeMachine (Ahamed & Cheng, 2024) and S-Mamba (Wang et al., 2025b) typically construct temporally mixed embeddings and then apply bidirectional Mamba along the channel axis to mitigate scan-order sensitivity. By contrast, adaptations to TSC in general are comparatively scarce, with TSCMamba (Ahamed & Cheng, 2025) being the only variant. Across TSF and TSC alike, Mamba variants tend to update the SSM state along non-temporal axes (channel or frequency) rather than the original time axis.

## 3 METHODS

### 3.1 PROBLEM DEFINITION

Let $\boldsymbol{X} = \boldsymbol{x}_{1:L} = [\boldsymbol{x}_1; \ldots; \boldsymbol{x}_L]$ be a multivariate time series of length $L$, where each $x_t$ at time $t$ is a $d_{\mathrm{x}}$-D vector. TSC infers the label $y \in \{1, \ldots, d_{\mathrm{y}}\}$ from $\boldsymbol{X}$, where $d_{\mathrm{y}}$ is the number of classes.

Many TSC architectures consist of three fundamental modules: an input projection $\Phi_{\mathrm{I}}$ that maps the input subsequence $\boldsymbol{x}_{t-k+1:t}$ (receptive field of size $k$) into a $d_{\mathrm{m}}$-D vector $\tilde{\boldsymbol{x}}_t = [\tilde{x}_t^{(1)}, \ldots, \tilde{x}_t^{(d_{\mathrm{m}})}]$,

a feature extractor $\Phi_{\mathrm{FE}}$ that produces per-timestep feature vectors $\boldsymbol{f}_{1:L}$ from $\tilde{\boldsymbol{x}}_{1:L}$ (typically of the same size), and an output projection $\Phi_{\mathrm{CLF}}$ that aggregates $\boldsymbol{f}_{1:L}$ into logits $\boldsymbol{l} \in \mathbb{R}^{d_{\mathrm{y}}}$. Formally,

$$\tilde{\boldsymbol{x}}_t = \Phi_{\mathrm{I}}\big(\boldsymbol{x}_{t-k+1:t}\big), \tag{1}$$

$$\boldsymbol{f}_{1:L} = \Phi_{\mathrm{FE}}\big(\tilde{\boldsymbol{x}}_{1:L}\big), \tag{2}$$

$$\boldsymbol{l} = \Phi_{\mathrm{CLF}}\big(\boldsymbol{f}_{1:L}\big), \tag{3}$$

$$\hat{y} = \underset{i \in \{1,\ldots,d_{\mathrm{y}}\}}{\arg\max} \ \mathrm{softmax}(\boldsymbol{l})_i. \tag{4}$$

The aggregation function of $\Phi_{\mathrm{CLF}}$ can be instantiated as a fully connected layer, global average/max pooling, or a last-timestep readout. Whereas many recent studies primarily emphasize improving $\Phi_{\mathrm{FE}}$, we examine and improve the entire TSC pipeline for Mamba in section 3.3.

## 3.2 TIME-VARIANT SSM IN MAMBA

At a high level, Mamba can be viewed as a selective SSM core augmented with a lightweight gated linear block (Hua et al., 2022; Mehta et al., 2023a). We next review the selective SSM, clarifying its components and establishing the notation used in our proposed method (section 3.3.2).

### 3.2.1 LINEAR TIME-INVARIANT SSM

An SSM maps a signal $u(t) \in \mathbb{R}$ to a $d_{\mathrm{s}}$-D state vector $\boldsymbol{s}(t)$ and projects it onto an output $y(t) \in \mathbb{R}$:

$$\dot{\boldsymbol{s}}(t) = \boldsymbol{A}\boldsymbol{s}(t) + \boldsymbol{B}u(t), \qquad y(t) = \boldsymbol{C}\boldsymbol{s}(t) + \boldsymbol{D}u(t), \tag{5}$$

with four parameters, $\boldsymbol{A} \in \mathbb{R}^{d_{\mathrm{s}} \times d_{\mathrm{s}}}$, $\boldsymbol{B} \in \mathbb{R}^{d_{\mathrm{s}} \times 1}$, $\boldsymbol{C} \in \mathbb{R}^{1 \times d_{\mathrm{s}}}$, and $\boldsymbol{D} \in \mathbb{R}$. This yields a linear time-invariant (LTI) system when the parameters are fixed over time.

With a step size $\Delta > 0$, the system is discretized as

$$\boldsymbol{s}_t = \bar{\boldsymbol{A}}\boldsymbol{s}_{t-1} + \bar{\boldsymbol{B}}u_t, \qquad y_t = \boldsymbol{C}\boldsymbol{s}_t + \boldsymbol{D}u_t, \tag{6}$$

where $\bar{\boldsymbol{A}} = \mathcal{F}_A(\Delta, \boldsymbol{A})$ and $\bar{\boldsymbol{B}} = \mathcal{F}_B(\Delta, \boldsymbol{A}, \boldsymbol{B})$ are discretization rules. Different discretization rules exist; the zero-order hold (ZOH) method is discussed in section 3.2.2.

While these forms are standard, we retain them here as the basis for our *multivariate* extension. In SSM-based DL models, this extension is typically implemented by sharing $\boldsymbol{A}$ across channels and broadcasting the other parameters (Gu et al., 2021; 2022; Gu & Dao, 2024). For instance, when using the SSM as the feature extractor $\Phi_{\mathrm{FE}}$ in equation 2, each input channel $\tilde{x}_t^{(j)}$ is mapped to a $d_{\mathrm{s}}$-D state via channel-specific $\boldsymbol{B}^{(j)}$ (and step size $\Delta^{(j)}$) and shared $\boldsymbol{A}$. Thus, at time $t$, the hidden state $\boldsymbol{S}_t = [\,\boldsymbol{s}_t^{(1)}; \ldots; \boldsymbol{s}_t^{(d_{\mathrm{m}})}\,] \in \mathbb{R}^{d_{\mathrm{s}} \times d_{\mathrm{m}}}$ where $\boldsymbol{s}_t^{(j)} = \mathcal{F}_A(\Delta^{(j)}, \boldsymbol{A})\boldsymbol{s}_{t-1}^{(j)} + \mathcal{F}_B(\Delta^{(j)}, \boldsymbol{A}, \boldsymbol{B}^{(j)})\tilde{x}_t^{(j)}$. The feature vector $\boldsymbol{f}_t$ is also generated by channel-specific $\boldsymbol{C}^{(j)}$ and $\boldsymbol{D}^{(j)}$. That is, in DL practice, the multivariate SSM is fundamentally a channel-independent LTI system.

To our knowledge, however, existing works rarely formalize how $\Delta$, $\boldsymbol{B}$, and $\boldsymbol{C}$ operate across channels. We show explicitly that $\Delta$ primarily acts along the temporal axis on each channel, whereas $\boldsymbol{B}$ and $\boldsymbol{C}$ primarily serve as channel-wise independent or mixing projections in Mamba's selective SSM. This clarification, often left implicit in prior literature, sets the stage for the following section.

### 3.2.2 SELECTIVE SSM

Mamba (Gu & Dao, 2024) proposed a selective SSM that allows $\Delta$, $\boldsymbol{B}$, and $\boldsymbol{C}$ to selectively propagate or forget information from given input. Given the projected input $\tilde{\boldsymbol{x}}_t \in \mathbb{R}^{d_{\mathrm{m}}}$, three learnable maps, $\phi_\Delta$, $\phi_B$, and $\phi_C$, produce the input-dependent parameters $\boldsymbol{\Delta}_t$, $\boldsymbol{B}_t$, and $\boldsymbol{C}_t$, respectively:

$$\begin{aligned}
\boldsymbol{\Delta}_t &= \phi_\Delta(\tilde{\boldsymbol{x}}_t) = \zeta\big(\mathrm{Linear}_{d_{\mathrm{m}}}^{\mathrm{bias}}\big(\mathrm{Linear}_{d_{\mathrm{r}}}(\tilde{\boldsymbol{x}}_t)\big)\big) = (\Delta_t^{(1)}, \ldots, \Delta_t^{(d_{\mathrm{m}})}) &&\in \mathbb{R}_{>0}^{d_{\mathrm{m}}}, \\
\boldsymbol{B}_t &= \phi_B(\tilde{\boldsymbol{x}}_t) = \mathrm{Linear}_{d_{\mathrm{s}}}(\tilde{\boldsymbol{x}}_t) &&\in \mathbb{R}^{d_{\mathrm{s}} \times 1}, \\
\boldsymbol{C}_t &= \phi_C(\tilde{\boldsymbol{x}}_t) = \mathrm{Linear}_{d_{\mathrm{s}}}(\tilde{\boldsymbol{x}}_t)^\top &&\in \mathbb{R}^{1 \times d_{\mathrm{s}}}.
\end{aligned} \tag{7}$$

Here, $\zeta$ is the softplus activation, $\mathrm{Linear}_d^{\mathrm{bias}}(x)$ and $\mathrm{Linear}_d(x)$ denote linear projections into $\mathbb{R}^d$ (identified with $\mathbb{R}^{d \times 1}$) with and without bias, and $d_{\mathrm{r}}$ is the rank of low-rank projection for $\boldsymbol{\Delta}_t$.

The selective SSM is then formulated as follows:[1]

$$\boldsymbol{s}_t^{(j)} = \bar{\boldsymbol{A}}_t^{(j)} \boldsymbol{s}_{t-1}^{(j)} + \bar{\boldsymbol{B}}_t^{(j)} \tilde{x}_t^{(j)}, \qquad f_t^{(j)} = \boldsymbol{C}_t \boldsymbol{s}_t^{(j)} + \boldsymbol{D}^{(j)} \tilde{x}_t^{(j)}, \quad j = 1, \ldots, d_{\mathrm{m}}, \qquad (8)$$

where $\bar{\boldsymbol{A}}_t^{(j)}$ and $\bar{\boldsymbol{B}}_t^{(j)}$ are discretized by the ZOH rule (Mehta et al., 2023b; Gu & Dao, 2024):

$$\begin{aligned} \bar{\boldsymbol{A}}_t^{(j)} &= \mathcal{F}_A(\Delta_t^{(j)}, \boldsymbol{A}) = \exp(\Delta_t^{(j)} \boldsymbol{A}), \\ \bar{\boldsymbol{B}}_t^{(j)} &= \mathcal{F}_B(\Delta_t^{(j)}, \boldsymbol{A}, \boldsymbol{B}_t) = (\Delta_t^{(j)} \boldsymbol{A})^{-1}(\exp(\Delta_t^{(j)} \boldsymbol{A}) - \boldsymbol{I}) \Delta_t^{(j)} \boldsymbol{B}_t. \end{aligned} \qquad (9)$$

This transformation changes the SSM from an LTI to a time-variant (TV) system, enabling context-aware reasoning that selectively stores or extracts key information. The explicit formulation also highlights that $\boldsymbol{A}$, $\boldsymbol{B}_t$, and $\boldsymbol{C}_t$ are shared across channels, while $\bar{\boldsymbol{A}}_t^{(j)}$ and $\bar{\boldsymbol{B}}_t^{(j)}$ differ due to the channel-wise $\Delta_t^{(j)}$.

### 3.2.3 ON TIME VARIANCE IN $\Delta$, $\boldsymbol{B}$, AND $\boldsymbol{C}$

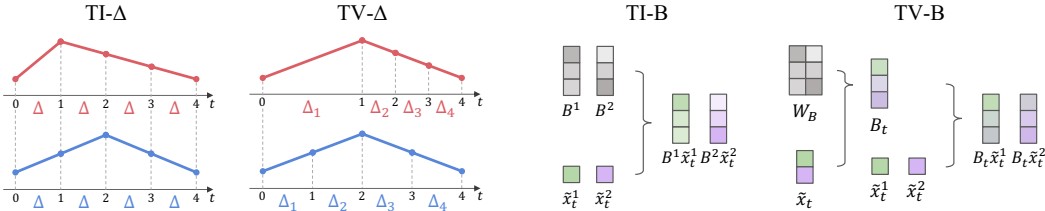

Figure 1: TI/TV parameterization of (left) $\Delta$ and (right) $\boldsymbol{B}$ in the SSM. TI-$\Delta$ fixes the update rate, while TV-$\Delta$ adapts it to align sequences with varying speeds. TI-$\boldsymbol{B}$ preserves channel independence, whereas TV-$\boldsymbol{B}$ introduces input-dependent mixing. $\boldsymbol{C}$ follows the same pattern as $\boldsymbol{B}$ at the output stage, highlighting the temporal pacing of $\Delta$ versus the spatial routing of $\boldsymbol{B}$ and $\boldsymbol{C}$.

While the parameters in Mamba share the same mechanism to yield a TV system by default, as defined in equation 7, their main roles differ. We denote $\Delta$, $\boldsymbol{B}$, and $\boldsymbol{C}$ in the LTI SSM using TI-notation (section 3.2.1), and $\boldsymbol{\Delta}_t$, $\boldsymbol{B}_t$, and $\boldsymbol{C}_t$ in the selective SSM using TV- notation (section 3.2.2).

$\Delta$**: Temporal Update Rate.** $\Delta$ controls the timescale of state updates. TI-$\Delta$ keeps a constant rate, whereas TV-$\Delta$ adapts to context: larger $\Delta$ accelerates updates, and smaller $\Delta$ prolongs memory. This resembles dynamic time warping (DTW), aligning sequences with varying local speeds (see the left side of Figure 1).

$\boldsymbol{B}$**: Input-to-State Routing.** $\boldsymbol{B}$ determines how each input channel drives the latent state. TI-$\boldsymbol{B}$ enforces channel-independent routing, whereas TV-$\boldsymbol{B}$ enables context-dependent mixing of input channels before entering the state space (see the right side of Figure 1).

$\boldsymbol{C}$**: State-to-Output Readout.** $\boldsymbol{C}$ maps states to output features. TI-$\boldsymbol{C}$ applies a fixed readout, preserving channel independence. TV-$\boldsymbol{C}$ introduces adaptive mixing at the output stage, potentially capturing richer cross-channel interactions.

**Interplay.** TV-$\Delta$ governs *temporal dynamics*, whereas TV-$\boldsymbol{B}$ and TV-$\boldsymbol{C}$ govern *spatial mixing*. The extreme cases are:

- TI-$\Delta$, TI-$\boldsymbol{B}$, TI-$\boldsymbol{C}$: LTI system with channel-independent I/O;
- TV-$\Delta$, TV-$\boldsymbol{B}$, TV-$\boldsymbol{C}$: fully TV system with channel-mixed I/O.

Thus, temporal and spatial variability can be decoupled and selectively controlled, motivating our second hypothesis (H2) in section 3.3.2. Note that, while these roles are primary, the ZOH rule couples them to some extent: $\Delta$ can indirectly modulate the influence of $\boldsymbol{B}$ and $\boldsymbol{C}$, and vice versa.

### 3.3 SINGLE-LAYER MAMBA FOR TSC

We refine the vanilla Mamba and projection layers through four hypotheses (H1–H4) motivated by their limitations in the standard TSC pipeline. The full framework is shown in Figure 2.

---

[1]For simplicity, we treat the selective SSM without shift as the feature extractor itself, using $\tilde{\boldsymbol{x}}_t$ in place of $u_t$ and $\boldsymbol{f}_t$ in place of $y_t$ from equation 6. In practice, $\Delta$, $\boldsymbol{B}$, and $\boldsymbol{C}$ are generated from an expanded input after a linear projection and a local convolution, but we omit this detail to keep the notation concise.

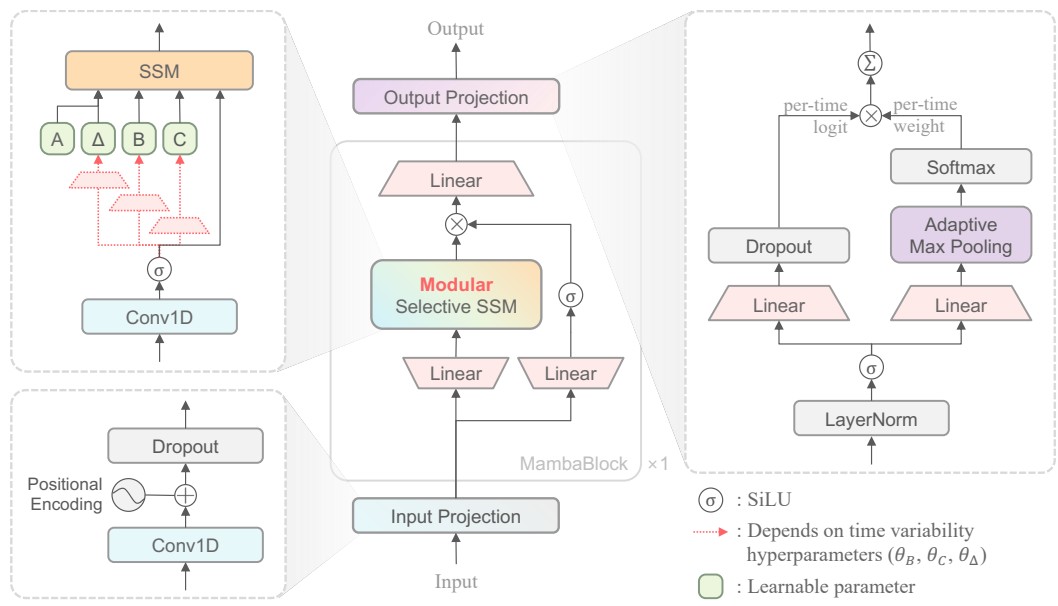

Figure 2: Overall structure of MambaSL, a single-layer Mamba framework designed for TSC.

### 3.3.1 INPUT PROJECTION

**H1: Scale input projection** In recent time series models, $\Phi_\text{I}$ is implemented as a 1-D convolution with a fixed kernel size $k = 3$ (Wu et al., 2023; Zhou et al., 2023), unless alternatives such as patching or frequency-domain transforms are employed (Nie et al., 2023; Wang et al., 2025a).

Since Mamba's gating mechanism modulates the SSM output based on this projection, we hypothesize that longer sequences require proportionally larger receptive fields. We therefore define

$$k = \max(k_\text{min}, \lfloor \lambda L \rfloor), \tag{10}$$

with $k_\text{min} = 3$ (minimum $k$), $\lambda = 0.02$ (sequence ratio), and stride $= 1$ to isolate kernel-size effects.

### 3.3.2 FEATURE EXTRACTOR

We adopt the vanilla Mamba as our feature extractor, introducing minimal changes to its SSM core under two hypotheses: (H2) modularizing time (in)variance and (H3) removing the skip connection.

**H2: Modularize time (in)variance** Building on section 3.2.3, we hypothesize that the optimal TI/TV configuration of $\Delta$, $B$, and $C$ is dataset-dependent. Evidence from TSF shows that channel independence may outperform mixing (Zeng et al., 2023; Nie et al., 2023), and shape-based distance—akin to TI-$\Delta$—can rival DTW in TSC (Paparrizos & Gravano, 2015). As aggregation with channel mixing is inevitable at the output, we expect $C$ to have a comparatively smaller effect, while $\Delta$ and $B$ are likely to exert a larger dataset-specific influence.

To systematically examine this, we introduce binary switches $\theta_\Delta, \theta_B, \theta_C \in \{0, 1\}$ that determine whether each parameter follows a TI or TV form. For channel $j \in \{1, \ldots, d_\text{m}\}$ and time $t$, the effective parameters are

$$\begin{aligned}
\Delta_t^{(j)\star} &= (1 - \theta_\Delta)\, \Delta^{(j)} + \theta_\Delta\, \Delta_t^{(j)} = (1 - \theta_\Delta)\, \Delta^{(j)} + \theta_\Delta\, \phi_\Delta(\tilde{x}_t)^{(j)} \in \mathbb{R}_{>0}, \\
B_t^{(j)\star} &= (1 - \theta_B)\, B^{(j)} + \theta_B\, B_t \;\;= (1 - \theta_B)\, B^{(j)} + \theta_B\, \phi_B(\tilde{x}_t) \quad \in \mathbb{R}^{d_s \times 1}, \\
C_t^{(j)\star} &= (1 - \theta_C)\, C^{(j)} + \theta_C\, C_t \;\;= (1 - \theta_C)\, C^{(j)} + \theta_C\, \phi_C(\tilde{x}_t) \quad \in \mathbb{R}^{1 \times d_s},
\end{aligned} \tag{11}$$

where all $\phi$ are defined in equation 7. This modularization yields $2^3 = 8$ configurations, ranging from a fully LTI system (all $\theta = 0$) to the selective SSM (all $\theta = 1$). Substituting equation 11 into equations 8 and 9 produces our modularized selective SSM, illustrated in the upper left of Figure 2.

**H3: Remove skip connection** Skip (residual) connections improve optimization in deep networks, yet their effect diminishes in shallow networks (He et al., 2016; Kim et al., 2017). InceptionTime (Ismail Fawaz et al., 2020) further shows that the skip connection makes little difference across 85 UCR datasets (Chen et al., 2015), indicating that it is not always essential for TSC.

In the single-layer setting, such shortcuts may bypass the SSM and hinder Mamba's representation learning. We therefore hypothesize that removing the skip connection forces the model to rely solely on the SSM's state evolution. This is implemented by omitting the $\boldsymbol{D}^{(j)}\tilde{x}_t^{(j)}$ term in equation 8 [2]:

$$f_t^{(j)} = \boldsymbol{C}_t \boldsymbol{s}_t^{(j)} \quad \text{(no skip term)}. \tag{12}$$

This modification positions learning the state vector $\boldsymbol{s}_t^{(j)}$ as the core of TSC.

### 3.3.3 OUTPUT PROJECTION

**H4: Aggregate via adaptive pooling** After the Mamba block, per-timestep features are aggregated into a logit vector $\boldsymbol{l}$. Conventional pooling methods, such as average or max, treat all steps equally or rely on a single dominant one, thus ignoring data-specific temporal importance. This issue is particularly critical for recurrent models, where the predicted label may shift over time (e.g., a `Handwriting` sequence labeled $g$ may initially resemble class $a$ before later aligning with $g$).

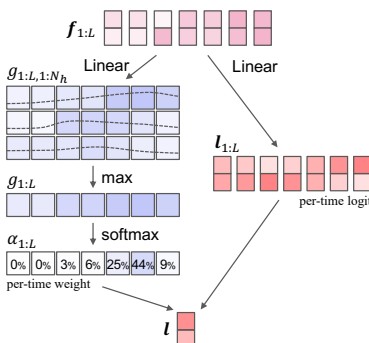

We propose a multi-head adaptive pooling with learnable gates (Figure 3). Specifically, $N_h$ independent gating heads each produce a scalar score $g_{t,h}$ for time step $t$:

$$g_{t,h} = \boldsymbol{w}_h^\top \boldsymbol{f}_t + b_h, \quad h = 1, \ldots, N_h. \tag{13}$$

For each $t$, the maximum gate value across heads is selected and normalized via softmax to obtain per-timestep weight $\alpha_t$, which is then applied to the per-timestep logit vector $\boldsymbol{l}_t$:

Figure 3: Illustration of proposed multi-head adaptive pooling with $(L, d_{\mathrm{m}}, N_h, d_{\mathrm{y}}) = (7, 2, 3, 2)$.

$$g_t = \max_h g_{t,h}, \qquad \alpha_t = \frac{\exp(g_t)}{\sum_{i=1}^{L} \exp(g_i)}, \qquad \boldsymbol{l} = \sum_{t=1}^{L} \alpha_t \boldsymbol{l}_t. \tag{14}$$

This formulation generalizes conventional pooling: uniform $\alpha_t$ recovers averaging, while a sharply peaked $\alpha_t$ approximates max pooling. Compared to attention pooling (Bahdanau et al., 2016), our design is lightweight yet expressive: multi-head gating explores diverse patterns, while adaptive max pooling selects the most confident signals, enabling robust dataset-specific aggregation.

## 4 EXPERIMENT

We conduct experiments on the full UEA benchmark (Bagnall et al., 2018), covering diverse sequence lengths, input dimensions, and sample sizes. For conciseness, we will hereafter refer to each dataset by its code (e.g., `EC` for EthanolConcentration; see Table 4). To ensure fairness, we allocate an identical search budget per model–dataset under a unified protocol, selecting the best configuration for each dataset. Detailed descriptions of the environment, datasets, and metrics are provided in appendix A, and hyperparameter settings are provided in appendix B.

**Baselines** We include (1) non-DL methods: DTW-based nearest neighbor (DTW$_D$; Berndt & Clifford, 1994), ROCKET (Dempster et al., 2020), HIVE-COTE 2.0 (HC2; Middlehurst et al., 2021), Hydra (Dempster et al., 2023), MultiRocket (MR; Tan et al., 2022)+Hydra; (2) MLP-based models: DLinear (Zeng et al., 2023), LightTS (Zhang et al., 2022), MTS-Mixer (Li et al., 2023); (3) CNN-based models: TimesNet (Wu et al., 2023), ModernTCN (Luo & Wang, 2024), TSLANet (Eldele et al., 2024), TimeMixer++ (Wang et al., 2025a); (4) Transformer-based models: FEDformer (Zhou et al., 2022), ETSformer (Woo et al., 2022), Crossformer (Zhang & Yan, 2023), PatchTST (Nie et al., 2023), GPT4TS (Zhou et al., 2023), iTransformer (Liu et al., 2024); (5) shape-based model: InterpGN (Wen et al., 2025); and (6) Mamba-based model: TSCMamba (Ahamed & Cheng, 2025).

---

[2] While Mamba and related literature typically derive the SSM equation without the skip connection $\boldsymbol{D}$, their official implementations default to enabling it (state-spaces, 2025). We changed this to be tunable.

# 5 RESULTS AND DISCUSSION

## 5.1 CLASSIFICATION RESULTS ON THE UEA BENCHMARK

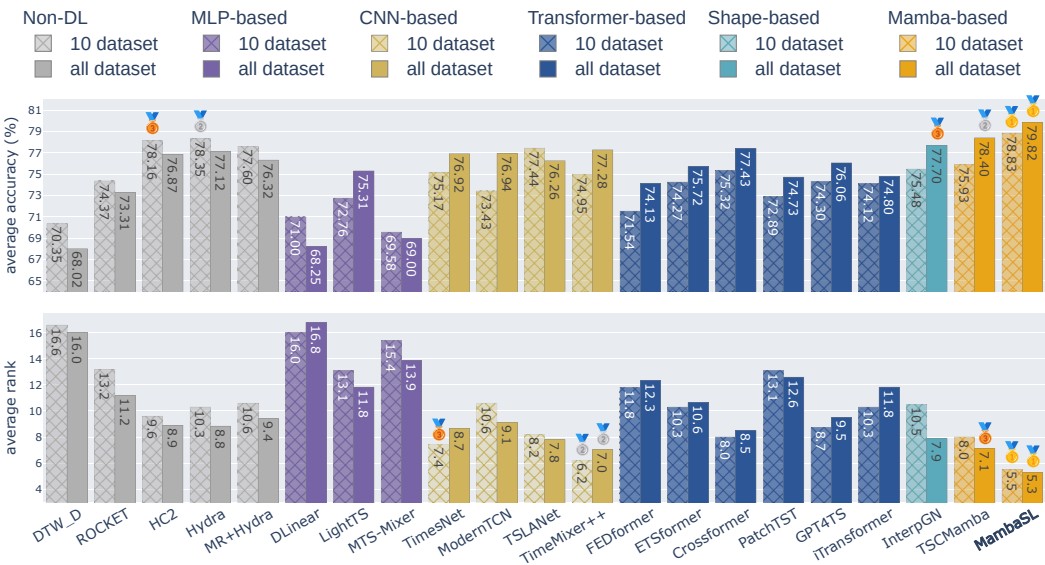

Figure 4: Comparison of average classification performances on the UEA benchmark. Non-DL models include DTW-D, ROCKET, HC2, Hydra, and MR+Hydra; DL models are grouped by their backbone structures.

As shown in Figure 4, our proposed model—MambaSL—established a new state of the art by achieving the best average accuracy and rank across both the widely used 10-dataset subset and the full set of 30 UEA datasets. In particular, MambaSL surpassed the second-best model, TSC-Mamba, by an average margin of 1.41%p. Wilcoxon signed-rank tests (Wilcoxon, 1945) across the 30 datasets confirmed statistically significant gains ($p < 0.05$) over all models except HC2 ($p = 0.56$), despite HC2 ranking only 8th overall (see appendix C.1 for full results and statistics).

As detailed in appendix C.1, MambaSL tends to perform strongly on datasets where multi-scale convolutions or frequency-domain features are typically dominant. For example, AWR, CR, EW, EP, HW, PS, RS, UW, and SWJ generally favor non-DL, TSLANet, InterpGN, and TSCMamba, yet MambaSL remains competitive. In particular, for CR and HW, MambaSL is the only DL model that surpasses DTW$_D$. At the same time, MambaSL shows competitive results on datasets such as FD, HMD, NATO, and SRS1, where MLP- or Transformer-based backbones have relative advantages. These results, achieved without additional domain transforms or complex mechanisms, demonstrate both the inherent capability of Mamba and the effectiveness of our architectural refinements.

To further contextualize the results, Figure 5 visualizes each model using a 2-D embedding of its 30 accuracy values via uniform manifold approximation and projection (UMAP) (McInnes et al., 2020). DL and non-DL methods are clearly clustered in separate regions, while InterpGN and MambaSL appear in between. The placement of InterpGN is reasonable since it ensembles shape-based and DL models. This suggests that MambaSL shares the strengths of both DL and non-DL approaches.

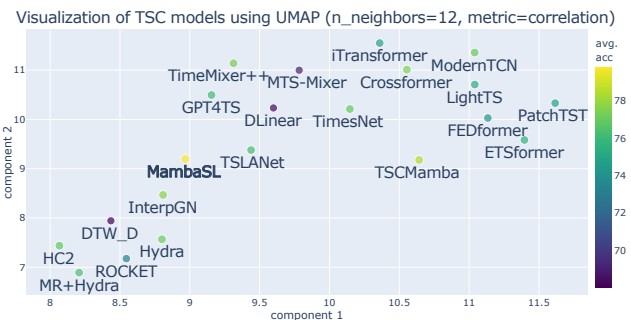

Figure 5: Visualization of the UEA classification results of each TSC model using the UMAP algorithm.

To complement the model-level analysis in Figure 5, we visualize the UEA datasets via UMAP in Figure 6. Here, each point represents a dataset embedded by its 21 accuracy values. Compared to the full results in appendix C.1, this view clearly shows the pattern of non-DL methods; datasets on which all non-DL baselines perform poorly form a tight cluster in the lower right (FD, JV, SRS2, AF, FM, HMD, MI, SWJ), whereas the upper-left cluster tends to outperform under the non-DLs.

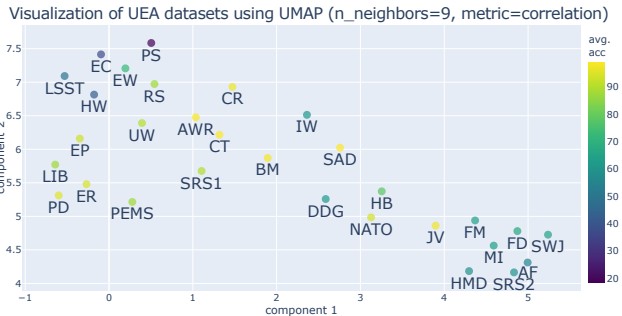

Figure 6: Visualization of the UEA classification results using the UMAP algorithm along dataset axis.

Finally, reproducibility remains a challenge in DL models due to randomness in initialization and optimization. We therefore compared our results with prior reports (appendix C.3). Our evaluation yielded higher performance than previously reported results, except for ModernTCN and TimeMixer++. However, their average ranks decline when the reported results are applied, confirming the fairness and rigor of our protocol across all baselines. For ModernTCN, although the average accuracy was lower than expected, we achieved higher accuracy with fewer hyperparameter trials compared to the revisited study (Akacik & Hoogendoorn, 2025). Notably, TSF-origin models (DLinear, LightTS, MTS-Mixer, FEDformer, ETSformer, Crossformer, PatchTST, and iTransformer) improved by over 3%p, suggesting that their methodologies can be reconsidered for TSC.

## 5.2 Model ablation

Table 1: Model ablation for four hypotheses (H1–H4). The best and the second-best are highlighted in bold and underline, respectively.

| Model | (H1) kernel size ✓: $k = 0.02L$ ✗: $k = 3$ | (H2) modular SSM ✓: modular TV/TI ✗: TV only | (H3) skip connection ✓: not use $D$ ✗: use $D$ | (H4) aggregation ✓: adaptive pool ✗: the others | avg. acc (10) | avg. acc | avg. rank | # win /draw | # lose /draw | rank test |
|---|---|---|---|---|---|---|---|---|---|---|
| MambaSL | ✓ | ✓ | ✓ | ✓ | **78.779** | 79.802 | **2.43** | | | |
| *w/o H1* | ✗ | ✓ | ✓ | ✓ | 77.413 | **80.215** | 2.53 | 25 | 22 | 0.217 |
| *w/o H2* | ✓ | ✗ | ✓ | ✓ | 75.659 | 77.075 | 5.93 | 30 | 5 | 0.000 |
| *w/o H3* | ✓ | ✓ | ✗ | ✓ | 77.715 | 79.163 | 3.50 | 22 | 14 | 0.071 |
| *w/o H4* | ✓ | ✓ | ✓ | ✗ (fully conn) | 76.438 | 77.722 | 4.87 | 24 | 10 | 0.003 |
| | ✓ | ✓ | ✓ | ✗ (avg pool) | 76.900 | 78.610 | 3.60 | 22 | 15 | 0.044 |
| | ✓ | ✓ | ✓ | ✗ (max pool) | 76.448 | 78.535 | 3.90 | 22 | 12 | 0.004 |
| | ✓ | ✓ | ✓ | ✗ (last step) | 75.856 | 76.848 | 4.87 | 25 | 9 | 0.001 |
| *only H2* | ✗ | ✓ | ✗ | ✗ (avg pool) | 77.092 | 77.935 | | 20 | 13 | 0.011 |
| Mamba | ✗ | ✗ | ✗ | ✗ (fully conn) | 73.286 | 74.243 | | 26 | 5 | 0.000 |

Table 1 summarizes the average classification accuracy of ablated models for each hypothesis (H1–H4), providing a systematic exploration of MambaSL's design space. The full results are included in appendix C.2. MambaSL remained competitive with state-of-the-art baselines, even when excluding H1, H3, or H4 individually. Also, none of the three ablations resulted in a statistically significant gains in the rank test. Motivated by this, we additionally ran an ablation that only applies H2; however, this variant failed to surpass the second-best baseline and showed statistically inferior performance ($p = 0.011$), suggesting that all our four hypotheses contribute positively to performance.

Disabling H1 ($k = 3$) yielded the highest average accuracy, surpassing 80%. Its overall ranking, however, remained second, as the gain was largely driven by marginal accuracy changes in the AF and SWJ datasets, which both have only 15 test samples. Additionally, on three datasets—AF, ER, and PEMS—where MambaSL was outside the top 10 (appendix C.1), a fully connected layer outperformed other classifiers. This suggests that for fixed-length datasets, fully connected readouts can be competitive as long as overfitting is controlled. Nevertheless, across the full benchmark, our multi-head adaptive pooling consistently performs better with greater generalizability.

## 5.3 FURTHER ANALYSIS

**Time variance ablation** Table 2 shows the classification accuracy of MambaSL under eight configurations of time variance hyperparameters, $\theta_\Delta$, $\theta_B$, and $\theta_C$.

No single combination is overwhelmingly superior, yet models with the LTI setting (all marked ✗) tend to outperform the fully TV setting (all marked ✓) of selective SSM. This contrasts with the findings of Gu & Dao (2024) in their ablation study, where the fully TV setting was superior for language modeling.

Table 2: Ablation on time variance of $\Delta$, $B$, and $C$. The best and the second-best are highlighted in bold and underline, respectively.

| TV-$\Delta$ | TV-$B$ | TV-$C$ | avg. acc (10) | avg. acc | avg. rank |
|---|---|---|---|---|---|
| ✗ | ✗ | ✗ | **77.280** | 77.400 | 3.87 |
| ✗ | ✗ | ✓ | 76.938 | 77.102 | 3.93 |
| ✗ | ✓ | ✗ | 76.451 | 77.042 | 3.80 |
| ✗ | ✓ | ✓ | 76.422 | **77.767** | **3.60** |
| ✓ | ✗ | ✗ | 76.253 | 77.707 | 3.63 |
| ✓ | ✗ | ✓ | 76.299 | 77.148 | 3.70 |
| ✓ | ✓ | ✗ | 76.309 | 77.085 | 3.80 |
| ✓ | ✓ | ✓ | 75.938 | 77.055 | 4.13 |

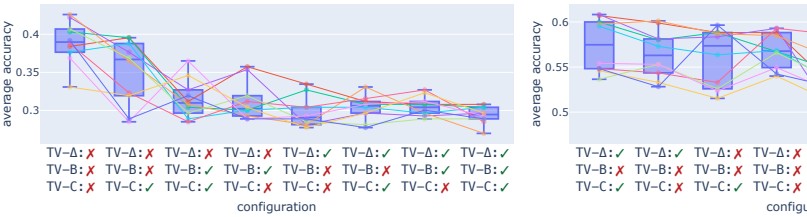

Figure 7: Visualization of classification accuracy of MambaSL along TI/TV configurations on (left) `EC` and (right) `HW` datasets, in order of maximum accuracy.

In some datasets, the trend is particularly clear. As shown on the left side of Figure 7, in `EC`, accuracy increases as more parameters are set to be time-invariant, with $\Delta$ having the largest impact. This indicates that dynamic warping of the sampling intervals can distort the chemical reaction signal of ethanol, which can impair classification performance. Furthermore, similar to the right of Figure 7, using TI-$B$ consistently yields better results than using TV-$B$ across many datasets (`HW`, `HB`, `CT`, `DDG`, `UW`, and `RS`), while the opposite trend is also observed (`PEMS`, `CR`, `PD`, and `PS`). By contrast, no dataset exhibits a consistent preference with respect to $C$. These observations support our hypothesis that while the time variance of $\Delta$ and $B$ can be dataset-specific, that of $C$ has limited impact, as output-channel mixing naturally occurs in the classifier. The visualizations for all 30 datasets are provided in appendix C.4.

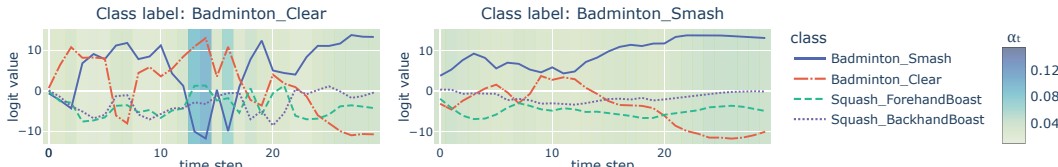

Figure 8: Visualization of adaptive pooling on the `RS` dataset for two samples. Line plots show per-timestep logits $l_t$, and heatmaps illustrate the corresponding per-timestep weights $a_t$.

**Analysis of pooling behavior** To further examine the role of adaptive pooling, we analyzed the `RS` dataset, which contains relatively few classes and yields strong performance under max pooling. The left plot in Figure 8 depicts a sample where the predicted label differs between a simple average of per-timestep logits and the weighted average using $a_t$. As shown, large logit values do not necessarily correspond to large weights, and regions with high weights are often localized. As in the right plot of Figure 8, there are also samples with nearly uniform weights, effectively reducing adaptive pooling to average pooling. This adaptive weighting enables MambaSL to outperform average pooling, achieving the second-highest accuracy on the `RS` dataset (appendices C.1 and C.2).

**Effect of model depth** Although our main architecture uses a single Mamba block as shown in Figure 2, we also ablated the number of stacked blocks to examine whether a deeper MambaSL

further improves performance. Table 3 compares variants with 1, 2, and 3 Mamba blocks under the same training protocol (full results are provided in appendix C.5). We observed that a single layer achieved the best average accuracy and the best average rank across the 30 UEA datasets. While performance gradually decreases as the number of layers increases, the differences are small, and all three variants perform similarly overall. Given these results, we conclude that a carefully configured *single-layer* Mamba is sufficient as a backbone for TSC in this regime, and we adopt the 1-layer MambaSL as our default architecture for both simplicity and efficiency.

Table 3: Classification accuracy (%) of MambaSL according to its model depth. The best and the second-best are highlighted in bold and underline, respectively.

| depth | avg. acc (10) | avg. acc | avg. rank (10) | avg. rank | # of win/draw | # of lose/draw | rank test |
|---|---|---|---|---|---|---|---|
| 1 | 78.827 | **79.819** | **1.30** | **1.33** | | | |
| 2 | 78.853 | 79.316 | 1.90 | 1.80 | 26 | 15 | 0.035 |
| 3 | **78.918** | 79.279 | 2.10 | 2.07 | 24 | 12 | 0.004 |

## 5.4 ADDITIONAL EXPERIMENTS

**UEA benchmark classification considering data leakage issue**    To account for the data-leakage issue reported for `TSLib` in TSC settings (Akacik & Hoogendoorn, 2025; Talukder et al., 2024), we additionally trained MambaSL under the InceptionTime protocol (Ismail Fawaz et al., 2020), where the checkpoint with the lowest training loss is used for evaluation. For comparison, we selected the top three high-performing models from each source—HC2 (Middlehurst et al., 2021), BORF (Spinnato et al., 2024), and our own experiments—to form a comparable set of classical baselines. These results, along with the corresponding evaluation tables, are provided in appendix C.6. While MambaSL shows lower accuracy under the stricter InceptionTime protocol than under the default `TSLib` setting, it remains competitive relative to the strongest non-DL baselines reported in previous studies and in our re-evaluation.

**Classification on recent domain-specific dataset**    Although the UEA archive (Bagnall et al., 2018) covers a broad range of TSC datasets, it does not include more recently released real-world benchmarks. Therefore, we additionally evaluated MambaSL on ADFTD (Miltiadous et al., 2023) and FLAAP (Kumar & Suresh, 2022), two of the most recent datasets benchmarked by Medformer (Wang et al., 2024a) in the medical and human activity recognition (HAR) domains, respectively. Classification results compared with strong baselines in Medformer are provided in appendix C.7. Across both datasets, MambaSL achieves competitive performance without any domain-specific modifications, demonstrating its generalizability to recent TSC tasks.

## 6 CONCLUSION

In this study, we introduce four hypotheses as to why Mamba has remained underexplored in TSC despite its success in other sequence domains. With minimal architectural modifications, our *single-layer Mamba* framework, MambaSL, achieved state-of-the-art performance across the entire UEA benchmark. By re-examining 20 prior baselines, we further highlighted systematic differences across backbone structures and demonstrated that Mamba can serve as a strong multivariate TSC backbone.

Nevertheless, our work has two main limitations. First, while our model is robust across the UEA benchmark, it rarely achieves overwhelming superiority on individual datasets, implying that practical use may require domain-specific adaptations. For instance, prior knowledge of input embedding may constrain kernel size, making receptive-field scaling with sequence length suboptimal in certain domains. Second, our study is limited to the first version of Mamba (Gu & Dao, 2024). While Mamba-2 (Dao & Gu, 2024) introduces an efficient parallel Mamba block, applying time-variance modularization requires low-level optimizations beyond the scope of this study.

Overall, our findings demonstrate that a carefully configured single-layer Mamba can serve as a competitive backbone for TSC. Beyond TSC, future work may further improve Mamba by leveraging receptive-field scaling and time-variance control, along with shallow, task-aligned architectural choices and adaptive aggregation strategies.

## REPRODUCIBILITY STATEMENT

Source code and scripts are publicly available at GitHub repository , and datasets, along with the best-performing model checkpoints, have been uploaded to Google Drive.

- The source code offers a complete implementation of Mamba and all other baselines, with the exception of dataset and checkpoint downloads. The repository includes grid search script generation, local training execution, pretrained model evaluation, and result visualization.
- Full training logs, corresponding to the grid search results reported in Figure 4 and Table 1, are also provided in the repository. We are confident that the experimental setup and hyperparameter settings mentioned in appendices A and B are identical to the scripts and logs inside the repository.
- The dataset folder also contains preprocessed CWT and ROCKET features for TSCMamba, compatible with the released checkpoints.
- Detailed reproduction instructions are available in the `README.md` file.

## THE USE OF LARGE LANGUAGE MODELS

The authors acknowledge the use of ChatGPT-5 and Google Translate only during the writing process to ensure rigorous and concise academic English writing.

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

# A EXPERIMENTAL SETUP

**Environment**    All experiments were implemented in Python 3.12.8 and PyTorch 2.5.1. Most were run on NVIDIA GTX 1080 Ti (11GB), while a few required NVIDIA A100 (40GB) on Google Colab due to memory limits. Classical baselines used the `aeon` toolkit (Middlehurst et al., 2024), and all deep models except TSLANet were integrated into the Time-Series Library (`TSLib`) (Wu et al., 2023; Wang et al., 2024b). For models supported in `TSLib`, we used the provided code; for others, we adapted the authors' official implementations. All source code is available in our GitHub repository (see reproducibility statement).

**Dataset**    The UEA archive (Bagnall et al., 2018) provides 30 multivariate TSC datasets with diverse sample sizes, input dimensions, lengths, and class counts (Table 4). As `TSLib` has become a widely used framework, a subset of 10 datasets (`EC`, `FD`, `HW`, `HB`, `JV`, `PEMS`, `SRS1`, `SRS2`, `SAD`, and `UW`) is commonly used in recent TSC benchmarking practices.

Table 4: Summary of the 30 UEA datasets used in our experiments

| Dataset | Code | Train Size | Test Size | Length | Variables | Classes |
|---|---|---|---|---|---|---|
| EthanolConcentration | EC | 261 | 263 | 1751 | 3 | 4 |
| FaceDetection | FD | 5890 | 3524 | 62 | 144 | 2 |
| Handwriting | HW | 150 | 850 | 152 | 3 | 26 |
| Heartbeat | HB | 204 | 205 | 405 | 61 | 2 |
| JapaneseVowels | JV | 270 | 370 | 7–29 | 12 | 9 |
| PEMS-SF | PEMS | 267 | 173 | 144 | 963 | 7 |
| SelfRegulationSCP1 | SRS1 | 268 | 293 | 896 | 6 | 2 |
| SelfRegulationSCP2 | SRS2 | 200 | 180 | 1152 | 7 | 2 |
| SpokenArabicDigits | SAD | 6599 | 2199 | 4–93 | 13 | 10 |
| UWaveGestureLibrary | UW | 120 | 320 | 315 | 3 | 8 |
| ArticularyWordRecognition | AWR | 275 | 300 | 144 | 9 | 25 |
| AtrialFibrillation | AF | 15 | 15 | 640 | 2 | 3 |
| BasicMotions | BM | 40 | 40 | 100 | 6 | 4 |
| CharacterTrajectories | CT | 1422 | 1436 | 60–182 | 3 | 20 |
| Cricket | CR | 108 | 72 | 1197 | 6 | 12 |
| DuckDuckGeese | DDG | 50 | 50 | 270 | 1345 | 5 |
| EigenWorms | EW | 128 | 131 | 17984 | 6 | 5 |
| Epilepsy | EP | 137 | 138 | 206 | 3 | 4 |
| ERing | ER | 30 | 270 | 65 | 4 | 6 |
| FingerMovements | FM | 316 | 100 | 50 | 28 | 2 |
| HandMovementDirection | HMD | 160 | 74 | 400 | 10 | 4 |
| InsectWingbeat | IW | 25000 | 25000 | 2–22 | 200 | 10 |
| Libras | LIB | 180 | 180 | 45 | 2 | 15 |
| LSST | LSST | 2459 | 2466 | 36 | 6 | 14 |
| MotorImagery | MI | 278 | 100 | 3000 | 64 | 2 |
| NATOPS | NATO | 180 | 180 | 51 | 24 | 6 |
| PenDigits | PD | 7494 | 3498 | 8 | 2 | 10 |
| PhonemeSpectra | PS | 3315 | 3353 | 217 | 11 | 39 |
| RacketSports | RS | 151 | 152 | 30 | 6 | 4 |
| StandWalkJump | SWJ | 12 | 15 | 2500 | 4 | 3 |

**Metrics**    The evaluation metrics reported in our figures and tables are summarized as follows:

- **avg. acc:** Average accuracy (%) across all UEA datasets.
- **avg. acc (10):** Average accuracy (%) for 10 UEA datasets, following prior TSC practices.
- **avg. rank:** Average rank of each model across all UEA datasets.
- **# of top-$N$:** Count of datasets in which each model is in top-$N$.
- **# of win/draw:** Count of datasets where proposed model achieved higher or the same accuracy.
- **# of loss/draw:** Count of datasets where proposed model achieved lower or the same accuracy.
- **rank test:** $p$-value of Wilcoxon signed-rank test (Wilcoxon, 1945). A value smaller than the significance level (0.05) indicates that proposed model ranked greater than the selected.

# B  HYPERPARAMETER SETTINGS

## B.1  BASIC HYPERPARAMETER SETTINGS

We standardized experimental hyperparameter settings across all models to focus on model tuning, where most were chosen as common values across prior UEA benchmarks and `TSLib`:

- **Batch size**: 16   (some experiments used smaller batch size as described in appendix C.1.)
- **Learning rate**: 0.001
- **Learning rate scheduler**: none
- **Optimizer**: RAdam (Liu et al., 2020)
- **Train epochs**: 100
- **Patience**: 10
- **Dropout**: 0.1
- **Seed**: 2021   (for DL models)

Although this unified setting may deviate from model-specific defaults, extensive model-level grid searches compensated for potential performance degradation (see appendix C.3).

## B.2  MODEL HYPERPARAMETER SETTINGS FOR GRID SEARCH

We primarily considered the hyperparameter settings that were tested in either the original papers or the source code scripts, and we also referred to the hyperparameters that were set in `TSLib` scripts. For TSF-origin and foundation models that do not provide classification scripts, we tend to choose the same or smaller values than in forecasting scripts since we empirically found that many UEA datasets require compact model sizes compared to TSF datasets. Over 200 candidate settings were evaluated per model–dataset pair, except for some models that did not require significant extension. The hyperparameters explored are as below:

- **Non-DL**
  - $DTW_D$: fixed implementations (no learnable hyperparameters).

  - ROCKET, HC2, Hydra, MR+Hydra: 3 combinations
    * `seed`: 0, 1, 2
    * Other parameters are fixed to the default optimal setting of the original papers since the models themselves have either large number (1e-4) of kernels or ensemble mechanism.

- **MLP-based models**
  - DLinear: 15 combinations
    * `moving_avg`: 0.5%, 1%, 2%, 3%, 4%, 5%, 10%, 15%, 20%, 25%, 30%, 35%, 40%, 45%, 50% of sequence length
    * Since DLinear only has one model parameter to be tuned, we didn't over-expand the number of hyperparameter combinations beyond 200.

  - LightTS: 100 combinations
    * `d_model`: 32, 64, 128, 256, 512
    * `chunk_size`: $1/21 - 1/2$ of sequence length
    * Since LightTS only has two model parameters to be tuned, we didn't over-expand the number of hyperparameter combinations beyond 200.

  - MTS-Mixer: 256 combinations
    * `e_layers`: 2
    * `d_model`: 128, 256, 512, 1024
    * `d_ff`: 0, 2, 4, 8, 16, 32, 64, 128
    * `fac_C`: 0 (False) if `d_ff` is 0, else 1 (True)
    * `down_sampling_window`: 0, 1%, 2%, 3%, 5%, 7.5%, 10%, 12.5% of sequence length
    * `fac_T`: 0 (False) if `down_sampling_window` is 0, else 1 (True)
    * `use_norm`: 1 (True)

- **CNN-based models**
  - TimesNet: 252 combinations
    * `e_layers`: 2, 3, 4
    * `(d_model, d_ff)`: (8,16), (8,32), (8,64), (16,16), (16,32), (16,64), (32,32), (32,64), (32,128), (64,64), (64,128), (64,256), (128,128), (128,256)
    * `top_k`: 1, 2, 3
    * `num_kernels`: 4, 6
  - ModernTCN: 252 combinations
    * `ffn_ratio`: 1, 2, 4
    * `patch_size`: 2.5%, 5%, 7.5%, 10%, 15%, 20%, 25% of sequence length
    * `patch_stride`: 50% of `patch_size`
    * `(num_blocks, large_size, small_size, dims)`: ("1", "13", "5", "32"), ("1", "13", "5", "64"), ("1", "13", "5", "128"), ("1", "13", "5", "256"), ("1 1", "9 9", "5 5", "32 64"), ("1 1", "9 9", "5 5", "64 128"), ("1 1", "9 9", "5 5", "128 256"), ("1 1", "13 13", "5 5", "32 64"), ("1 1", "13 13", "5 5", "64 128"), ("1 1", "13 13", "5 5", "128 256"), ("1 1 1", "9 9 9", "5 5 5", "32 64 128"), ("1 1 1", "13 13 13", "5 5 5", "32 64 128")
  - TSLANet: 252 combinations
    * `depth`: 1, 2, 3
    * `emb_dim`: 32, 64, 128, 256
    * `mlp_ratio`: 1, 2, 3
    * `patch_size`: 2.5%, 5%, 7.5%, 10%, 15%, 20%, 25% of sequence length
    * `patch_stride`: 50% of `patch_size`
    * `masking_ratio`: 0.4
    * `ICB`: 1 (True)
    * `ASB`: 1 (True)
    * `adaptive_filter`: 1 (True)
    * `load_from_pretrained`: 1 (True)
    * `pretrain_lr`: 0.001
    * `pretrain_epoch`: 50
  - TimeMixer++: 216 combinations
    * `down_sampling_method`: "conv"
    * `down_sampling_layers`: 1, 2, 3
    * `down_sampling_window`: 2
    * `num_kernels`: 6
    * `e_layers`: 1, 2, 3, 4
    * `d_model`: 16, 32, 64
    * `d_ff`: 16, 32, 64
    * `n_heads`: 8
    * `top_k`: 2, 3

- **Transformer-based models**
  - FEDformer: 250 combinations
    * `moving_avg`: 0.5%, 1%, 2%, 3%, 5%, 10%, 20%, 30%, 40%, 50% of sequence length
    * `e_layers`: 2
    * `d_model`: 32, 64, 128, 256, 512
    * `d_ff`: 128, 256, 512, 1024, 2048
    * `n_heads`: 8
  - ETSformer: 240 combinations
    * `e_layers`: 1, 2
    * `d_model`: 32, 64, 128, 256, 512
    * `d_ff`: 64, 128, 256, 512, 1024, 2048
    * `n_heads`: 8
    * `top_k`: 1, 2, 3, 4
  - Crossformer: 288 combinations
    * `e_layers`: 1, 2, 3
    * `d_model`: 32, 64, 128, 256
    * `d_ff`: 64, 128, 256, 512
    * `n_heads`: 4
    * `factor`: 3, 10
    * `seg_len`: 6, 12, 24 (change the code of `TSLib` to modify the fixed value)
  - PatchTST: 252 combinations
    * `e_layers`: 1, 2, 3
    * `d_model`: 16, 32, 64, 128
    * `d_ff`: 64, 128, 256
    * `n_heads`: 4 if `d_model` $\in \{16, 32\}$ else 16
    * `patch_size`: 2.5%, 5%, 7.5%, 10%, 15%, 20%, 25% of sequence length
    * `patch_stride`: 50% of `patch_size`
  - GPT4TS: 252 combinations
    * `e_layers`: 3, 4, 5, 6
    * `d_model`: 768 (fixed since the model use pretrained GPT-2)
    * `d_ff`: 768 (fixed since the model use pretrained GPT-2)
    * `patch_size`: 1/25 – 1/5 of sequence length
    * `patch_stride`: 25%, 50%, 100% of `patch_size`
  - iTransformer: 240 combinations
    * `e_layers`: 1, 2, 3, 4
    * `d_model`: 64, 128, 256, 512, 1024, 2048
    * `d_ff`: 64, 128, 256, 512, 1024
    * `n_heads`: 8
    * `factor`: 1, 3

- **Shape-based model**
  - InterpGN: 243 combinations
    * `dnn_type`: "FCN"
    * `num_shaplet`: 5, 10, 15
    * `lambda_div`: 0, 0.1, 1.0
    * `lambda_reg`: 0, 0.1, 1.0
    * `epsilon`: 0.5, 1.0, 2.0
    * `gating_value`: 0.5, 0.75, 1.0
    * `distance_func`: "euclidean"
    * `memory_efficient`: False
    * `sbm_cls`: "linear"

- **Mamba-based models**
  - TSCMamba: 288 combinations
    * `d_model`: 64, 128, 256
    * `e_layers`: 1, 2
    * `expand`: 1, 2
    * `d_conv`: 2, 4
    * `d_ff` (`d_state`): 32, 64, 128
    * `no_rocket`: 0 (False)
    * `half_rocket`: 0 (False), 1 (True)
    * `additive_fusion`: 0 (multiplicative fusion), 1 (additive fusion)
    * `max_pooling`: 0 (avg pooling), 1 (max pooling)
    * `channel_token_mixing`: 0 (False)
    * `only_forward_scan`: 0 (False)
    * `flip_dir`: 2 (vertical flip)
    * `reverse_flip`: 0 (False)
    * `patch_size`: 8
    * `patch_stride`: 8
    * `rescale_size`: 64
    * `variation`: 64
    * `initial_focus`: 1.0
  - MambaSL (ours): 240 combinations
    * `d_model` ($d_{\mathrm{m}}$): 32, 64, 128, 256, 512, 1024
    * `d_ff` ($d_{\mathrm{s}}$): 1, 2, 4, 8, 16
    * `tv_dt` ($\theta_{\Delta}$): 0 (False), 1 (True)
    * `tv_B` ($\theta_{\boldsymbol{B}}$): 0 (False), 1 (True)
    * `tv_C` ($\theta_{\boldsymbol{C}}$): 0 (False), 1 (True)
    * `use_D`: 0 (False)
    * `expand`: 1
    * `d_conv`: 4

# C ADDITIONAL EXPERIMENTAL RESULTS

## C.1 FULL RESULTS OF TSC MODELS ON THE UEA BENCHMARK

Table 5 presents the full experimental results corresponding to Figure 4. As mentioned in appendix A, some experiments could not be run on the GTX 1080 Ti GPU with batch size 16. Footnotes in the table indicate such experiments. We only used the A100 GPU for cases that could not be run on the 1080 Ti with batch size 4. Also, note that only one hyperparameter combination was tested in our limited resource environment for TSCMamba (Ahamed & Cheng, 2025)–IW pair, since preprocessing took over 2 days and training took over 12 hours per epoch.

Table 5: Classification accuracy (%) of TSC models on 30 datasets from the UEA archives. The best and the second-best are highlighted in bold and underline, respectively.

(a) Classification results from $DTW_D$ to ModernTCN.

| | non-DL | | | | | MLP-based | | | CNN-based | |
|---|---|---|---|---|---|---|---|---|---|---|
| | $DTW_D$ (1994) | ROCKET (2020) | HC2 (2021) | Hydra (2023) | MR+ Hydra (2023) | DLinear (2023) | LightTS (2022) | MTS-Mixer (2023) | TimesNet (2023) | Modern TCN (2024) |
| EC | 32.319 | 44.487 | 52.852 | 54.373 | **60.076** | 31.179 | 32.319 | 30.798 | 33.460 | 32.319 |
| FD | 52.866 | 61.606 | 61.379 | 60.982 | 61.294 | 69.495 | 68.473 | 68.530 | 70.148 | 66.998 |
| HW | 60.706 | 59.294 | 56.000 | 56.353 | 53.529 | 23.765 | 25.059 | 19.647 | 37.529 | 30.118 |
| HB | 71.707 | 75.610 | 78.537 | 76.585 | 77.073 | 76.585 | 77.561 | 79.024 | **83.902** | 78.049 |
| JV | 95.946 | 83.784 | 98.378 | 97.838 | 98.378 | 96.486 | 96.757 | 95.946 | 98.649 | 98.649 |
| PEMS | 71.098 | 84.393 | 97.688 | **98.266** | 82.659 | 82.659 | 87.861 | 82.081 | 87.861 | 86.705[A] |
| SRS1 | 77.474 | 85.666 | 89.078 | 86.689 | **94.881** | 92.833 | 92.833 | 78.840 | 92.150 | 93.515 |
| SRS2 | 53.889 | 55.556 | 54.444 | 58.889 | 55.000 | 57.222 | 59.444 | 58.333 | 60.000 | 60.000 |
| SAD | 97.226 | 99.227 | 99.136 | 99.136 | 99.045 | 96.953 | 98.226 | 99.454 | 99.545 | 99.500 |
| UW | 90.313 | 94.063 | 94.063 | 94.375 | 94.063 | 82.813 | 89.063 | 83.125 | 88.438 | 88.438 |
| AWR | 98.667 | **99.333** | **99.333** | **99.333** | **99.333** | 97.667 | 98.333 | 99.000 | 98.667 | 98.667 |
| AF | 20.000 | 6.667 | 20.000 | 26.667 | 6.667 | 53.333 | 73.333 | 60.000 | 66.667 | 73.333 |
| BM | 97.500 | **100.000** | **100.000** | **100.000** | **100.000** | 87.500 | **100.000** | **100.000** | **100.000** | **100.000** |
| CT | 98.816 | 99.304 | 99.304 | 99.164 | 99.373 | 98.120 | 98.955 | 97.981 | 98.747 | 99.304 |
| CR | **100.000** | **100.000** | **100.000** | **100.000** | 98.611 | 91.667 | 93.056 | 97.222 | 97.222 | 97.222 |
| DDG | 58.000 | 52.000 | 70.000 | 68.000 | 60.000 | 66.000 | 56.000 | 68.000[8] | 68.000 | 66.000[A] |
| EW | 61.832 | 91.603 | 97.710 | **98.473** | **98.473** | 42.748[A] | 54.198[8] | 66.412 | 63.359 | 64.122 |
| EP | 96.377 | 99.275 | **100.000** | **100.000** | **100.000** | 65.217 | 97.101 | 66.667 | 96.377 | 96.377 |
| ER | 91.481 | 98.889 | 98.889 | 98.889 | **99.259** | 91.852 | 94.444 | 77.778 | 95.556 | 97.407 |
| FM | 53.000 | 57.000 | 61.000 | 55.000 | 60.000 | 62.000 | 64.000 | 63.000 | 66.000 | 67.000 |
| HMD | 18.919 | 51.351 | 48.649 | 51.351 | 36.486 | 66.216 | 66.216 | 70.270 | **72.973** | 66.216 |
| IW | 48.692 | 38.524 | 66.796 | 67.052 | 65.744 | 18.996 | 71.676 | 72.424 | 64.024 | **75.588** |
| LIB | 87.222 | 90.556 | 94.444 | 93.333 | 93.333 | 70.556 | 89.444 | 34.444 | 88.333 | 90.000 |
| LSST | 55.109 | 64.031 | 66.383 | **66.504** | 64.274 | 34.023 | 41.525 | 34.185 | 44.120 | 43.593 |
| MI | 50.000 | 54.000 | 50.000 | 54.000 | 51.000 | 64.000 | 68.000 | 66.000 | 68.000 | 67.000 |
| NATO | 88.333 | 89.444 | 91.111 | 90.000 | 93.333 | 96.111 | 97.222 | 95.000 | 98.333 | 95.556 |
| PD | 97.713 | 98.228 | 97.656 | 97.856 | 97.284 | 92.481 | 96.998 | 46.941 | 98.914 | 98.513 |
| PS | 15.121 | 27.826 | 33.015 | 33.164 | **33.642** | 6.651 | 16.224 | 5.428 | 14.524 | 13.749 |
| RS | 80.263 | 90.789 | 90.132 | 91.447 | 90.132 | 78.947 | 81.579 | 86.842 | 89.474 | 84.211 |
| SWJ | 20.000 | 46.667 | 40.000 | 40.000 | 66.667 | 53.333 | 73.333 | 66.667 | 66.667 | 80.000 |
| avg. acc (10) | 70.354 | 74.368 | 78.156 | 78.349 | 77.600 | 70.999 | 72.760 | 69.578 | 75.168 | 73.429 |
| avg. acc | 68.020 | 73.306 | 76.866 | 77.124 | 76.320 | 68.247 | 75.308 | 69.001 | 76.921 | 76.938 |
| avg. rank | 16.03 | 11.20 | 8.90 | 8.83 | 9.40 | 16.80 | 11.83 | 13.90 | 8.67 | 9.13 |
| # of top-1 | 1 | 3 | 4 | 7 | **8** | 0 | 1 | 1 | 3 | 2 |
| # of top-5 | 2 | 10 | 13 | 13 | 12 | 0 | 6 | 3 | 9 | 8 |
| # of top-10 | 5 | 15 | 19 | 17 | 16 | 5 | 11 | 9 | 17 | 16 |
| # of win/draw | 29 | 24 | 20 | 20 | 21 | 28 | 25 | 28 | 24 | 24 |
| # of lose/draw | 2 | 9 | 14 | 13 | 11 | 3 | 7 | 4 | 8 | 8 |
| rank test | 0.000 | 0.001 | 0.056 | 0.034 | 0.015 | 0.000 | 0.000 | 0.000 | 0.004 | 0.001 |

All experiments were run on 1080 Ti with batch size 16 except below:

[8] The experiments were run with batch size 8

[4] The experiments were run with batch size 4

[A] The experiments were run on A100 GPU

Table 5: *(Continued.)* Classification accuracy (%) of TSC models on 30 datasets from the UEA archives. The best and the second-best are highlighted in bold and underline, respectively.

(b) Classification results from TSLANet to MambaSL. "*" indicates "former" in Transformer-based models.

| | CNN-based | | Transformer-based | | | | | | shape-based | Mamba-based | |
| --- | --- | --- | --- | --- | --- | --- | --- | --- | --- | --- | --- |
| | TSLANet (2024) | Time Mixer++ (2025a) | FED* (2022) | ETS* (2022) | Cross* (2023) | PatchTST (2023) | GPT4TS (2023) | iTrans* (2024) | InterpGN (2025) | TSC Mamba (2025) | **MambaSL (proposed)** |
| EC | 31.939 | 34.601 | 33.840 | 33.080 | 43.346 | 32.700 | 32.319 | 31.939 | 28.897 | 30.798 | 42.586 |
| FD | 66.969 | 69.665 | 69.892 | 68.331 | 69.637 | 68.076 | 68.615 | 68.417 | 63.791 | **70.204** | 69.296 |
| HW | **62.000** | 33.647 | 33.412 | 35.647 | 30.353 | 29.529 | 34.118 | 31.294 | 58.706 | 45.059 | 60.824 |
| HB | 79.512 | 80.488 | 78.049 | 78.537 | 79.024 | 73.659 | 79.024 | 78.049 | 80.488 | 78.537 | 80.488 |
| JV | 98.919 | 99.189 | 97.568 | 98.378 | 98.378 | 97.027 | 98.108 | 98.108 | **99.730** | 98.378 | 98.649 |
| PEMS | 89.017 | 89.017 | 89.017 | 87.861 | 89.595[8] | 89.017[4] | 88.439 | 91.329[8] | 86.127 | 90.173 | 85.549 |
| SRS1 | 89.761 | 92.150 | 75.085 | 92.150 | 93.857 | 89.078 | 93.174 | 93.857 | 90.444 | 92.833 | 92.491 |
| SRS2 | 63.889 | 60.556 | 59.444 | 60.000 | 61.667 | 61.667 | 60.556 | 58.889 | 58.333 | 62.222 | **65.000** |
| SAD | 98.045 | 99.545 | 99.454 | 99.000 | 98.863 | 98.818 | 99.591 | 99.318 | 99.864 | 95.816 | **99.955** |
| UW | 94.375 | 90.625 | 79.688 | 89.688 | 88.438 | 89.375 | 89.063 | 90.000 | 88.438 | **95.313** | 93.438 |
| AWR | **99.333** | 99.000 | 92.333 | 98.667 | 98.667 | 98.667 | 98.667 | 99.000 | 99.000 | **99.333** | **99.333** |
| AF | 66.667 | 66.667 | 73.333 | **80.000** | 73.333 | **80.000** | 66.667 | 66.667 | 46.667 | 66.667 | 53.333 |
| BM | **100.000** | **100.000** | 95.000 | **100.000** | **100.000** | 85.000 | **100.000** | 97.500 | **100.000** | **100.000** | **100.000** |
| CT | 99.164 | 99.373 | 97.284 | 98.468 | 98.955 | 98.398 | 99.373 | 98.886 | **99.791** | 99.513 | 99.721 |
| CR | 91.667 | 98.611 | 90.278 | 98.611 | 97.222 | 97.222 | 97.222 | 97.222 | **100.000** | **100.000** | **100.000** |
| DDG | 74.000 | 74.000 | 42.000 | 42.000 | 72.000[4] | 40.000[4] | 66.000 | 60.000[8] | **76.000**[A] | 50.000 | 70.000 |
| EW | 81.679 | 61.069[A] | 63.359[4] | 70.992[4] | 58.015[4] | 57.252 | 66.412 | 59.542 | 82.443[A] | **88.550** | 83.969[4] |
| EP | **100.000** | 94.928 | 92.754 | 92.754 | 95.652 | 98.551 | 89.855 | 97.826 | 97.826 | 99.275 | 97.826 |
| ER | 85.926 | 96.296 | 95.926 | 96.296 | **97.407** | 96.667 | 96.296 | 94.815 | 87.778 | 97.037 | 93.704 |
| FM | 68.000 | 67.000 | **72.000** | 66.000 | 64.000 | 63.000 | 66.000 | 64.000 | 67.000 | 66.000 | 71.000 |
| HMD | 48.649 | **72.973** | 71.622 | 63.514 | 66.216 | 63.514 | 70.270 | 58.108 | 45.946 | 60.811 | 70.270 |
| IW | 10.012 | 64.192 | 65.244 | 64.868 | 73.712 | 60.936 | 61.396 | 73.436 | 67.180 | 68.244[*] | 66.304 |
| LIB | 95.556 | 88.333 | 93.333 | 90.000 | 90.556 | 81.667 | 87.778 | 91.667 | **97.222** | 91.667 | 91.667 |
| LSST | 51.014 | 44.444 | 40.998 | 41.241 | 41.768 | 56.732 | 38.078 | 40.673 | 59.570 | 60.057 | 45.580 |
| MI | **71.000** | 67.000 | 65.000 | 67.000[8] | 68.000[4] | 69.000 | 67.000 | 67.000 | 67.000[A] | 67.000 | 69.000 |
| NATO | **99.444** | 96.111 | 96.111 | 93.333 | 93.889 | 80.556 | 96.111 | 88.889 | 98.889 | 92.222 | 98.889 |
| PD | 99.028 | 98.599 | 98.599 | 98.542 | 98.513 | 98.485 | 98.513 | 98.428 | **99.342** | 98.513 | 99.257 |
| PS | 4.921 | 15.508 | 11.691 | 14.316 | 14.256 | 15.091 | 13.123 | 11.542 | 33.075 | 23.800 | 30.331 |
| RS | **94.079** | 91.447 | 84.868 | 85.526 | 87.500 | 85.526 | 86.842 | 81.579 | 91.447 | 90.789 | 92.763 |
| SWJ | 73.333 | 73.333 | 66.667 | 66.667[8] | 80.000 | **86.667** | 73.333 | 73.333 | 60.000 | 73.333 | 73.333 |
| avg. acc (10) | 77.443 | 74.948 | 71.545 | 74.267 | 75.316 | 72.895 | 74.301 | 74.120 | 75.482 | 75.933 | **78.827** |
| avg. acc | 76.263 | 77.279 | 74.128 | 75.716 | 77.427 | 74.729 | 76.065 | 74.797 | 77.700 | 78.405 | **79.819** |
| avg. rank | 7.80 | 7.03 | 12.33 | 10.63 | 8.50 | 12.60 | 9.50 | 11.83 | 7.87 | 7.10 | **5.30** |
| # of top-1 | 7 | 2 | 1 | 2 | 1 | 2 | 1 | 0 | 7 | 5 | 5 |
| # of top-5 | 16 | 12 | 6 | 2 | 12 | 4 | 6 | 4 | 14 | 12 | **18** |
| # of top-10 | 21 | 24 | 10 | 13 | 17 | 8 | 18 | 10 | 20 | 25 | **27** |
| # of win/draw | 18 | 23 | 23 | 27 | 21 | 24 | 26 | 25 | 21 | 20 | |
| # of lose/draw | 15 | 10 | 7 | 4 | 10 | 7 | 7 | 7 | 14 | 15 | |
| rank test | 0.029 | 0.005 | 0.000 | 0.000 | 0.005 | 0.000 | 0.000 | 0.000 | 0.026 | 0.039 | |

All experiments were run on 1080 Ti with batch size 16 except below:

[8] The experiments were run with batch size 8

[4] The experiments were run with batch size 4

[A] The experiments were run on A100 GPU

[*] Little hyperparameter search was performed since it took over two weeks conducting a single experiment.

## C.2 Full results of model ablation on the UEA benchmark

Table 6 presents the complete ablation results corresponding to Table 1. For a clear comparison, we also performed the hyperparameter grid search for each ablated model. Model rankings were computed within the ablated variants, while additional experimental results are included for reference.

Since removing H2 reduces the number of valid hyperparameter configurations to one-eighth, we extended the search space of Mamba's hyperparameters (`expand` and `d_conv`) and compared the results. As expected, performance improved with broader exploration, yet the model still underperformed on datasets such as `EC` and `NATO`, where the LTI setting remains advantageous.

Table 6: Classification accuracy (%) of ablated models on all 30 datasets from the UEA archives. `d_model`, `d_state`, `expand`, and `d_conv` are four basic hyperparameters that affect the model capacity of Mamba. The best and the second-best are highlighted in bold and underline, respectively.

| | ablation | | | | | | | | additional | | |
|---|---|---|---|---|---|---|---|---|---|---|---|
| | w/o H1 | w/o H2 | w/o H3 | w/o H4 | | | | **proposed** | Mamba | w/ H2 | w/o H2 |
| **kernel size** ✓: $k = 0.02L$ ✗: $k = 3$ | ✗ | ✓ | ✓ | ✓ | ✓ | ✓ | ✓ | ✓ | ✗ | ✗ | ✓ |
| **modular SSM** ✓: modular TV/TI ✗: TV only | ✓ | ✗ | ✓ | ✓ | ✓ | ✓ | ✓ | ✓ | ✗ | ✓ | ✗ |
| **skip connection** ✓: not use D ✗: use D | ✓ | ✓ | ✗ | ✓ | ✓ | ✓ | ✓ | ✓ | ✗ | ✗ | ✓ |
| **aggregation** ✓: adaptive pool. ✗: the others | ✓ | ✓ | ✓ | ✗ (full) | ✗ (avg) | ✗ (max) | ✗ (last) | ✓ | ✗ (full) | ✗ (avg) | ✓ |
| d_model | 32..1024 | 32..1024 | 32..1024 | 32..1024 | 32..1024 | 32..1024 | 32..1024 | 32..1024 | 32..1024 | 32..1024 | 32..1024 |
| d_state | 1..16 | 1..16 | 1..16 | 1..16 | 1..16 | 1..16 | 1..16 | 1..16 | 1..16 | 1..16 | 1..16 |
| expand | 1 | 1 | 1 | 1 | 1 | 1 | 1 | 1 | 1 | 1 | *1,2,4* |
| d_conv | 4 | 4 | 4 | 4 | 4 | 4 | 4 | 4 | 4 | 4 | *1,2,4* |
| # of exp | 240 | 30 | 240 | 240 | 240 | 240 | 240 | 240 | 30 | 240 | 270 |
| EC | 33.080 | 32.319 | 39.544 | 34.601 | 32.700 | 41.825 | 33.080 | **42.586** | 30.418 | 34.221 | 33.460 |
| FD | 69.296 | 68.587 | 69.211 | 70.006 | **70.034** | 69.012 | 66.827 | 69.296 | 68.785 | 69.154 | 69.495 |
| HW | 60.235 | 54.000 | 58.000 | 47.294 | 59.882 | 45.529 | 56.706 | **60.824** | 30.941 | 60.471 | 56.000 |
| HB | **80.976** | 79.024 | 80.000 | 80.000 | 80.488 | **80.976** | 78.537 | 80.488 | 77.073 | 80.976 | 80.976 |
| JV | 98.649 | 98.108 | 98.649 | 98.649 | 98.108 | **98.919** | **98.919** | 98.649 | 97.297 | 97.838 | 98.919 |
| PEMS | 85.549 | 82.659 | 87.283 | **90.751** | 82.659 | 85.549 | 89.595 | 85.549 | 93.064 | 83.815 | 85.549 |
| SRS1 | 92.150 | 89.761 | 91.809 | **93.174** | 91.468 | 90.102 | 87.713 | 92.491 | 92.833 | 91.126 | 91.809 |
| SRS2 | 61.111 | 61.111 | 60.556 | 61.111 | 60.000 | 63.333 | 61.111 | **65.000** | 58.889 | 60.556 | 62.222 |
| SAD | **99.955** | 99.773 | 99.909 | 99.727 | 99.909 | 99.864 | 99.818 | **99.955** | 99.181 | 99.955 | 99.818 |
| UW | 93.125 | 91.250 | 92.188 | 89.063 | **93.750** | 89.375 | 86.250 | 93.438 | 84.375 | 92.813 | 93.438 |
| AWR | **99.333** | **99.333** | 99.000 | 99.000 | **99.333** | 99.000 | 97.000 | **99.333** | 97.667 | 99.000 | 99.333 |
| AF | **73.333** | 46.667 | 53.333 | 66.667 | 53.333 | 53.333 | 53.333 | 53.333 | 66.667 | 46.667 | 60.000 |
| BM | **100.000** | **100.000** | **100.000** | **100.000** | **100.000** | **100.000** | **100.000** | **100.000** | 97.500 | 100.000 | 100.000 |
| CT | 99.582 | 99.443 | 99.513 | 99.095 | **99.791** | 99.234 | 99.582 | 99.721 | 98.816 | 99.791 | 99.582 |
| CR | **100.000** | **100.000** | **100.000** | 98.611 | **100.000** | **100.000** | **100.000** | **100.000** | 97.222 | 100.000 | 100.000 |
| DDG | 70.000 | 66.000 | 74.000 | 64.000 | 72.000 | 76.000 | **80.000** | 70.000 | 52.000 | 72.000 | 74.000 |
| EW | **85.496** | 83.206 | 84.733 | 67.176 | 83.969 | 84.733 | 51.908 | 83.969 | 67.176 | 75.573 | 85.496 |
| EP | 97.826 | 97.826 | 97.101 | 96.377 | 97.826 | **98.551** | 97.826 | 97.826 | 92.754 | 97.101 | 97.826 |
| ER | 93.704 | 91.852 | 94.444 | **95.926** | 95.185 | 91.852 | 92.222 | 93.704 | 94.444 | 96.296 | 94.815 |
| FM | **71.000** | 64.000 | 65.000 | 69.000 | 64.000 | 67.000 | 68.000 | **71.000** | 64.000 | 64.000 | 68.000 |
| HMD | **72.973** | 64.865 | 68.919 | 68.919 | 68.919 | 56.757 | 50.000 | 70.270 | 52.703 | 74.324 | 68.919 |
| IW | 66.304 | 65.952 | 66.220 | 65.124 | 66.084 | 66.392 | **66.584** | 66.304 | 62.684 | 66.396 | 66.756 |
| LIB | **91.667** | **91.667** | 90.556 | 86.111 | 91.111 | 88.889 | 88.889 | **91.667** | 82.778 | 92.778 | 92.222 |
| LSST | 45.580 | 43.593 | **46.513** | 43.390 | 45.174 | 43.593 | 42.701 | 45.580 | 37.307 | 43.917 | 44.282 |
| MI | 65.000 | 65.000 | **70.000** | **70.000** | 65.000 | 66.000 | 65.000 | 69.000 | 65.000 | 63.000 | 66.000 |
| NATO | **98.889** | 97.222 | **98.889** | 97.778 | **98.889** | **98.889** | **98.889** | **98.889** | 95.556 | 98.333 | 98.333 |
| PD | 99.257 | 99.142 | 99.371 | 99.114 | **99.428** | 99.085 | 98.999 | 99.257 | 98.799 | 99.314 | 99.314 |
| PS | 29.615 | 29.764 | 30.063 | 16.880 | **31.792** | 27.528 | 27.140 | 30.331 | 13.809 | 31.226 | 29.764 |
| RS | 92.763 | 90.132 | 93.421 | 90.789 | 90.789 | **94.737** | 88.816 | 92.763 | 84.211 | 94.079 | 90.789 |
| SWJ | **80.000** | 60.000 | 66.667 | 73.333 | 66.667 | 66.667 | 66.667 | 73.333 | 73.333 | 53.333 | 73.333 |
| avg. acc (10) | 77.413 | 75.659 | 77.715 | 76.438 | 76.900 | 76.448 | 75.856 | **78.827** | 73.286 | 77.092 | 77.169 |
| avg. acc | **80.215** | 77.075 | 79.163 | 77.722 | 78.610 | 78.535 | 76.848 | 79.819 | 74.243 | 77.935 | 79.348 |
| avg. rank | 2.53 | 5.93 | 3.53 | 4.90 | 3.60 | 3.90 | 4.87 | **2.40** | | | |
| # of win/draw | 25 | 30 | 22 | 24 | 22 | 22 | 25 | | 26 | 20 | 20 |
| # of lose/draw | 22 | 5 | 14 | 10 | 15 | 12 | 9 | | 5 | 13 | 17 |
| rank test | 0.217 | 0.000 | 0.071 | 0.003 | 0.044 | 0.004 | 0.001 | | 0.000 | 0.011 | 0.106 |

## C.3 COMPARISON WITH PREVIOUSLY REPORTED RESULTS

Although we compared various hyperparameter settings against the original paper to ensure as fair a comparison as possible, our results may not represent those in the original paper due to some differences in basic hyperparameter settings (e.g. fixing the learning rate to 0.001). Therefore, we compared our results with previously reported results, and the results are presented in Table 7.

Among the baseline models, only papers that directly evaluated the classification results are used and mentioned in the table. If multiple papers reported results for a particular model–dataset pair, the highest value was used for comparison. For Hydra and MR+Hydra, the results reported in BORF (Spinnato et al., 2024) were used. Note that we approximated the reported values using the number of test samples since they were typically rounded to one or two decimal places.

The metrics used in Table 7 are as follows:

- **report ≤ ours:** Count of datasets on which our experimental results of grid search achieved higher or equal accuracy compared to the reported results. A value higher than "report ≥ ours" indicates that existing reports likely underestimated the model.
- **report ≥ ours:** Count of datasets on which our experimental results of grid search achieved lower or equal accuracy compared to the reported results. A value higher than "report ≤ ours" indicates that our experiments likely underestimated the model.
- **acc difference:** Average difference between our experimental accuracy and the reported accuracy, calculated only for datasets with paired results. A positive number indicates that our experiments achieved higher accuracy, while a negative number indicates the opposite.

Table 7: Comparison of our experimental results and previous reported results.

(a) Comparison results on TSC-specific models.

| | Distance | | TSC-specific | | | | | | | | | | | |
| | $DTW_D$ (1994) | | ROCKET (2020) | | HC2 (2021) | | Hydra (2023) | | MR+Hydra (2023) | | InterpGN (2025) | | TSCMamba (2025) | |
| | TimesNet | **ours** | TimesNet TSLANet TSCMamba | **ours** | original | **ours** | BORF | **ours** | BORF | **ours** | original | **ours** | original | **ours** |
|---|---|---|---|---|---|---|---|---|---|---|---|---|---|---|
| EC | 32.281 | 32.319 | 45.209 | 44.487 | 79.087 | 52.852 | 54.791 | 54.373 | 58.897 | 60.076 | 30.418 | 28.897 | 62.015 | 30.798 |
| FD | 52.900 | 52.866 | 64.699 | 61.606 | 71.348 | 61.379 | 52.301 | 60.982 | 61.901 | 61.294 | 66.402 | 63.791 | 69.401 | 70.204 |
| HW | 28.600 | 60.706 | 58.800 | 59.294 | 56.341 | 56.000 | 46.600 | 56.353 | 51.400 | 53.529 | 61.882 | 58.706 | 53.306 | 45.059 |
| HB | 71.707 | 71.707 | 75.610 | 75.610 | 72.878 | 78.537 | 76.098 | 76.585 | 73.220 | 77.073 | 79.512 | 80.488 | 76.585 | 78.537 |
| JV | 94.892 | 95.946 | 96.189 | 83.784 | | 98.378 | 97.811 | 97.838 | 97.811 | 98.378 | 99.189 | 99.730 | 97.000 | 98.378 |
| PEMS | 71.098 | 71.098 | 75.087 | 84.393 | 99.827 | 97.688 | 80.289 | 98.266 | 77.514 | 82.659 | 88.439 | 86.127 | 90.173 | 90.173 |
| SRS1 | 77.713 | 77.474 | 90.785 | 85.666 | 87.884 | 89.078 | 86.689 | 86.689 | 95.188 | 94.881 | 92.491 | 90.444 | 92.491 | 92.833 |
| SRS2 | 53.889 | 53.889 | 54.444 | 55.556 | 50.444 | 54.444 | 58.889 | 58.889 | 56.722 | 55.000 | 57.778 | 58.333 | 66.722 | 62.222 |
| SAD | 96.298 | 97.226 | 99.200 | 99.227 | | 99.136 | 98.899 | 99.136 | 99.300 | 99.045 | 99.818 | 99.864 | 99.000 | 95.816 |
| UW | 90.313 | 90.313 | 94.406 | 94.063 | 94.875 | 94.063 | 90.906 | 94.375 | 94.094 | 94.063 | 91.875 | 88.438 | 93.813 | 95.313 |
| AWR | | 98.667 | 99.333 | 99.333 | 99.567 | 99.333 | 99.000 | 99.333 | 99.000 | 99.333 | 99.667 | 99.000 | 97.000 | 99.333 |
| AF | | 20.000 | 20.000 | 6.667 | 28.000 | 20.000 | 13.333 | 26.667 | 6.667 | 6.667 | 33.333 | 46.667 | 67.333 | 66.667 |
| BM | | 97.500 | 100.000 | 100.000 | 99.000 | 100.000 | 100.000 | 100.000 | 100.000 | 100.000 | 100.000 | 100.000 | 100.000 | 100.000 |
| CT | | 98.816 | 98.753 | 99.304 | | 99.304 | 98.600 | 99.164 | 99.199 | 99.373 | 99.791 | 99.791 | 98.997 | 99.513 |
| CR | | 100.000 | 98.611 | 100.000 | 100.000 | 100.000 | 98.611 | 98.611 | 98.611 | 98.611 | 100.000 | 100.000 | 98.611 | 100.000 |
| DDG | | 58.000 | 52.000 | 52.000 | 49.800 | 70.000 | 68.000 | 68.000 | 64.000 | 60.000 | 68.000 | 76.000 | 62.000 | 50.000 |
| EW | | 61.832 | 78.626 | 91.603 | 89.466 | 97.710 | 98.473 | 98.473 | 96.183 | 98.473 | 83.969 | 82.443 | 87.023 | 88.550 |
| EP | | 96.377 | 98.551 | 99.275 | 99.783 | 100.000 | 100.000 | 100.000 | 100.000 | 100.000 | 99.275 | 97.826 | 97.101 | 99.275 |
| ER | | 91.481 | 94.074 | 98.889 | 98.519 | 98.889 | 98.519 | 98.889 | 98.889 | 99.259 | 95.185 | 87.778 | 91.519 | 97.037 |
| FM | | 53.000 | 61.000 | 57.000 | 55.200 | 61.000 | 55.000 | 55.000 | 56.000 | 60.000 | 64.000 | 67.000 | 69.000 | 66.000 |
| HMD | | 18.919 | 50.000 | 51.351 | 39.730 | 48.649 | 21.622 | 51.351 | 36.486 | 36.486 | 48.649 | 45.946 | 71.622 | 60.811 |
| IW | | 48.692 | 10.000 | 38.524 | | 66.796 | 67.900 | 67.052 | 66.700 | 65.744 | 66.752 | 67.180 | 14.300 | 68.244 |
| LIB | | 87.222 | 83.889 | 90.556 | 92.667 | 94.444 | 93.278 | 93.333 | 92.778 | 93.333 | 97.222 | 97.222 | 90.000 | 91.667 |
| LSST | | 55.109 | 54.100 | 64.031 | 63.694 | 66.383 | 60.702 | 66.504 | 63.601 | 64.274 | 60.665 | 59.570 | 60.300 | 60.057 |
| MI | | 50.000 | 53.000 | 54.000 | 53.200 | 50.000 | 50.000 | 54.000 | 48.000 | 51.000 | 64.000 | 67.000 | 62.000 | 67.000 |
| NATO | | 88.333 | 83.333 | 89.444 | 89.222 | 91.111 | 87.222 | 90.000 | 91.722 | 93.333 | 98.889 | 98.889 | 94.444 | 92.222 |
| PD | | 97.713 | 97.341 | 98.228 | 99.557 | 97.656 | 96.801 | 97.856 | 96.801 | 97.284 | 99.142 | 99.342 | 98.542 | 98.513 |
| PS | | 15.121 | 17.599 | 27.826 | 29.427 | 33.015 | 32.401 | 33.164 | 35.401 | 33.642 | 32.180 | 33.075 | 24.659 | 23.800 |
| RS | | 80.263 | 86.184 | 90.789 | 93.026 | 90.132 | 91.382 | 91.447 | 90.789 | 90.132 | 90.789 | 91.447 | 91.447 | 90.789 |
| SWJ | | 20.000 | 46.667 | 46.667 | 44.000 | 40.000 | 40.000 | 40.000 | 60.000 | 66.667 | 53.333 | 60.000 | 73.333 | 73.333 |
| report ≤ ours | 8 | | 23 | | 15 | | 28 | | 21 | | 18 | | 17 | |
| report ≥ ours | 6 | | 12 | | 12 | | 10 | | 15 | | 17 | | 16 | |
| acc difference | 3.385 | | 2.056 | | 0.224 | | 3.320 | | 0.747 | | 0.278 | | 0.080 | |

Table 7: *(Continued.)* Comparison of our experimental results and previous reported results.

(b) Comparison results on foundation models.

| | Foundation | | | | | | | | | | | | | |
| | TimesNet (2023) | | | ModernTCN (2024) | | | TSLANet (2024) | | | TimeMixer++ (2025a) | | GPT4TS (2023) | | |
| | original | TSLANet TSCMamba | **ours** | original | revisit | **ours** | original | TSCMamba | **ours** | original | **ours** | original | TSLANet TSCMamba | **ours** |
|---|---|---|---|---|---|---|---|---|---|---|---|---|---|---|
| EC | 35.703 | 27.719 | 33.460 | 36.312 | 31.901 | 32.319 | 30.418 | | 31.939 | 39.886 | 34.601 | 34.183 | 25.475 | 32.319 |
| FD | 68.601 | 67.469 | 70.148 | 70.800 | 68.700 | 66.998 | 66.771 | | 66.969 | 71.799 | 69.665 | 69.200 | 65.579 | 68.615 |
| HW | 32.106 | 26.176 | 37.529 | 30.600 | 28.400 | 30.118 | 57.882 | | 62.000 | 26.506 | 33.647 | 32.706 | 3.765 | 34.118 |
| HB | 78.000 | 74.488 | 83.902 | 77.220 | 77.122 | 78.049 | 77.561 | | 79.512 | 79.122 | 80.488 | 77.220 | 36.585 | 79.024 |
| JV | 98.405 | 97.838 | 98.649 | 98.811 | 98.108 | 98.649 | 99.189 | | 98.919 | 97.892 | 99.189 | 98.595 | 98.108 | 98.108 |
| PEMS | 89.595 | 88.150 | 87.861 | 89.075 | 83.179 | 86.705 | 83.815 | | 89.017 | 90.983 | 89.017 | 87.919 | 87.283 | 88.439 |
| SRS1 | 91.809 | 77.440 | 92.150 | 93.413 | 92.799 | 93.515 | 91.809 | | 89.761 | 93.106 | 92.150 | 93.208 | 91.468 | 93.174 |
| SRS2 | 57.222 | 52.833 | 60.000 | 60.278 | 61.722 | 60.000 | 61.667 | | 63.889 | 65.611 | 60.556 | 59.389 | 51.667 | 60.556 |
| SAD | 99.000 | 98.358 | 99.545 | 98.699 | 98.099 | 99.500 | 99.909 | | 98.045 | 99.800 | 99.545 | 99.200 | 99.363 | 99.591 |
| UW | 85.313 | 83.125 | 88.438 | 86.688 | 85.906 | 88.438 | 91.250 | | 94.375 | 88.188 | 90.625 | 88.094 | 84.375 | 89.063 |
| AWR | | 96.167 | 98.667 | | | 98.667 | 99.000 | | 99.333 | | 99.000 | | 93.333 | 98.667 |
| AF | | 33.333 | 66.667 | | | 73.333 | 40.000 | | 66.667 | | 66.667 | | 33.333 | 66.667 |
| BM | | 100.000 | 100.000 | | | 100.000 | 100.000 | | 100.000 | | 100.000 | | 92.500 | 100.000 |
| CT | | 97.981 | 98.747 | | | 99.304 | | 98.747 | 99.164 | | 99.373 | | 98.259 | 99.373 |
| CR | | 87.500 | 97.222 | | | 97.222 | 98.611 | | 91.667 | | 98.611 | | 8.333 | 97.222 |
| DDG | | 56.000 | 68.000 | | | 66.000 | | 24.000 | 74.000 | | 74.000 | | 50.000 | 66.000 |
| EW | | 58.779 | 63.359 | | | 64.122 | | 41.221 | 81.679 | | 61.069 | | 48.092 | 66.412 |
| EP | | 78.116 | 96.377 | | | 96.377 | 98.551 | | 100.000 | | 94.928 | | 85.507 | 89.855 |
| ER | | 94.074 | 95.556 | | | 97.407 | | 91.481 | 85.926 | | 96.296 | | 95.926 | 96.296 |
| FM | | 59.400 | 66.000 | | | 67.000 | 61.000 | | 68.000 | | 67.000 | | 57.000 | 66.000 |
| HMD | | 50.000 | 72.973 | | | 66.216 | 52.703 | | 48.649 | | 72.973 | | 18.919 | 70.270 |
| IW | | 10.000 | 64.024 | | | 75.588 | 10.000 | | 10.012 | | 64.192 | | 10.000 | 61.396 |
| LIB | | 77.833 | 88.333 | | | 90.000 | 92.778 | | 95.556 | | 88.333 | | 79.444 | 87.778 |
| LSST | | 59.209 | 44.120 | | | 43.593 | 66.338 | | 51.014 | | 44.444 | | 46.391 | 38.078 |
| MI | | 51.000 | 68.000 | | | 67.000 | 62.000 | | 71.000 | | 67.000 | | 50.000 | 67.000 |
| NATO | | 81.833 | 98.333 | | | 95.556 | 95.556 | | 99.444 | | 96.111 | | 91.667 | 96.111 |
| PD | | 98.190 | 98.914 | | | 98.513 | 98.939 | | 99.028 | | 98.599 | | 97.742 | 98.513 |
| PS | | 18.240 | 14.524 | | | 13.749 | 17.751 | | 4.921 | | 15.508 | | 3.012 | 13.123 |
| RS | | 82.632 | 89.474 | | | 84.211 | 90.789 | | 94.079 | | 91.447 | | 76.974 | 86.842 |
| SWJ | | 53.333 | 66.667 | | | 80.000 | 46.667 | | 73.333 | | 73.333 | | 33.333 | 73.333 |
| report ≤ ours | 8 | 27 | | 4 | 8 | | 19 | 3 | | 4 | | 6 | 29 | |
| report ≥ ours | 2 | 4 | | 6 | 2 | | 8 | 1 | | 6 | | 4 | 2 | |
| acc difference | 1.593 | 9.014 | | (0.761) | 0.835 | | 2.161 | 21.330 | | (0.341) | | 0.330 | 15.617 | |

(c) Comparison results on TSF-origin models.

| | DLinear (2023) | | LightTS (2022) | | MTS-Mixer (2023) | | FEDformer (2022) | | ETSformer (2022) | | Crossformer (2023) | | PatchTST (2023) | | iTransformer (2024) | |
| | TimesNet TSLANet TSCMamba | **ours** | TimesNet | **ours** | ModernTCN | **ours** | TimesNet | **ours** | TimesNet | **ours** | ModernTCN TSLANet TSCMamba | **ours** | ModernTCN TSLANet TSCMamba | **ours** | TimeMixer++ | **ours** |
|---|---|---|---|---|---|---|---|---|---|---|---|---|---|---|---|---|
| EC | 33.460 | 31.179 | 29.696 | 32.319 | 33.802 | 30.798 | 31.217 | 33.840 | 28.099 | 33.080 | 37.985 | 43.346 | 32.814 | 32.700 | 28.099 | 31.939 |
| FD | 67.999 | 69.495 | 67.500 | 68.473 | 70.199 | 68.530 | 65.999 | 69.892 | 66.300 | 68.331 | 68.700 | 69.637 | 68.961 | 68.076 | 66.300 | 68.417 |
| HW | 27.000 | 23.765 | 26.106 | 25.059 | 26.000 | 19.647 | 28.000 | 33.412 | 32.506 | 35.647 | 28.800 | 30.353 | 29.600 | 29.529 | 24.200 | 31.294 |
| HB | 75.122 | 76.585 | 75.122 | 77.561 | 77.122 | 79.024 | 73.707 | 78.049 | 71.220 | 78.537 | 77.610 | 79.024 | 76.585 | 73.659 | 75.610 | 78.049 |
| JV | 97.838 | 96.486 | 96.189 | 96.757 | 94.297 | 95.946 | 98.405 | 97.568 | 95.892 | 98.378 | 99.108 | 98.378 | 98.649 | 97.027 | 96.595 | 98.108 |
| PEMS | 82.081 | 82.659 | 88.382 | 87.861 | 80.925 | 82.081 | 80.925 | 89.017 | 86.012 | 87.861 | 85.896 | 89.595 | 89.306 | 89.017 | 87.919 | 91.329 |
| SRS1 | 88.396 | 92.833 | 89.795 | 92.833 | 91.706 | 78.840 | 88.703 | 75.085 | 89.590 | 92.150 | 92.491 | 93.857 | 90.717 | 89.078 | 90.205 | 93.857 |
| SRS2 | 51.667 | 57.222 | 51.111 | 59.444 | 55.000 | 58.333 | 54.389 | 59.444 | 55.000 | 60.000 | 58.278 | 61.667 | 57.778 | 61.667 | 54.389 | 58.889 |
| SAD | 96.680 | 96.953 | 100.000 | 98.226 | 97.399 | 99.454 | 100.000 | 99.454 | 99.000 | 99.000 | 97.899 | 98.863 | 99.682 | 98.818 | 95.998 | 99.318 |
| UW | 82.094 | 82.813 | 80.313 | 89.063 | 82.313 | 83.125 | 85.313 | 79.688 | 85.000 | 89.688 | 85.313 | 88.438 | 85.813 | 89.375 | 85.906 | 90.000 |
| AWR | 97.333 | 97.667 | | 98.333 | | 99.000 | | 92.333 | | 98.667 | 98.000 | 98.667 | 97.667 | 98.667 | | 99.000 |
| AF | 46.667 | 53.333 | | 73.333 | | 60.000 | | 73.333 | | 80.000 | 46.667 | 73.333 | 53.333 | 80.000 | | 66.667 |
| BM | 85.000 | 87.500 | | 100.000 | | 100.000 | | 95.000 | | 100.000 | 90.000 | 100.000 | 92.500 | 85.000 | | 97.500 |
| CT | 97.347 | 98.120 | | 98.955 | | 97.981 | | 97.284 | | 98.468 | 98.329 | 98.955 | 97.347 | 98.398 | | 98.886 |
| CR | 91.667 | 91.667 | | 93.056 | | 97.222 | | 90.278 | | 98.611 | 84.722 | 97.222 | 84.722 | 97.222 | | 97.222 |
| DDG | 62.000 | 66.000 | | 56.000 | | 68.000 | | 42.000 | | 42.000 | 44.000 | 72.000 | 24.000 | 40.000 | | 60.000 |
| EW | 41.221 | 42.748 | | 54.198 | | 66.412 | | 63.359 | | 70.992 | 54.962 | 58.015 | 54.198 | 57.252 | | 59.542 |
| EP | 60.145 | 65.217 | | 97.101 | | 66.667 | | 92.754 | | 92.754 | 73.188 | 95.652 | 65.942 | 98.551 | | 80.435 |
| ER | 91.481 | 91.852 | | 94.444 | | 77.778 | | 95.926 | | 96.296 | 94.074 | 97.407 | 92.593 | 96.667 | | 94.815 |
| FM | 64.000 | 62.000 | | 64.000 | | 63.000 | | 72.000 | | 66.000 | 64.000 | 64.000 | 62.000 | 63.000 | | 64.000 |
| HMD | 58.108 | 66.216 | | 66.216 | | 70.270 | | 71.622 | | 63.514 | 58.108 | 66.216 | 58.108 | 63.514 | | 58.108 |
| IW | 10.000 | 18.996 | | 71.676 | | 72.424 | | 65.244 | | 64.868 | 10.000 | 73.712 | 10.000 | 60.936 | | 72.456 |
| LIB | 73.333 | 70.556 | | 89.444 | | 34.444 | | 93.333 | | 90.000 | 76.111 | 90.556 | 81.111 | 81.667 | | 91.667 |
| LSST | 35.770 | 34.023 | | 41.525 | | 34.185 | | 40.998 | | 41.241 | 42.818 | 41.768 | 67.798 | 56.732 | | 40.673 |
| MI | 61.000 | 64.000 | | 68.000 | | 66.000 | | 65.000 | | 67.000 | 61.000 | 68.000 | 61.000 | 69.000 | | 67.000 |
| NATO | 93.889 | 96.111 | | 97.222 | | 95.000 | | 96.111 | | 93.333 | 88.333 | 93.889 | 96.667 | 80.556 | | 88.889 |
| PD | 92.939 | 92.481 | | 96.998 | | 46.941 | | 98.599 | | 98.542 | 93.651 | 98.513 | 99.231 | 98.485 | | 98.428 |
| PS | 7.101 | 6.651 | | 16.224 | | 5.428 | | 11.691 | | 14.316 | 7.551 | 14.256 | 11.691 | 15.091 | | 11.542 |
| RS | 78.947 | 78.947 | | 81.579 | | 86.842 | | 84.868 | | 85.526 | 81.579 | 87.500 | 84.211 | 85.526 | | 81.579 |
| SWJ | 60.000 | 53.333 | | 73.333 | | 66.667 | | 66.667 | | 66.667 | 53.333 | 80.000 | 60.000 | 86.667 | | 73.333 |
| report ≤ ours | 21 | | 7 | | 6 | | 6 | | 9 | | 28 | | 18 | | 10 | |
| report ≥ ours | 11 | | 3 | | 4 | | 4 | | 1 | | 3 | | 12 | | 0 | |
| acc difference | 1.237 | | 2.338 | | (1.298) | | 0.879 | | 3.305 | | 9.010 | | 5.262 | | 3.598 | |

## C.4 VISUALIZATION OF TIME VARIANCE ABLATION

Figure 9 shows visualizations of MambaSL's classification accuracy results on the UEA benchmark, tested for various hyperparameter combinations along with TI/TV configurations. Among 240 combinations, 10 sets of hyperparameter combinations that showed high performance regardless of the TI/TV configurations were selected (8×10=80 in total) and depicted in box and line plots. Each line represents the same hyperparameter settings except for time variance configurations.

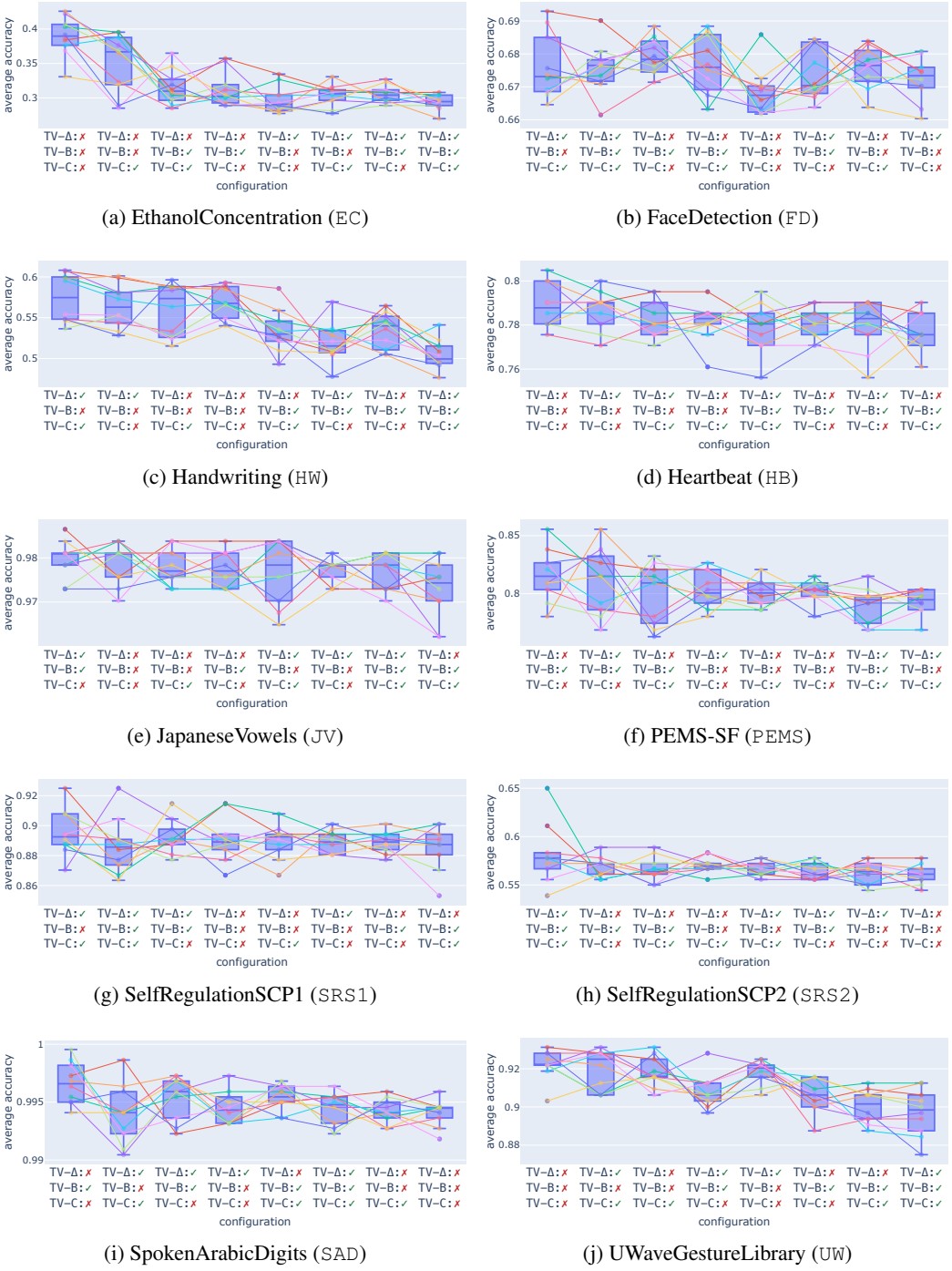

Figure 9: Visualization of classification accuracy of MambaSL along TI/TV configurations on the UEA benchmark, in order of maximum accuracy.

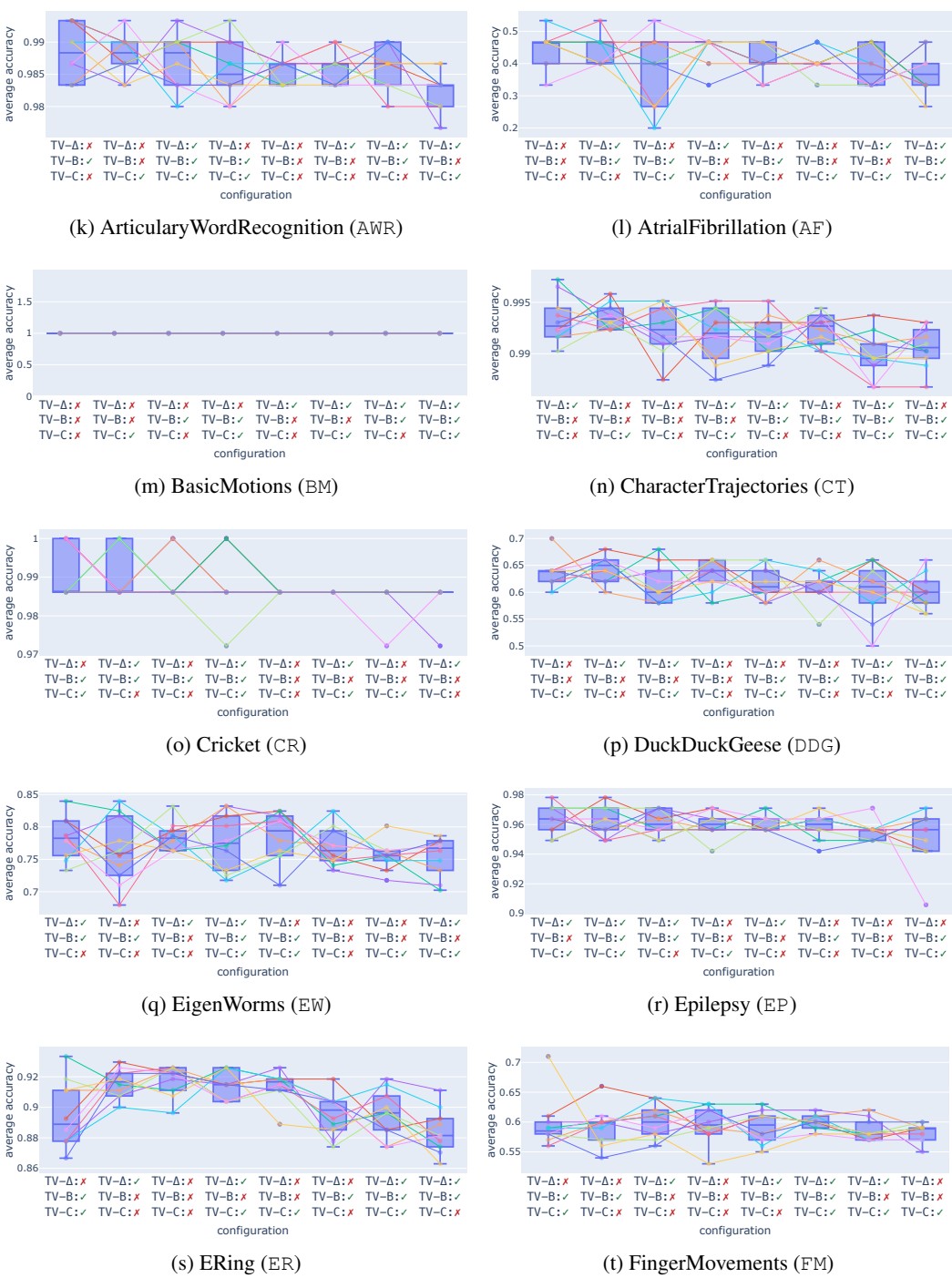

Figure 9: *(Continued.)* Visualization of classification accuracy of MambaSL along TI/TV configurations on the 30 UEA datasets, in order of maximum accuracy.

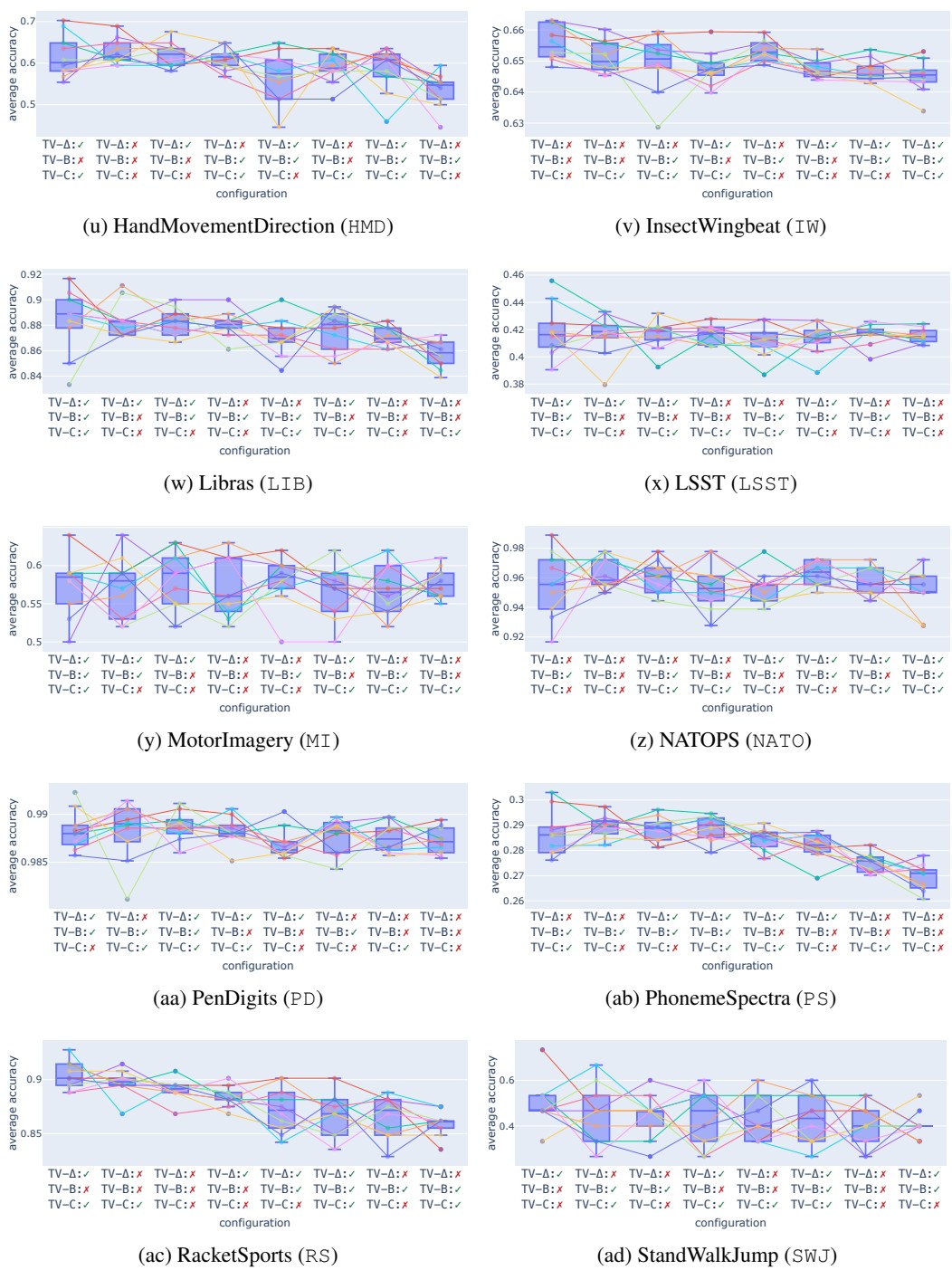

Figure 9: *(Continued.)* Visualization of classification accuracy of MambaSL along TI/TV configurations on the 30 UEA datasets, in order of maximum accuracy.

## C.5 FULL RESULTS OF MAMBASL PERFORMANCE ACCORDING TO MODEL DEPTH

Table 8 presents the complete comparison results corresponding to Table 3. Although using three layers yielded the highest average accuracy on the 10 datasets commonly evaluated in `TSLib`, this observation does not hold when extended to the full set of 30 UEA datasets. When examining the average rank on these 10 datasets, we find that the single-layer configuration consistently outperforms the multi-layer variants, indicating that the higher average accuracy for deeper models is driven by a small subset of datasets. In particular, the `HW` dataset exhibits a nearly 5%p improvement with 2–3 layers compared to a single layer, which disproportionately affects the average accuracy metric.

Table 8: Classification accuracy (%) of MambaSL according to its model depth on 30 datasets from the UEA archives. The best and the second-best are highlighted in bold and underline, respectively.

|  | model depth | | |
|---|---|---|---|
|  | 1 (proposed) | 2 | 3 |
| EC | 42.586 | 41.065 | 41.825 |
| FD | 69.296 | 68.785 | 69.098 |
| HW | 60.824 | 65.765 | 66.353 |
| HB | 80.488 | 80.488 | 80.488 |
| JV | 98.649 | 98.649 | 98.378 |
| PEMS | 85.549 | 85.549 | 86.705 |
| SRS1 | 92.491 | 92.150 | 91.126 |
| SRS2 | 65.000 | 62.778 | 62.222 |
| SAD | 99.955 | 99.864 | 99.864 |
| UW | 93.438 | 93.438 | 93.125 |
| AWR | 99.333 | 99.333 | 99.333 |
| AF | 53.333 | 53.333 | 60.000 |
| BM | 100.000 | 100.000 | 100.000 |
| CT | 99.721 | 99.652 | 99.721 |
| CR | 100.000 | 100.000 | 100.000 |
| DDG | 70.000 | 70.000 | 68.000 |
| EW | 83.969 | 83.206 | 85.496 |
| EP | 97.826 | 97.826 | 97.101 |
| ER | 93.704 | 95.185 | 94.074 |
| FM | 71.000 | 67.000 | 65.000 |
| HMD | 70.270 | 68.919 | 68.919 |
| IW | 66.304 | 64.772 | 65.596 |
| LIB | 91.667 | 91.111 | 89.444 |
| LSST | 45.580 | 44.039 | 42.539 |
| MI | 69.000 | 67.000 | 66.000 |
| NATO | 98.889 | 98.889 | 98.333 |
| PD | 99.257 | 99.171 | 99.200 |
| PS | 30.331 | 31.435 | 30.987 |
| RS | 92.763 | 93.421 | 92.763 |
| SWJ | 73.333 | 66.667 | 66.667 |
| avg. acc (10) | 78.827 | 78.853 | **78.918** |
| avg. acc | **79.819** | 79.316 | 79.279 |
| avg. rank (10) | **1.30** | 1.90 | 2.10 |
| avg. rank | **1.33** | 1.80 | 2.07 |
| # of win/draw |  | 26 | 24 |
| # of lose/draw |  | 15 | 12 |
| rank test |  | 0.035 | 0.004 |

## C.6 Classification results on the UEA benchmark under InceptionTime setting

To address the data-leakage concern raised for `TSLib`, we trained MambaSL under the Inception-Time protocol, which selects the checkpoint with the lowest training loss (Ismail Fawaz et al., 2020). For comparison, we selected the top three high-performing models from each source—HC2 (Middlehurst et al., 2021), BORF (Spinnato et al., 2024), and our own experiments—to form a representative conventional TSC baseline set. For HC2, we used only the results from fold 0, the official train–test split[3], and for BORF, we approximated the reported accuracies using the number of test samples as described in appendix C.3.

Table 9 summarizes the results. Although MambaSL achieved slightly lower average accuracy than HC2 and Hydra, it remained competitive, achieving the largest number of top-1 results (15 datasets) and outperforming every non-DL baseline in direct head-to-head comparisons. These findings indicate that MambaSL maintains strong performance under leakage-free model selection.

Since all experimental hyperparameters and optimization settings were fixed to the same values as in our original experiments, the performance differences between Tables 5 and 9 arise solely from the change in the model selection criterion.

Table 9: UEA classification accuracy (%) of MambaSL using the InceptionTime protocol (selecting the checkpoint with the lowest training loss). The comparison set includes the top three models from each HC2 (Middlehurst et al., 2021), BORF (Spinnato et al., 2024), and our own evaluations. The best and the second-best are highlighted in bold and underline, respectively.

| | HC2 reported (2021) | | | BORF reported (2024) | | | our experiments | | | |
|---|---|---|---|---|---|---|---|---|---|---|
| | CIF (2020) | DrCIF (2021) | HC2 (2021) | Mini Rocket (2021) | DrCIF (2021) | MR+ Hydra (2023) | HC2 (2021) | Hydra (2023) | MR+ Hydra (2023) | MambaSL (proposed) |
| EC | 73.384 | 69.202 | **77.186** | 48.289 | 63.118 | 58.935 | 52.852 | 54.373 | 60.076 | 39.163 |
| FD | 62.713 | 62.003 | 66.033 | 58.201 | 59.109 | 61.890 | 61.379 | 60.982 | 61.294 | **68.331** |
| HW | 35.647 | 34.588 | 54.824 | 51.765 | 36.235 | 51.412 | 56.000 | 56.353 | 53.529 | 59.294 |
| HB | 78.049 | **79.024** | 73.171 | 74.146 | **79.024** | 73.171 | 78.537 | 76.585 | 77.073 | 76.585 |
| JV | | | | **99.189** | 97.297 | 97.838 | 98.378 | 97.838 | 98.378 | 98.649 |
| PEMS | **100.000** | **100.000** | **100.000** | 82.659 | **100.000** | 77.457 | 97.688 | 98.266 | 82.659 | 83.237 |
| SRS1 | 86.007 | 87.713 | 89.078 | 91.809 | 87.031 | **95.222** | 89.078 | 86.689 | 94.881 | 88.737 |
| SRS2 | 50.000 | 49.444 | 50.000 | 54.444 | 51.111 | 56.667 | 54.444 | 58.889 | 55.000 | **61.111** |
| SAD | | | | 99.318 | 97.817 | 99.318 | 99.136 | 99.136 | 99.045 | **99.727** |
| UW | 92.500 | 90.938 | 92.813 | 93.125 | 93.750 | 94.063 | 94.063 | 94.375 | 94.063 | 92.500 |
| AWR | 98.333 | 98.000 | **99.333** | **99.333** | **99.333** | **99.333** | **99.333** | **99.333** | **99.333** | **99.333** |
| AF | 33.333 | 33.333 | 26.667 | 13.333 | 40.000 | 6.667 | 20.000 | 26.667 | 6.667 | **46.667** |
| BM | **100.000** | **100.000** | **100.000** | **100.000** | **100.000** | **100.000** | **100.000** | **100.000** | **100.000** | **100.000** |
| CT | | | | 99.304 | 98.120 | 99.234 | 99.304 | 99.164 | 99.373 | **99.582** |
| CR | 98.611 | 98.611 | **100.000** | 98.611 | 98.611 | 98.611 | **100.000** | **100.000** | 98.611 | **100.000** |
| DDG | 44.000 | 54.000 | 56.000 | **70.000** | 48.000 | 64.000 | **70.000** | 68.000 | 60.000 | **70.000** |
| EW | 91.603 | 92.366 | 94.656 | 93.893 | 93.130 | 96.183 | 97.710 | **98.473** | **98.473** | 83.206 |
| EP | 98.551 | 97.826 | **100.000** | **100.000** | **100.000** | **100.000** | **100.000** | **100.000** | **100.000** | 97.826 |
| ER | 98.148 | **99.259** | 98.889 | 97.778 | **99.259** | 98.889 | 98.889 | 98.889 | **99.259** | 92.593 |
| FM | 52.000 | 60.000 | 53.000 | 57.000 | 47.000 | 56.000 | 61.000 | 55.000 | 60.000 | **64.000** |
| HMD | 59.459 | 52.703 | 47.297 | 39.189 | 51.351 | 36.486 | 48.649 | 51.351 | 36.486 | **62.162** |
| IW | | | | 67.300 | 66.400 | 66.700 | 66.796 | 67.052 | 65.744 | 64.524 |
| LIB | 91.111 | 89.444 | 93.333 | 91.667 | 89.444 | 92.778 | **94.444** | 93.333 | 93.333 | 90.556 |
| LSST | 57.259 | 55.596 | 64.274 | 65.085 | 56.488 | 63.585 | 66.383 | **66.504** | 64.274 | 25.182 |
| MI | 50.000 | 44.000 | 54.000 | 50.000 | 50.000 | 48.000 | 50.000 | 54.000 | 51.000 | **62.000** |
| NATO | 85.556 | 84.444 | 89.444 | 94.444 | 82.778 | 91.667 | 91.111 | 90.000 | 93.333 | **98.889** |
| PD | 96.741 | 97.656 | 97.913 | | 97.313 | 96.798 | 97.656 | 97.856 | 97.284 | **99.228** |
| PS | 26.543 | 30.778 | 29.049 | 29.407 | **39.994** | 35.401 | 33.015 | 33.164 | 33.642 | 29.824 |
| RS | 88.158 | 90.132 | 90.789 | 85.526 | 88.816 | 90.789 | 90.132 | **91.447** | 90.789 | 90.789 |
| SWJ | 40.000 | 53.333 | 46.667 | 40.000 | 40.000 | 60.000 | 40.000 | 40.000 | **66.667** | 60.000 |
| # of win/draw | 18 | 19 | 17 | 19 | 20 | 20 | 18 | 19 | 18 | |
| # of lose/draw | 10 | 9 | 13 | 13 | 12 | 14 | 16 | 15 | 14 | |
| rank test | 0.081 | 0.088 | 0.173 | 0.046 | 0.072 | 0.082 | 0.307 | 0.240 | 0.221 | |
| acc difference | 2.058 | 1.416 | (0.123) | 2.061 | 1.777 | 1.218 | (0.076) | (0.334) | 0.470 | |

---

[3]The full set of reported results is available at timeseriesclassification.com.

The higher performance observed in our re-evaluation of certain non-DL baselines in Table 9 stems from running three random seeds and reporting the best-performing run. Using only a single seed (seed 0) yields results comparable to those originally reported for HC2 and BORF.

To further assess stability, we conducted two additional trials for MambaSL and report the mean and standard deviation across three trials in Table 10.

In contrast to Table 9, where HC2 and Hydra showed the strongest performance based on the best-performing trial, averaging across all trials indicates that MR+Hydra achieves the strongest mean performance and exhibits more consistent results. Nevertheless, MambaSL remains competitive, outperforming all conventional non-DL baselines in direct head-to-head comparisons and achieving the largest number of top-1 results (13 datasets). The Wilcoxon signed-rank test indicates no significant difference between MambaSL and MR+Hydra, suggesting comparable overall effectiveness under this protocol.

Table 10: Mean and standard deviation of UEA classification accuracy (%) of MambaSL using the InceptionTime protocol (selecting the checkpoint with the lowest training loss). The comparison set includes the four non-DL models from our own evaluations. The best and the second-best are highlighted in bold and underline, respectively.

| | our experiments | | | | |
| | ROCKET (2020) | HC2 (2021) | Hydra (2023) | MR+Hydra (2023) | **MambaSL (proposed)** |
| --- | --- | --- | --- | --- | --- |
| EC | **99.333** $\pm$**0.000** | 99.111 $\pm$0.385 | 99.222 $\pm$0.192 | 99.222 $\pm$0.192 | 99.111 $\pm$0.385 |
| FD | 6.667 $\pm$0.000 | 17.778 $\pm$3.849 | 17.778 $\pm$7.698 | 6.667 $\pm$0.000 | **42.222** $\pm$**7.698** |
| HW | **100.000** $\pm$**0.000** | **100.000** $\pm$**0.000** | **100.000** $\pm$**0.000** | **100.000** $\pm$**0.000** | **100.000** $\pm$**0.000** |
| HB | 99.257 $\pm$0.040 | 98.793 $\pm$0.464 | 99.071 $\pm$0.106 | 99.327 $\pm$0.040 | **99.513** $\pm$**0.070** |
| JV | **100.000** $\pm$**0.000** | 99.537 $\pm$0.802 | 99.537 $\pm$0.802 | 98.611 $\pm$0.000 | **100.000** $\pm$**0.000** |
| PEMS | 48.667 $\pm$3.055 | 59.333 $\pm$13.614 | 50.000 $\pm$19.079 | 56.667 $\pm$3.055 | **65.333** $\pm$**6.429** |
| SRS1 | 90.331 $\pm$1.166 | 94.402 $\pm$4.473 | 93.639 $\pm$4.341 | **97.201** $\pm$**1.166** | 82.188 $\pm$1.166 |
| SRS2 | 98.792 $\pm$0.418 | **100.000** $\pm$**0.000** | **100.000** $\pm$**0.000** | **100.000** $\pm$**0.000** | 97.343 $\pm$0.837 |
| SAD | 98.642 $\pm$0.428 | 98.395 $\pm$0.566 | **98.765** $\pm$**0.214** | 98.395 $\pm$0.932 | 91.728 $\pm$1.497 |
| UW | 42.966 $\pm$1.742 | 50.824 $\pm$3.189 | 50.190 $\pm$3.627 | **58.175** $\pm$**2.662** | 37.896 $\pm$1.335 |
| AWR | 61.076 $\pm$0.512 | 55.789 $\pm$4.842 | 57.454 $\pm$4.565 | 60.036 $\pm$1.122 | **67.263** $\pm$**0.928** |
| AF | 55.000 $\pm$2.000 | 55.333 $\pm$6.028 | 53.667 $\pm$2.309 | **59.333** $\pm$**1.155** | 58.667 $\pm$6.807 |
| BM | 49.550 $\pm$3.121 | 36.486 $\pm$10.554 | 38.739 $\pm$15.368 | 34.234 $\pm$2.064 | **63.514** $\pm$**2.341** |
| CT | **58.863** $\pm$**0.378** | 49.451 $\pm$5.831 | 53.098 $\pm$5.335 | 52.588 $\pm$0.941 | 58.667 $\pm$1.087 |
| CR | 74.472 $\pm$1.228 | **76.260** $\pm$**2.200** | 75.772 $\pm$1.015 | 75.610 $\pm$1.291 | 72.683 $\pm$3.683 |
| DDG | 38.348 $\pm$0.189 | 64.677 $\pm$3.635 | 61.279 $\pm$5.004 | **65.692** $\pm$**0.074** | 64.264 $\pm$0.227 |
| EW | 82.883 $\pm$0.780 | 94.324 $\pm$7.022 | 90.270 $\pm$6.642 | 97.928 $\pm$0.563 | **98.198** $\pm$**0.780** |
| EP | 90.185 $\pm$0.321 | 93.148 $\pm$1.398 | 92.222 $\pm$0.962 | **93.333** $\pm$**0.000** | 89.815 $\pm$1.283 |
| ER | 63.869 $\pm$0.247 | 62.693 $\pm$3.199 | **64.355** $\pm$**3.145** | 63.328 $\pm$1.071 | 18.194 $\pm$6.194 |
| FM | 51.667 $\pm$2.082 | 47.667 $\pm$4.041 | 52.000 $\pm$2.000 | 50.333 $\pm$0.577 | **60.667** $\pm$**4.163** |
| HMD | 88.333 $\pm$1.111 | 89.074 $\pm$1.951 | 88.333 $\pm$1.470 | 91.481 $\pm$1.604 | **97.778** $\pm$**1.470** |
| IW | 83.815 $\pm$0.578 | 85.356 $\pm$10.927 | **91.908** $\pm$**10.029** | 80.539 $\pm$2.030 | 82.659 $\pm$3.218 |
| LIB | 98.132 $\pm$0.100 | 96.884 $\pm$0.722 | 97.484 $\pm$0.619 | 97.170 $\pm$0.103 | **99.123** $\pm$**0.259** |
| LSST | 27.378 $\pm$0.517 | 31.663 $\pm$1.849 | 31.057 $\pm$1.827 | **32.995** $\pm$**0.707** | 28.780 $\pm$1.002 |
| MI | **90.789** $\pm$**0.000** | 87.939 $\pm$2.310 | 90.789 $\pm$0.658 | 89.035 $\pm$1.005 | 89.912 $\pm$1.519 |
| NATO | 85.324 $\pm$0.341 | 87.827 $\pm$1.097 | 86.121 $\pm$0.710 | **94.084** $\pm$**0.859** | 87.372 $\pm$2.365 |
| PD | 52.407 $\pm$3.349 | 52.407 $\pm$1.786 | 54.630 $\pm$4.170 | 53.889 $\pm$0.962 | **56.667** $\pm$**4.194** |
| PS | 99.151 $\pm$0.131 | 98.530 $\pm$0.533 | 98.909 $\pm$0.355 | 98.939 $\pm$0.095 | **99.576** $\pm$**0.131** |
| RS | 46.667 $\pm$0.000 | 37.778 $\pm$3.849 | 40.000 $\pm$0.000 | **57.778** $\pm$**15.396** | 53.333 $\pm$11.547 |
| SWJ | 93.646 $\pm$0.477 | 92.188 $\pm$1.740 | 93.021 $\pm$2.081 | **93.750** $\pm$**0.541** | 91.979 $\pm$0.902 |
| # of win/draw | 18 | 18 | 18 | 16 | |
| # of lose/draw | 14 | 14 | 13 | 15 | |
| rank test | 0.127 | 0.169 | 0.185 | 0.461 | |
| acc difference | 2.608 | 1.360 | 1.171 | (0.063) | |

## C.7 CLASSIFICATION RESULTS ON ADFTD AND FLAAP

While the UEA archive provides a standardized benchmark for multivariate TSC, it mainly consists of datasets released between 1999 and 2018, many of which are relatively clean and small in scale (Bagnall et al., 2018). To evaluate the robustness of our model beyond these settings, we additionally tested MambaSL on two more recent and domain-specific TSC datasets.

Following the evaluation protocol of Medformer (Wang et al., 2024a), we selected ADFTD (Miltiadous et al., 2023) from the medical domain and FLAAP (Kumar & Suresh, 2022) from the human activity recognition (HAR) domain. A summary of both datasets is provided in Table 11.

Table 11: Summary of ADFTD and FLAAP datasets

| Dataset | Domain | Samples | Length | Variables | Classes | File Size |
|---------|--------|---------|--------|-----------|---------|-----------|
| ADFTD (2023) | EEG | 69752 | 256 | 3 | 19 | 2.52GB |
| FLAAP (2022) | HAR | 13123 | 100 | 10 | 6 | 60.2MB |

Preprocessing and experimental settings were aligned with the official Medformer implementation. We first evaluated MambaSL using the 240 hyperparameter configurations employed for the UEA experiments, after which we conducted four additional trials on some best-performing configurations, resulting in a total of five runs. Table 12 reports the average accuracy and F1-score across these trials. For comparison, we also include the results of the strongly performing baselines reported by Medformer (Wang et al., 2024a): Crossformer (Zhang & Yan, 2023), Reformer (Kitaev et al., 2020), Transformer (Vaswani et al., 2017), TCN (Bai et al., 2018), ModernTCN (Luo & Wang, 2024), and Mamba (Gu & Dao, 2024). Since the original Medformer paper did not report results for TCN, ModernTCN, and Mamba on ADFTD, these entries are left blank.

As shown in Table 12, MambaSL performed competitively against previous baselines on both datasets, and achieved the highest accuracy on FLAAP. In particular, we observe more than a 10%p improvement over the vanilla Mamba. Taken together, the results on ADFTD and FLAAP show that MambaSL remains stable across domains and dataset scales, reinforcing that its performance is not limited to the characteristics of the UEA archive.

Table 12: Classification accuracy and F1-score (%) of MambaSL and other baselines on ADFTD and FLAAP datasets. The best and the second-best are highlighted in bold and underline, respectively.

| | ADFTD | | FLAAP | |
|---|---|---|---|---|
| | accuracy | F1-score | accuracy | F1-score |
| Crossformer (2023) | 50.45 $\pm$1.31 | 45.50 $\pm$1.70 | 75.84 $\pm$0.52 | 75.52 $\pm$0.66 |
| Reformer (2020) | 50.78 $\pm$1.17 | 47.94 $\pm$1.59 | 71.65 $\pm$1.27 | 71.14 $\pm$1.45 |
| Transformer (2017) | 50.47 $\pm$2.14 | 48.09 $\pm$1.59 | 74.96 $\pm$1.25 | 74.49 $\pm$1.39 |
| TCN (2018) | | | 66.48 $\pm$1.66 | 65.29 $\pm$1.74 |
| ModernTCN (2024) | | | 74.80 $\pm$0.96 | 74.35 $\pm$0.85 |
| Mamba (2024) | | | 64.87 $\pm$2.78 | 64.14 $\pm$2.70 |
| Medformer (2024a) | **53.27** $\pm$**1.54** | **50.65** $\pm$**1.51** | 76.44 $\pm$0.64 | 76.25 $\pm$0.65 |
| **MambaSL (proposed)** | 51.68 $\pm$0.89 | 48.73 $\pm$3.05 | **77.47** $\pm$**1.03** | **77.59** $\pm$**1.15** |

