# OpenReview forum: "MambaSL: Exploring Single-Layer Mamba for Time Series Classification"
_ICLR.cc/2026/Conference — ICLR 2026 Poster_

### Official Review · Reviewer_GbtJ · 2025-10-31

**Soundness:** 4
**Presentation:** 4
**Contribution:** 3
**Rating:** 8
**Confidence:** 2

**Summary:**

This paper introduces MambaSL, a single-layer Mamba architecture for time series classification. The authors walk through the Mamba architecture in detail, explaining the individual components and the intuition behind how these components related to time series classification. They then propose their method which makes use of these observations in four succinct hypotheses that they use to build MambaSL. They then demonstrate state-of-the-art results across the UEA dataset, a common time series classification dataset, and examine their hypotheses through detailed ablations.

**Strengths:**

- The authors walk through the Mamba architecture in incredible detail, explaining the components that are necessary to understand their newly proposed architecture. This is greatly appreciated especially for readers with limited knowledge of SSMs.
- The architecture choices are well-motivated and explained. It’s greatly appreciated that authors are succinct and organized in their description of augmentations to the Mamba architecture, and the ablation results support their decisions.
- The work puts incredible emphasis on reproducibility and goes to great lengths to discuss the differences in reported vs. optimized accuracy for baseline methods. I found this incredibly rigorous and greatly appreciated, boosting my confidence in the MambaSL performance given their careful considerations of baselines. In addition, the number of baselines included is quite substantial and covers a wide range of methods.
- This work is overall very principled and novel, with authors clearly laying out the augmentations made on top of the Mamba architecture that are suitable for time series. I think this paves the way for more exploration in this space, and it lays forth a foundation for rigorous and proper benchmarking and principled development of time series models.

**Weaknesses:**

- The experiments could use errors bars for presentation of results across datasets. Many of the performances are very close to each other on Figure 4, and it’s unclear whether the difference between MambaSL is statistically significant from other methods. In addition, error bars are needed in ablations to understand significance of differences in the dataset.
- The model seems to not transfer well to variable-length settings, an assumption made implicitly in Hypothesis 1 where the k value is chosen based on the length of the dataset. Can authors comment on the ability of the model in variable-length settings?
- The UEA dataset, while extensive, contains many small and curated datasets that are not representative of many real-world time series datasets. Did the authors test the method on other datasets for other challenging tasks? The work could use some demonstration on a frontier dataset, such as one released recently that represents a challenging real-world task.

**Questions:**

- Did authors test MambaSL on datasets with very long samples? One benefit of the Mamba architecture and other SSMs are the ability to capture long contexts; this could be a beneficial demonstration but is also not required.
- The UMAP showing comparisons of results across datasets is very cool! I’d love to see this UMAP reduction done along the dataset dimension as well to understand which datasets on which MambaSL performs well compared to non-DL vs. DL methods. This is not required at all but could also be an interesting analysis.
- The authors focus on time series classification in this case, but could this be extended to forecasting?
- Further, does the work easily extend to irregular time series, where time points are collected in irregular intervals. Can the architecture make use of the irregularity in observations when making predictions?

---

> ### Author Response · Authors · 2025-11-20
> **Official Comment by Authors (part 1/3)**
>
> Dear reviewer GbtJ,
>
> Before addressing your questions, we would like to sincerely thank you for raising issues that would be impossible to identify without a precise understanding of both Mamba and MambaSL. Your comments demonstrate a clear grasp of what we aimed to argue in this work.
>
> You correctly highlighted the two points we most wanted to emphasize:
> 1. that Mamba can be improved with only minimal design modifications [S2,S4], and
> 2. that we placed great importance on running the experiments as rigorously and principled as possible [S3,S4].
> We were also delighted by your comment [S1] that the theoretical explanations of SSMs—though relatively uncommon in time series literature—were "necessary to understand their newly proposed architecture," which suggests that our effort to write for readers without prior Mamba knowledge paid off.
>
>
> We now respond to your comments by grouping them into three categories:
> 1. Visualization of the Experimental Results
> 2. Structural Strength/Weakness of MambaSL for Various Types of Sequence Data
> 3. Extensibility for Different Domain and Task
>
> &nbsp;
>
> ---
> ## Question Group 1 : Visualization of the Experimental Results
> ---
>
> ### [W1] Recommendation to add error bar in both `Figure 4 (UEA barplot)` and `Table 1 (Ablation result)`.
>
> Thank you for the suggestion and for proposing a concrete fix. As you noted, adding error bars can indeed help compare models whose performances appear close. However, because UEA datasets vary drastically in difficulty, *the error bars become extremely long and substantially reduce readability in the bar plot*.
> Instead, we updated **`Figure 4 (UEA barplot)` by refining the y-tick intervals** to make model-to-model differences easier to compare, and added the requested error-bar version to the supplementary material.
> For **`Table 1 (Ablation result)`**, we also **added Wilcoxon test values** to indicate statistical significance, as you suggested.
>
> &nbsp;
>
> ### [Q2] What about UMAP visualization done along the dataset dimension, similar to `Figure 5 (UMAP scatterplot)`?
>
> We appreciate your interest in the visualization. Color-coding the full accuracy table made the separation between 'easy for non-DL' and 'hard for non-DL' datasets particularly obvious, which motivated us to attempt this visualization. We had only considered the model axis and had not thought of reversing the view—your suggestion was excellent.
>
> We therefore added your requested **`Figure 6 (UMAP scatterplot along dataset axis)` and its description right after `Figure 5`**.
> Compared with `Appendix C.1 (full UEA results)`, one can see that the lower-right cluster ( $\mathtt{FD,JV,SRS2,AF,FM,HMD,MI,SWJ}$ ) consists of datasets on which *all non-DL methods consistently perform poorly*, whereas those toward the upper-left tend to be datasets where non-DL methods achieve comparatively higher accuracy.
>
>
> &nbsp;

---

> ### Author Response · Authors · 2025-11-23
> **Official Comment by Authors (part 2/3)**
>
> ---
> ## Question Group 2 : Structural Strength/Weakness of MambaSL for Various Types of Sequence Data
> ---
>
> ### [W2] MambaSL seems to not transfer well to variable-length settings due to H1 (choosing $k$ proportional to maximum seq length)
>
> We fully agree with the comment. Using 2% of the maximum sequence length is not optimal for all datasets, which is why we noted in `Section 6 (Conclusion)` that "prior knowledge of input embedding may constrain kernel size, making receptive-field scaling with sequence length suboptimal."
> This may stem either from the variation in sequence lengths or from domain-specific receptive-field needs.
> For example, should all samples be embedded with window size 4 (=2% of 182) in the CharacterTrajectories dataset, even though some samples have lengths as short as 60?
> Do we need a kernel of size 360 for EigenWorms, which has length 17,984?
> Thus, the **2% value was chosen only for experimental uniformity**, and real-world applications should incorporate domain knowledge.
>
>
> What we **wanted to argue in H1** is that, when applying Mamba to time series data, each data point often carries *much less contextual information* than a token in NLP. If we adopt a small fixed kernel such as the $k=3$ embedding used by $\mathtt{TSLib}$, the gating mechanism may fail to extract the appropriate information from the SSM state, especially for high-sampling-rate signals.
> Therefore, **the receptive field used to form token embeddings should be sufficiently large for Mamba’s gating mechanism to function properly**, and in many real-world cases this means using a window larger than 3.
>
> At the same time, we intentionally avoid overly large kernels so that token embeddings remain sufficiently fine-grained. Compared with InterpGN—which uses patterns at 10%, 20%, or even 50% of the full sequence—our 2% receptive field is relatively small.
>
> &nbsp;
>
>
> ### [Q1] Did authors test MambaSL on datasets with very long samples?
>
> Although we could not test sequences as long as the $10^6$ tokens used in the original Mamba paper, we did evaluate on UEA datasets with long inputs—EigenWorms has length 17,984.
> More directly, we computed average ranks **only for the six datasets whose lengths exceed 1000** (EigenWorms, MotorImagery, StandWalkJump, EthanolConcentration, Cricket, SelfRegulationSCP2). As shown below, **MambaSL’s rank improves from 5.30 (overall) to 3.33** on this subset:
>
> |  | avg. acc | avg. rank |
> |---|---:|---:|
> | DTW_D | 53.007 | 15.00 |
> | ROCKET | 65.385 | 10.33 |
> | HC2 | 65.834 | 11.00 |
> | Hydra | 67.622 | 8.83 |
> | MR+Hydra | 71.638 | 9.83 |
> | Dlinear | 56.692 | 18.00 |
> | LightTS | 63.392 | 11.50 |
> | MTS-Mixer | 64.239 | 13.33 |
> | TimesNet | 64.785 | 9.33 |
> | ModernTCN | 66.777 | 8.67 |
> | TSLANet | 68.918 | 8.33 |
> | TimeMixer++ | 65.862 | 8.00 |
> | FEDformer | 63.098 | 13.17 |
> | ETSformer | 66.058 | 8.83 |
> | Crossformer | 68.042 | 7.33 |
> | PatchTST | 67.418 | 8.00 |
> | GPT4TS | 66.140 | 8.33 |
> | iTransformer | 64.654 | 11.33 |
> | InterpGN | 66.112 | 11.17 |
> | TSCMamba | 70.317 | 6.50 |
> | MambaSL | 72.315 | 3.33 |
>
>
> We also note, as mentioned on `line 247`, that we adjust only the kernel size of the input embedding, unlike patch-based models such as PatchTST that increase stride. Thus, **MambaSL ingests longer and more redundant raw sequences** than patch-based models.
> For example, Crossformer reduces EthanolConcentration from length 1751 to roughly 146 tokens via window size 12, whereas MambaSL feeds the **sequence of length 1751** to Mamba.
> Despite this, runtime remains similar, further highlighting Mamba’s efficiency as a backbone.
>
> &nbsp;
>
> ### [Q4] Can MambaSL be applied to time series data with irregular interval?
>
> Vanilla Mamba’s main contribution is its selective mechanism, which allows *time-varying* input processing by inferring $\Delta_t$ from the input rather than fixing step size $\Delta$ as in LTI SSMs.
> Thus, even before MambaSL, **Mamba itself is inherently designed to handle irregular intervals**.
> For example, pronunciations of the same word vary in speed across speakers, implying that the underlying phoneme timings differ. Mamba would naturally infer smaller $\Delta_t$ for faster pronunciations.
>
> The difference between the "irregular time series" you refer to and natural language is largely whether the irregularity is explicitly observable or implicitly embedded.
> Therefore, we believe MambaSL is fully applicable to time series with irregular intervals.
> Instead, unlike NLP, many time series domains contain scenarios where strict equal-spacing in time is crucial (e.g., EthanolConcentration), and this motivated us to explicitly incorporate such considerations into H2.
>
> &nbsp;

---

> ### Author Response · Authors · 2025-11-23
> **Official Comment by Authors (part 3/3)**
>
> ---
> ## Question Group 3 : Extensibility for Different Domain and Task
> ---
>
> ### [W3] UEA might not represent the time series in the wild. It should be tested on other challenging, recently released, real-world task frontier dataset.
>
> Since your comment did not specify particular datasets, we chose the two most recent datasets evaluated in Medformer—**ADFTD (2023)** from the medical domain and **FLAAP (2022)** from the HAR domain—which satisfy the "recent, real-world, challenging" criteria reasonably well.
>
> |  | Domain | Samples | Length | Classes | Variables | File Size |
> |:---:|:---:|---:|---:|---:|---:|---:|
> | ADFTD | EEG | 69752 | 256 | 3 | 19 | 2.52GB |
> | FLAAP | HAR | 13123 | 100 | 10 | 6 | 60.2MB |
>
> Below are the results (also in `Section 5.3 (Further analysis)`) where the baseline results are from Medformer. Without any domain-specific modification, **MambaSL matches or exceeds the baseline performance** and significantly improves over vanilla Mamba.
>
> |  | ADFTD (accuracy) | ADFTD (f1-score) | FLAAP (accuracy) | FLAAP (f1-score) |
> |:---:|---:|---:|---:|---:|
> | Crossformer | 50.45 ± 2.31 | 45.50 ± 1.70 | 75.84 ± 0.52 | 75.52 ± 0.66 |
> | Reformer | 50.78 ± 1.17 | 47.94 ± 1.59 | 71.65 ± 1.27 | 71.14 ± 1.45 |
> | Transformer | 50.47 ± 2.14 | 48.09 ± 1.59 | 74.96 ± 1.25 | 74.49 ± 1.39 |
> | TCN | N/A | N/A | 66.48 ± 1.66 | 65.29 ± 1.74 |
> | ModernTCN | N/A | N/A | 74.80 ± 0.96 | 74.35 ± 0.85 |
> | Mamba | N/A | N/A | 64.87 ± 2.78 | 64.14 ± 2.70 |
> | Medformer | 53.27 ± 1.54 | 50.65 ± 1.51 | 76.44 ± 0.64 | 76.25 ± 0.65 |
> | MambaSL | 51.68 ± 0.89 | 48.73 ± 3.05 | 77.68 ± 1.03 | 77.53 ± 1.09 |
>
> **> Reference**
>
> \[Medformer] *Medformer: A multi-granularity patching transformer for medical time series classification.* NeurIPS, 2024.
>
> \[ADFTD] *A dataset of scalp EEG recordings of alzheimer’s disease, frontotemporal dementia and healthy subjects from routine EEG.* Data, 2023.
>
> \[FLAAP] *FLAAP: An open human activity recognition (HAR) dataset for learning and finding the associated activity patterns.* Procedia Computer Science, 2022.
>
>
> &nbsp;
>
> ### [Q3] Could MambaSL be extended to forecasting?
>
> Our hypotheses fall into two groups:
> - *time series-specific* : H1, H2
> - *classification-specific* : H3, H4
>
> Therefore, **H1 and H2 can be applied to forecasting or other time series (TS) downstream tasks**.
>
> However, as noted in `Section 2 (Related Work)`, existing Mamba-based TS models apply **bidirectional Mamba along the channel axis**, not the time axis (e.g., [TimeMachine], [S-Mamba], [TSCMamba], [MambaAD], ...).  Under the formulation in `Section 3.2.3 (On Time Variance of B,C,Δ)`, this implies that:
> - $\boldsymbol{\Delta}$ regulates *channel* update rates, and
> - $\boldsymbol{B}$, $\boldsymbol{C}$ mix along the *time* axis.
> - If a model is in frequency-axis approaches, $\boldsymbol{\Delta}$ governs varying local speeds across frequencies.
>
> [TimeMachine]: https://github.com/Atik-Ahamed/TimeMachine/blob/main/TimeMachine_supervised/models/TimeMachine.py
> [S-Mamba]: https://github.com/wzhwzhwzh0921/S-D-Mamba/blob/main/model/S_Mamba.py
> [TSCMamba]: https://github.com/Atik-Ahamed/TSCMamba/blob/main/models/TSCMamba.py
> [MambaAD]: https://ars.els-cdn.com/content/image/1-s2.0-S0925231225020570-gr1_lrg.jpg
>
> Given that "channel invariance" or "frequency invariance" is less intuitive than time invariance, and that the meaning of temporal/frequency mixing is vague, **the theoretical implications of applying H1/H2 to existing Mamba TS models seem unclear**.
>
> Thus, MambaSL’s ideas can extend to other TS tasks, but applying them to existing Mamba TS architectures is not theoretically straightforward.
>
> &nbsp;
>
> **> Reference**
>
> \[TimeMachine] *TimeMachine: A time series is worth 4 mambas for long-term forecasting.* ECAI, 2024.
>
> \[S-Mamba] *Is Mamba effective for time series forecasting?* Neurocomputing, 2025.
>
> \[TSCMamba] *TSCMamba: Mamba meets multi-view learning for time series classification.* Information Fusion, 2025.
>
> \[MambaAD] *MambaAD: Multivariate time series anomaly detection in IoT via multi-view Mamba.* Neurocomputing, 2025.
>
> &nbsp;
>
> ---
> Once again, thank you again for your valuable feedback. The manuscript has been revised; and the source code, checkpoints, and experiment logs for the additional experiments will be updated in the near future.

---

> > ### Comment · Reviewer_GbtJ · 2025-11-23
> >
> > Q1 Group:
> > Thank you so much for the added changes! The error bars and statistical tests are very important for understanding the method's performance, and the improved visualization on Figure 4 makes results much clearer. I also appreciate you running the dataset comparison for the UMAP, these are very interesting results! This is an important contribution to the TS community to show sets of datasets on which non-DL methods excel, and it opens questions for future work as to examining why this is the case (although rightly not the focus of your work because it is out of scope).
> >
> > Q2 Group:
> > W2: This intuition makes total sense, and I think it makes sense at this time to not consider the variable-length time series tasks. There might be some methodological innovation in the future to advance to irregular lengths, but I agree it’s ok to keep this out of the scope of your present work.
> > Q1: Thank you for running this experiment on longer sequence lengths, this is super helpful for understanding model behavior. This is indeed an exciting result as it highlights how your model might be a SOTA choice in particular for long-horizon time series tasks.
> > Q4: Thank you for this clarification. While your connection to NLP here is a bit loose, I understand the reasoning behind how Mamba could work in irregularly-observed intervals.
> >
> > Q3 Group:
> > W3: Thank you for running this experiment! I appreciate you adopting two real-world datasets for this case, it’s more convincing of your model’s performance.
> > Q3: Thank you for this comment. While your argument is sound, I do believe this is a relative weakness of your proposed method, given that it will not extend to other types of time series tasks. However, I still believe you properly treat your proposed problem scope and won’t dock any points for this in my evaluation.
> >
> > Overall, I appreciate the authors taking time to carefully and thoroughly respond to my questions. Given that my score is already quite high, I cannot justify raising it, but I will raise my confidence as my concerns have been almost completely addressed.

---

> > > ### Author Response · Authors · 2025-12-04
> > > **Authors’ Sincere Gratitude to Reviewer GbtJ**
> > >
> > > Dear reviewer GbtJ,
> > >
> > > We sincerely thank you for taking the time to carefully read our response and for providing detailed follow-up feedback for each of the points. We deeply agree with the points you raised, and through your questions we were also able to gain several additional insights.
> > > It was a pleasure to engage in this constructive discussion with you.
> > >
> > > We were also very pleased to see that you significantly increased your confidence score, as this indicates that our explanations helped your understanding of the paper.

---

### Official Review · Reviewer_XLs9 · 2025-11-01

**Soundness:** 3
**Presentation:** 4
**Contribution:** 3
**Rating:** 6
**Confidence:** 3

**Summary:**

The paper proposes MambaSL, a single‑layer Mamba architecture tailored to time‑series classification (TSC). Four hypotheses drive minimal but targeted changes: H1 scales the input Conv1D kernel with sequence length; H2 modularizes time (in)variance of the SSM parameters; H3 disables the skip (D) connection; H4 introduces a multi‑head adaptive pooling readout. The overall block is in Figure 2 (p.5); ablation evidence is in Table 1 (p.8) and Table 2 (p.8–9). On the full 30‑dataset UEA archive, the method attains the top average accuracy and rank among 21 models.

**Strengths:**

+ Thorough ablations, including eight TI/TV combinations (Table 2) and multiple pooling alternatives (Table 1).
+ Excellent commitment to reproducibility across all 30 UEA datasets, with promises of public code, checkpoints, and full logs.
+ The re-evaluation of TSF-origin models, showing they were previously underestimated, is an important finding. The analysis of H2 (time variance) is the most interesting part, showing that for TSC, simpler LTI systems can be better than LTV.

**Weaknesses:**

+ The novelty of H1–H4 is moderate; each component is a small change rather than a new architectural principle.
+ The paper's own ablation study (Table 1) shows that H1 (scaling kernel size) is not clearly supported by the average accuracy metric.
+ The paper's title and focus on a "single-layer" model  is not well-justified. Why is one layer sufficient? The paper is missing a ablation study on the effect of model depth (i.e., stacking MambaSL layers).

**Questions:**

+ Could you provide a statistical test (e.g., a Wilcoxon signed-rank test) comparing the performance of MambaSL (with H1) directly against the "w/o H1" variant, or should H1 be re-framed in light of this evidence?
+ Did you try multi‑layer (2–3 layers) MambaSL to confirm single‑layer sufficiency? How does MambaSL perform on variable‑length sequences at test time (beyond the pooling stage)?

---

> ### Author Response · Authors · 2025-11-20
> **Official Comment by Authors (part 1/2)**
>
> Dear reviewer XLs9,
>
> Thank you for your thoughtful feedback. We appreciate your recognition of our re-evaluation of TSF-origin models, and we are glad that you found H2 (time variance) to be an especially compelling component of the paper.
>
> We respond to your three main comments below.
>
> &nbsp;
>
> ---
> ## Question Group 1: Novelty of Proposed Model and its Hypotheses (H1–H4)
> - [W1] The novelty of H1–H4 is moderate; each component is a small change rather than a new architectural principle.
> ---
>
> We fully agree. The proposed modifications are not conceptually complex; they are straightforward ideas that any researcher familiar with Mamba might consider.
>
> However, precisely because these ideas are simple, we believe they address aspects that many researchers have overlooked when applying Mamba to time series classification.
> In contrast to the rapid proliferation of Mamba-based models in NLP and forecasting over the past two years, **applications of Mamba to time series classification have been extremely limited** \[TSCMamba].
> We suspect this is because vanilla Mamba performs poorly enough on time series classification that researchers dismiss it at first glance—a point we highlighted in `Section 1 (Introduction)`: "the standalone capacity of Mamba for time series classification has seen little investigation", "Wang et al. (2024) ranked Mamba as the weakest time series classification backbone."
>
> Thus, showing that **small but targeted adjustments can significantly improve Mamba for time series classification** is, in our view, a meaningful contribution.
> Furthermore, because the modifications are simple and easy to adopt, we believe **practitioners can readily incorporate them**, potentially sparking more work on Mamba for time series classification.
>
> &nbsp;
>
> ---
> ## Question Group 2 : Lack of experimental evidence for H1
> - [W2] `Table 1 (Ablation study)` shows that H1 is not clearly supported by the average accuracy metric.
> - [Q1] Could you provide a statistical test (e.g., a Wilcoxon signed-rank test) comparing the performance of MambaSL (w/ H1) directly against the "w/o H1" variant, or should H1 be re-framed in light of this evidence?
> ---
>
> As noted in `Section 5.2 (Model Ablation)`, the UEA benchmark has structural limitations:  *dataset sizes differ drastically, and difficulty varies widely*.
> This means that a 1% improvement can be decisive on some datasets, while even a 5% improvement may be insignificant on others.
>
> AtrialFibrillation and StandWalkJump exemplify this—each has only 15 test samples, so one additional correct prediction changes the accuracy by 6.67% (≈0.22%p in overall average). Excluding these two, the mean accuracy is 80.96% for MambaSL (with H1) vs. 80.47% without H1. Thus, *average accuracy alone is insufficient to judge model superiority*, motivating the use of **ranks, win–loss counts, and Wilcoxon tests**.
>
> Replacing MambaSL with its **w/o H1** variant in the benchmark gives the following results. Notably, the **average rank drops from 4.97 to 5.43**, and unlike the original MambaSL—which only failed the Wilcoxon test against HC2—the w/o H1 model **shows no statistically significant difference against HC2, Hydra, TSLANet, InterpGN, and TSCMamba**.
>
> |  | MambaSL (w/o H1) | DTW_D | ROCKET | HC2 | Hydra | MR+Hydra | Dlinear | LightTS | MTS-Mixer | TimesNet | ModernTCN | TSLANet | TimeMixer++ | FEDformer | ETSformer | Crossformer | PatchTST | GPT4TS | iTransformer | InterpGN | TSCMamba |
> |---|---:|---:|---:|---:|---:|---:|---:|---:|---:|---:|---:|---:|---:|---:|---:|---:|---:|---:|---:|---:|---:|
> | avg. acc (10) | 77.413 | 70.354 | 74.368 | 78.156 | 78.349 | 77.600 | 70.999 | 72.760 | 69.578 | 75.168 | 73.429 | 77.443 | 74.948 | 71.545 | 74.267 | 75.316 | 72.895 | 74.301 | 74.120 | 75.482 | 75.933 |
> | avg. acc | 80.215 | 68.020 | 73.306 | 76.866 | 77.124 | 76.320 | 68.247 | 75.308 | 69.001 | 76.921 | 76.938 | 76.263 | 77.279 | 74.128 | 75.716 | 77.427 | 74.729 | 76.065 | 74.797 | 77.700 | 78.405 |
> | avg. rank | **5.43** | 16.00 | 11.20 | 8.90 | 8.83 | 9.40 | 16.83 | 11.83 | 13.93 | 8.60 | 9.10 | 7.83 | 7.03 | 12.30 | 10.53 | 8.43 | 12.57 | 9.57 | 11.87 | 7.87 | 7.10 |
> | # of win/draw |  | 28 | 24 | 20 | 20 | 21 | 28 | 25 | 28 | 24 | 25 | 18 | 23 | 24 | 26 | 21 | 22 | 26 | 25 | 20 | 19 |
> | # of lose/draw |  | 3 | 9 | 14 | 13 | 11 | 2 | 7 | 3 | 10 | 9 | 14 | 10 | 8 | 7 | 12 | 8 | 5 | 6 | 14 | 15 |
> | Wilcoxon test |  | 0.000 | 0.001 | **0.065** | **0.068** | 0.021 | 0.000 | 0.000 | 0.000 | 0.004 | 0.001 | **0.055** | 0.006 | 0.000 | 0.000 | 0.011 | 0.001 | 0.000 | 0.000 | **0.053** | **0.104** |
>
> We also added Wilcoxon test values to `Table 1 (Ablation study)`. As expected, the effect of H1 is not statistically significant.
>
> Note that `Equation 10 (setting kernel size)` and $λ=0.02$ enlarge the kernel only when $L>150$, which holds for only *16 out of 30 UEA datasets*—this further limits H1’s measurable impact.

---

> ### Author Response · Authors · 2025-11-23
> **Official Comment by Authors (part 2/2)**
>
> We also recognize that H1 does not necessarily have a positive effect on every dataset, which is why we explicitly state in `Section 6 (Conclusion)` that "prior knowledge of input embedding may constrain kernel size, making receptive-field scaling with sequence length suboptimal."
> Our intended message is that **when applying Mamba to time series classification, one should set the receptive field for input (token) embeddings with the role of Mamba’s gating mechanism in mind**, and **in the absence of any domain knowledge, scaling the receptive field with sequence length or sampling rate is a more reasonable default than simply fixing $k=3$**.
>
>
> &nbsp;
>
> ---
> ## Question Group 3 : Lack of ablation on the number of layers
> - [W3] The paper's title and focus on a "single-layer" model is not well-justified due to a missing ablation study on the effect of model depth.
> - [Q2] Did you try multi‑layer (2–3 layers) MambaSL to confirm single‑layer sufficiency? How does MambaSL perform on variable‑length sequences at test time (beyond the pooling stage)?
> ---
>
> Thank you for pointing this out. Although we knew empirically that adding layers rarely improves performance in these baselines, we mistakenly assumed this was obvious and failed to include the analysis.
>
> We therefore experimented with 2–3 Mamba blocks and compared them to the single-layer version. Results (also shown in `Section 5.3 (Further analysis)`) are shown below. As you can see, both average accuracy and rank are better for the single-layer model. However, the differences are small, and we wish to emphasize that **even with a single layer, a well-designed architecture for time series classification (H1–H4) can ensure sufficient classification performance**.
>
> | model depth | avg. acc (10) | avg. acc | avg. rank | # of win/draw | # of lose/draw |
> |:---:|---:|---:|---:|---:|---:|
> | 1 (ours) | 78.827 | **79.819** | **1.33** |  |  |
> | 2 | 78.853 | 79.361 | 1.77 | 26 | 15 |
> | 3 | **78.918** | 79.279 | 2.10 | 24 | 12 |
>
> ###### * ~~InsectWingbeat and PhonemeSpectra datasets have not yet been tested across all hyperparameter combinations, so there may be some discrepancies. However, the differences are expected to be minimal.~~ test finished
>
> &nbsp;
>
> Regarding [Q2]:
> It was unclear whether "beyond the pooling stage" refers to handling variable lengths *before* pooling or *after* pooling, so we address both.
>
> Using $\mathtt{TSLib}$, MambaSL receives **zero-padded sequences along with a mask** indicating valid positions. The Conv1D embedding layer uses repetitive padding; after the Mamba block, we **apply the mask again** so padded tokens do not influence outputs.
> Implementation details appear in `data_provider/uea.py` (`collate_fn`) and `model/MambaSingleLayer.py` (`Model.forward`).
>
>
> &nbsp;
>
> ---
> Once again, thank you again for your valuable feedback. The manuscript has been revised; and the source code, checkpoints, and experiment logs for the additional experiments will be updated in the near future.

---

> ### Author Response · Authors · 2025-12-04
> **Authors’ Sincere Gratitude to Reviewer XLs9**
>
> Dear reviewer XLs9,
>
> Although we did not have the opportunity to hear your further thoughts on our previous response, we hope that our answers were helpful in addressing your concerns.
>
> Thank you for pointing out several aspects that we had previously overlooked, which allowed us to further improve the clarity of the paper.

---

### Official Review · Reviewer_wsQb · 2025-11-09

**Soundness:** 3
**Presentation:** 2
**Contribution:** 2
**Rating:** 4
**Confidence:** 4

**Summary:**

This paper introduces MambaSL, a framework that minimally modifies the selective state space models and projection layers of a single-layer Mamba architecture for time series classification. Experimental evaluations conducted on 30 datasets from the UEA benchmark demonstrate that MambaSL consistently outperforms 20 competitive baseline methods.

**Strengths:**

1. This paper applies the Mamba architecture to time series classification tasks and achieves superior performance compared to Transformer, MLP, and CNN models on multivariate time series classification benchmarks. This demonstrates the strong potential of Mamba for time series classification modeling.
2. The authors provide open-source code for the proposed model, along with implementations of the baseline methods, ensuring the reproducibility and transparency of the experimental results.

**Weaknesses:**

1. The paper’s title, introduction, and model design focus primarily on time series classification without providing an in-depth discussion of how variable relationships are modeled in multivariate time series. In this case, the experiments are limited to the 30 datasets from the UEA multivariate time series classification archive, while excluding the 128 univariate datasets. This omission weakens the paper’s motivation and makes it difficult to assess whether the proposed method achieves state-of-the-art performance under univariate settings.

2. Although the application of Mamba to time series classification is commendable, the paper does not clearly explain how the proposed MambaSL framework captures intrinsic temporal properties—such as temporal dependencies, inter-variable relationships, and short- or long-term sequence characteristics. The introduction and contribution sections also fail to clarify how MambaSL learns discriminative feature patterns beneficial for classification.

3. The model section contains extensive background on basic Mamba concepts, making it difficult for readers unfamiliar with Mamba to follow. At the same time, readers experienced in time series classification may still find it unclear how the model effectively learns task-relevant features for classification.

4. Despite the considerable experimental effort and the inclusion of reproducible scripts, all deep learning results in the paper suffer from a **test data leakage issue**. As shown in the provided code (`MambaSL/exp/exp_classification.py`, lines 22–25), the test set is incorrectly used as the validation set:

   ```python
   self.train_data, self.train_loader = self._get_data(flag='TRAIN')
   if self.args.is_training:
       self.vali_data, self.vali_loader = self._get_data(flag='TEST')
   self.test_data, self.test_loader = self._get_data(flag='TEST')

This indicates that the test set is used for model validation and hyperparameter selection, resulting in a significant evaluation bias.

5. In Appendix A, the authors mention that the experimental framework is based on the **Time-Series Library (TSLib)** (Wu et al., 2023). While TSLib is widely recognized for fair benchmarking in forecasting tasks, prior research ([1], Section 4.4 “Leaky Baselines”) has shown that TSLib introduces **test data leakage** in classification experiments. Furthermore, several non-deep learning baselines (e.g., Rocket, HC2, Hydra) do not rely on validation sets during training and for model selection. Consequently, using the maximum test performance as the final evaluation metric for MambaSL constitutes an **unfair comparison** with these baselines.

**Reference**

[1] *TOTEM: Tokenized Time Series Embeddings for General Time Series Analysis*, TMLR, 2024.

**Questions:**

1. In univariate time series classification, the original **InceptionTime** paper evaluates models based on the checkpoint with the lowest training loss, and the same evaluation strategy is adopted in **[2]**. Under a comparable setting, how does **MambaSL** perform relative to **HC2** and **MultiRocket+Hydra** on the 30 UEA multivariate time series datasets?

2. On the 128 UCR univariate time series datasets, following the **InceptionTime evaluation protocol**—where the model with the lowest training loss is used for testing—how does **MambaSL** compare in classification performance with **HC2** and **MultiRocket+Hydra**?

3. Beyond the evaluation setups in **InceptionTime** and **[2]**, some studies (e.g., **TSLANet**) adopt a different experimental protocol that uses **20% of the training set as a validation subset** for model selection on both UCR and UEA datasets. Under this widely used setting, how does **MambaSL** perform compared with **HC2**, **MultiRocket+Hydra**, and other competitive baselines?

4. Compared to 20 baseline methods, what are the advantages and disadvantages of the proposed **MambaSL** method in terms of runtime?

**Reference**

[2] *Inherently Interpretable Time Series Classification via Multiple Instance Learning*, ICLR, 2024.

**Details Of Ethics Concerns:**

None.

---

> ### Author Response · Authors · 2025-11-20
> **Official Comment by Authors (part 1/4)**
>
> Dear reviewer wsQb,
>
> Thank you for your careful reading and positive evaluation. We appreciate your comments recognizing the potential of Mamba for time series classification and the importance of open-source reproducibility.
>
> We group your questions into three categories since you raised quite a few points of inquiry.
>
> &nbsp;
>
> ---
> ## Question Group 1
> - [W1] Why not UCR though it is time series classification research?
> - [W4,W5] Data leakage issue in time series-library (also proposed model)
> - [Q1] UEA performance under InceptionTime setting?
> - [Q2] UCR performance under InceptionTime setting?
> - [Q3] UCR/UEA performance under TSLANet setting?
> ---
>
> ### 1.1. The scope of our study
>
> We suspect you are deeply familiar with the long history of time series classification research, but we begin by briefly outlining the current landscape.
>
> Today's time series classification research, as you may know, **bifurcates into two main streams that have developed somewhat independently**.
>
> **(1) The aeon / classical ML/DL track**, typically built around the UCR/UEA bake-off and Hive-Cote (HC).
> Evaluation is typically based on ***the average over 30 stratified resamples***. Here, stratified resample means splitting the entire dataset (train + test) into new train/test sets while maintaining the original train/test ratio. There is no data leakage, but the **train/test split is replaced**, which is unusual for deep-learning researchers.
>
> **(2) The time series-library ($\mathtt{TSLib}$) / modern DL track**, which this work belongs to.
> Here, the original train/test split is preserved, but—as you noted—$\mathtt{TSLib}$ **suffers from data leakage**. We cannot claim to know the authors’ intention, but because many UEA datasets are extremely small, using the test set as validation (with patience) may have been a pragmatic choice to avoid unstable training. We have also verified that increasing patience from 10 to 20 improves performance.
>
> Beyond methodology, the **two tracks use different datasets**.
> Classical TS papers emphasize UCR; UEA results for modern classical methods such as Hydra are surprisingly scarce—BORF is almost the only source providing UEA results for univariate time series classification methods.
> Conversely, most recent UEA-based DL papers use $\mathtt{TSLib}$, and thus inherit its leakage issue. TOTEM and ModernTCN Revisited discuss this, but both avoid re-experimenting and instead simply exclude the classification setting to avoid deviating from prior work.
>
> Our study clearly falls into **the second category**, focusing on multivariate time series classification and using $\mathtt{TSLib}$ to minimize gaps against prior work—even though this means inheriting its leakage issue.
>
>
> &nbsp;
>
> **> Reference**
>
> \[UEA bake off] *The great multivariate time series classification bake off: a review and experimental evaluation of recent algorithmic advances.* KDD, 2021.
>
> \[BORF] *Fast, interpretable and deterministic time series classification with a bag-of-receptive-fields.* IEEE Access, 2024.
>
> \[TOTEM] *TOTEM: Tokenized Time Series Embeddings for General Time Series Analysis*, TMLR, 2024.
>
> \[ModernTCN Revisited] *ModernTCN Revisited: A Critical Look at the Experimental Setup in General Time Series Analysis*. TMLR, 2025.
>
> &nbsp;
>
> ### 1.2. Answering to questions related to univariate time series study [W1,Q2,Q3]
>
> Our entire study assumes **multivariate** settings, which we made *explicit in several places*:
> - `Section 1 (Introduction)`: "datasets with long sequences or high input dimensionality," contribution mentions "input dimension (2–1,345)"
> - `Section 3.1 (Problem definition)`: "multivariate time series of length $L$"
>
> Moreover, since conventional deep learning research typically assumes a multivariate setting, we felt it was not necessary to explicitly repeat the term "multivariate" throughout the paper. In addition, in time series works such as InceptionTime, MILLET, and others cited in your review, it is uncommon to see "univariate" or "multivariate" explicitly emphasized beyond the problem definition or dataset description, so we believed our level of clarification was sufficient.
>
> To make it clearer that this work is a study on multivariate time series classification, we have additionally mentioned "multivariate UEA datasets" in the **abstract** and **contribution**, and we also emphasize "multivariate TSC backbone" in the **Conclusion**. We hope this addresses your concerns in [W1, Q2, Q3].
>
> ~~In fact, we are also curious about the performance on the UCR datasets; however, given our limited time and resources, it was difficult to prioritize 85 UCR experiments over the UEA experiments within within the review period, so we decided to focus solely on UEA.~~

---

> ### Author Response · Authors · 2025-11-23
> **Official Comment by Authors (part 2/4)**
>
> ### 1.3. Answering to questions related to data leakage and fair comparison with non-DL [W4,W5,Q1]
>
> Before addressing the questions on data leakage and fair comparison, we would first like to note that the **20% dataset split was not adopted in our experiments on TSLANet due to a reproducibility issue**. Although the TSLANet paper states that 20% of the data is used as a validation set, we could not find any explicit preprocessing code nor any preprocessed dataset in the publicly available source code.
> We thus implemented the 20% split ourselves, but because many datasets have very few samples per class, random splits frequently resulted in runs that simply did not train.
>
> To address this, we implemented a *stratified 8:2 split per class*, ensuring that the overall validation proportion is 20%. However, even with the stratified split, the average accuracy on UEA was only 70.994% (70.5% on the 26 datasets used in the paper), which still does not reproduce the performance reported in the original TSLANet paper (72.73%). Finally, we were only able to reproduce the reported performance when we used the same setting as $\mathtt{TSLib}$. The corresponding preprocessing code can be found in `_run_TSLANet/dataloader.py` at `lines 133:161`.
>
> In summary, for TSLANet over 26 datasets, we observed the following results:
> ```
> (8:2 split) not trainable < (stratified 8:2 split) 70.5% > (paper result) 72.73% < (our experiment) 74.89%
> ```
> Furthermore, a rank test shows that our experimental results are actually closer to the paper’s reported numbers than those obtained with the stratified 8:2 split. Therefore, we believe that applying the **random 8:2 split** suggested in TSLANet directly to the UEA datasets **is not a particularly appropriate protocol**.
>
> &nbsp;
>
> Moreover, although we were **aware of the data leakage issue in advance**, the primary goal of this work—especially in light of the second issue emphasized in the introduction (i.e., that re-implementations of prior models often report lower performance or conduct fewer hyperparameter sweeps than the original papers)—was to **avoid deviating too far from the performance reported in prior work**. We would appreciate it if you could understand our choice in that context. As mentioned in Answer 1.1 above, $\mathtt{TSLib}$ uses **relatively small default values for patience and total epochs (10 and 100, respectively)**, which we believe already helps to mitigate the extent of leakage to some degree. Our choice of a relatively large learning rate of 1e-3 and fixing the weight decay to 0 was made for a similar reason. Compared with the hyperparameters used to train InceptionTime in HC2—learning rate 1e-3, total epochs of 1500, and halving the learning rate every 50 epochs without performance improvement—our setting can be seen as a more conservative (i.e., *less overfitting-prone*) choice.
>
> That said, we fully acknowledge that this does not completely resolve the concerns raised in [W4, W5, Q1]. In line with your request, we therefore **additionally trained MambaSL under the InceptionTime protocol** and report the results below. Due to time constraints, we could not run 1500 epochs as in the original InceptionTime setup, and instead reused our original hyperparameters with 100 epochs and patience 10.
>
> For the baselines, we constructed a comparison set using:
> (1) the non-DL models we re-evaluated,
> (2) the [fold0 results](https://timeseriesclassification.com/results/PublishedResults/HIVE-COTEV2/) reported in HC2, and
> (3) the results reported in the BORF paper.
> From each of these sources, we selected the three best-performing baseline models.
> Note that both HC2 and BORF omit some datasets from their experiments.
>
> As expected, MambaSL’s performance decreases compared to the $\mathtt{TSLib}$ setting. However, when compared with the non-DL models we ran and the baselines reported in other works, MambaSL remains comparable or better. For example, although Hydra achieves the highest average accuracy (77.124%) among all baselines, MambaSL still has a favorable head-to-head record against Hydra (19 wins and 15 losses). The corresponding analysis is provided in `Section 5.3 (Further analysis)` and `Appendix C.6 (UEA under InceptionTime setting)`.
>
> | reference | model | # of datasets | avg. acc | # of win/draw | # of lose/draw |
> |:---:|:---:|---:|---:|---:|---:|
> | HC2 | CIF | 26 | 72.604  | 18  | 10  |
> | HC2 | DrCIF | 26 | 73.246  | 19  | 9  |
> | HC2 | HC2 | 26 | 74.785  | 17  | 13  |
> | BORF | MiniRocket | 29 | 73.955  | 19  | 12  |
> | BORF | DrCIF | 30 | 75.013  | 20  | 11  |
> | BORF | MR+Hydra | 30 | 75.572  | 19  | 13  |
> | MambaSL | MR+Hydra | 30 | 76.320  | 18  | 14  |
> | MambaSL | HC2 | 30 | 76.866  | 18  | 16  |
> | MambaSL | Hydra | 30 | 77.124  | 19  | 15  |
> | MambaSL | **MambaSL** | 30 | 76.759  |  |  |
>
> ###### * Since experiments on some datasets are not yet complete, future results may vary slightly. However, the differences are expected to be minimal.
>
> &nbsp;

---

> > ### Comment · Reviewer_wsQb · 2025-11-24
> > **Reply to (part 2/4)**
> >
> > Thank you for the authors’ response. First, I want to clarify that my primary concern is not about the reproducibility of TSLANet or the repeated emphasis on reproducing the original deep learning baselines results. Variations in experimental setups and computational environments naturally lead to discrepancies, and the authors have ensured that all baseline methods are evaluated under a unified environment and have provided re-run results for all comparisons. This is a clear strength of the work. My suggestion of using 20% of the training set as a validation set was simply intended to offer an alternative that avoids using the test set for validation.
> >
> > However, my main concern lies in the fact that, in Figure 4 of the main text, all deep learning models use the test set as the validation set for model selection, while non-deep-learning methods such as MR+Hydra and HC2 do not involve the test set at all. This discrepancy makes the reported test-set classification performance of the deep learning methods unreliable. For example, when following the InceptionTime experimental protocol, the authors report that MambaSL achieves an average accuracy of 76.759, which is noticeably lower than the 79.80 shown in Figure 4.
> >
> > The new experimental results provided by the authors indicate that, once test-set leakage is removed, MambaSL struggles to surpass HC2 and MR+Hydra. This demonstrates that test-set leakage indeed produces unfair comparisons between deep learning and non-deep-learning approaches. In addition, the reported performances of HC2, Hydra, and MR+Hydra appear lower than those shown in Figure 4 (e.g., Hydra’s average accuracy of 78.35 vs 77.124 here). Could the authors clarify the source of this discrepancy?

---

> > > ### Author Response · Authors · 2025-12-04
> > > **Reply to the Follow-up Questions (part 3/4)**
> > >
> > > &nbsp;
> > >
> > > ### > Further discussion on experimental results under InceptionTime setting
> > >
> > > Although the average accuracy of MambaSL under the InceptionTime setting appeared somewhat lower than that of certain non-DL models,
> > > we observed that **MambaSL tends to be slightly superior in terms of average rank, win–loss counts, and the Wilcoxon signed-rank test**.
> > >
> > > For instance, when comparing our implementation of HC2 with MambaSL, the p-value of the rank test was measured as 0.307, which equivalently means that the probability that HC2 is better than MambaSL is 0.693. In other words, while we cannot claim a statistically significant difference, these numbers suggest that MambaSL is slightly better in terms of ranking and wins/losses.
> > >
> > > When we analyzed the cause of this discrepancy at the dataset level, we found that MambaSL’s accuracy on the LSST dataset dropped from 45.580% to 25.182%, a decrease of about 20 percentage points, and this was the main factor leading to a lower overall average accuracy, even though MambaSL retained an advantage in rank.
> > >
> > > &nbsp;
> > >
> > > Furthermore, the reason our non-DL baseline results are generally higher than those reported in previous papers is that we used the **maximum performance over multiple random seeds** for the non-DL models as well. As mentioned in our first comment, we observed occasional training failures depending on the seed (especially for InsectWingbeats). Moreover, since many non-DL models are based on ROCKET (which uses 10,000 random kernels) or combine multiple methods (HC2, MR+Hydra), we judged that they could be more sensitive to the seed choice than to hyperparameter tuning.
> > >
> > > Therefore, for the non-DL models as well, we ran experiments over multiple seeds and then aggregated the results by taking the maximum (of course, DTW is a 1-NN classifier, so we only ran it once).
> > > We hope this makes it clear that we made an effort to ensure a fair comparison not only for DL models but also for non-DL models.
> > >
> > > If we instead use only a single seed for each non-DL model, our results are as follows.
> > > They are reasonably close to previously reported numbers, and when we compare these results with MambaSL under the InceptionTime setting, **our model outperforms the existing non-DL baselines in both average accuracy and ranking**.
> > >
> > > | model | # of datasets | avg. acc | # of win/draw | # of lose/draw |
> > > |:---:|---:|---:|---:|---:|
> > > | **MambaSL** | 30 | 76.759  |  |  |
> > > | MR+Hydra | 30 | 75.397  | 19  | 13  |
> > > | HC2 | 30 | 74.010  | 22  | 11  |
> > > | Hydra | 30 | 73.795  | 21  | 10  |
> > >
> > > In the camera-ready version, we plan to additionally report mean ± std to enable clearer comparisons.

---

> ### Author Response · Authors · 2025-11-23
> **Official Comment by Authors (part 3/4)**
>
> ---
> ## Question Group 2
> - [W2] How MambaSL captures intrinsic temporal properties/patterns? Especially for classification?
> - [W3] Extensive Mamba background makes readers more confuse about the W2. At the same time, readers may still find it unclear how MambaSL effectively learns task-relevant features for classification.
> ---
>
> All four hypotheses (H1–H4) we propose in the paper correspond to minimal structural modifications designed to **help Mamba better learn from time series classification datasets**. Unlike classical time series classification models, which often explicitly focus on capturing temporal "patterns," **MambaSL is primarily aimed at helping Mamba’s SSM retain meaningful state information over long horizons**. In that sense, we believe the question "How does MambaSL capture temporal properties/patterns?" is essentially equivalent to "How does Mamba capture temporal properties/patterns?".
>
> Thus, the overarching motivation of this work is that **naively adopting the original Mamba architecture (or typical time series classification pipeline) can easily lead to overfitting in time series classification**. As mentioned at the end of `Section 6 (Conclusion)`, H1 and H2 are designed with the "time series" domain in mind, whereas H3 and H4 are tailored to the "classification" task:
> - **H1**: If the receptive field used to build input embeddings is too small, Mamba’s *gating mechanism may only capture very local patterns*. We therefore scale the receptive field with sequence length or sampling rate so that it can observe a longer context.
> - **H2**: When data are collected from a time-invariant system, training with the original selective SSM may actually *prevent the Mamba from capturing time-invariant properties*. We thus make time (in)variance a tunable hyperparameter.
> - **H3**: For classification, each label summarizes the entire sequence rather than a single position. *Overly strong skip connections and large model capacity can interfere with capturing global properties*, so we remove the skip connection and reduce capacity.
> - **H4**: In classification, the label may correspond to *either a global summary of the sequence or a specific temporal event*. Instead of pure global average pooling, we introduce an adaptive pooling mechanism that allows the model to choose pooling behavior (average or max) that better fits each domain. Since Mamba is a recurrent model, cherry-picking the most relevant time step in the classifier still allows the model to consider the entire sequence history.
>
>
> &nbsp;
>
> We first apologize for any confusion caused by the lengthy explanation of Mamba. We will try to clarify the motivation behind `Section 3.2 (Time Variant SSM in Mamba)` in this response.
>
> `Section 3.2.1 (LTI SSM)` introduces SSMs for readers who may not be familiar with Mamba or SSMs, so that they can still follow the paper. It also emphasizes that when applying an SSM to multivariate sequences, all parameters except $\boldsymbol{A}$ are broadcasted, making the dynamics **channel-independent**.
>
> The reason we emphasize this is that **Mamba’s selective SSM performs channel mixing** unlike earlier SSM-based models such as S4 (as described in `Section 3.2.2 (Selective SSM)`). Because this behavior is not explicitly described in the Mamba literature, Mamba is sometimes (incorrectly) assumed to be channel-independent like traditional SSMs. For instance, in [DeltaNet](https://openreview.net/forum?id=UvTo3tVBk2), the authors state that "this allows DeltaNet to perform simultaneous token- and channel-mixing, while Mamba can only perform token-mixing," which reflects such a misunderstanding.
>
> To our knowledge, very few papers have **explicitly formulated the selective SSM for multivariate inputs**. Thus, we **carefully specify the tensor dimensions of $\boldsymbol{B}$, $\boldsymbol{C}$, and $\boldsymbol{\Delta}$** in `Section 3.2.2 (Selective SSM)` to clearly articulate Mamba’s channel-mixing behavior. Building on this formulation, `Section 3.2.3 (On Time Variance of B,C,Δ)` then explains the **role of each parameter** and how Mamba captures temporal characteristics of time series. In that sense, `Section 3.2.2` and `Section 3.2.3` go beyond basic background and directly motivate our hypothesis H2.
>
> Thus, we view **the entirety of `Section 3.2` as a conceptual and mathematical setup specifically for H2**, rather than general background that might be safely omitted.
> We admit that this full explanation may have been too dense for readers unfamiliar with Mamba or SSMs, and we will strive to clarify the presentation in the final version.
>
> &nbsp;

---

> > ### Comment · Reviewer_wsQb · 2025-11-24
> > **Reply to (part 3/4)**
> >
> > The authors’ response has resolved this part of my concerns; however, several important clarifications are still needed. First, the authors should more explicitly explain how MambaSL models inter-variable relationships in multivariate time series. Second, the decision to omit evaluation on the univariate UCR archive and to focus exclusively on multivariate UEA datasets requires clearer justification.
> >
> > In addition, for deep learning models, it is not necessary for all methods to train for 1500 epochs as done in InceptionTime. Reasonable training schedules such as 300 or 500 epochs often achieve competitive classification performance while substantially reducing computational cost.

---

> ### Author Response · Authors · 2025-11-23
> **Official Comment by Authors (part 4/4)**
>
> ---
> ## Question Group 3
> - [Q4] Any memory/runtime merits in MambaSL?
> ---
>
> As is well known, Mamba generally has lower computational cost and smaller model size than Transformer-style architectures, and time series forecasting works based on Mamba mostly emphasize **computational efficiency rather than purely maximizing accuracy** [TimeMachine, S-Mamba, ...]. Since almost all Mamba-based time series papers stress this point, we considered it a lower priority compared to our other contributions and chose to omit a detailed efficiency discussion from the main text.
>
> On a more personal note, **the primary motivation behind this work was precisely our GPU memory constraints**. With only four GTX 1080 Ti GPUs available, it was very difficult to develop a TSC model comparable to the latest Transformer or CNN-based methods. This limitation motivated us to **explore Mamba** for TSC, even though there were essentially no prior time series classification works using Mamba at the time we began (this was before TSCMamba was published).
>
> &nbsp;
>
> We selected three representative UEA datasets—EigenWorms (longest sequence), InsectWingbeats (largest number of samples), and DuckDuckGeese (highest dimensionality)—and compared models in terms of training time, test time, number of parameters, and accuracy. For non-DL methods, parameter counts are often not clearly defined and checkpoints are not typically stored, so for non-DL methods we exclude model sizes.
>
> As shown in the table below, MambaSL demonstrates favorable training and test times compared to many other DL models, while maintaining competitive accuracy.
> Although not explicitly shown in the table, we also observed that training with InsectWingbeats using non-DL methods can terminate prematurely depending on the random seed. This suggests that non-DL learning methods may be less stable on very large datasets.
>
>
> |  | EigenWorms |  |  |  | InsectWingbeat |  |  |  | DuckDuckGeese |  |  |  |
> |---|---:|---:|---:|---:|---:|---:|---:|---:|---:|---:|---:|---:|
> |  | train time (s) | test&nbsp;time&nbsp;(s) | param&nbsp;# | acc&nbsp;(%) | train time (s) | test&nbsp;time&nbsp;(s) | param&nbsp;# | acc&nbsp;(%) | train time (s) | test&nbsp;time&nbsp;(s) | param&nbsp;# | acc&nbsp;(%) |
> | DTW_D | 0.0  | 33368.3  |  | 61.832  | 0.0  | 22790.5  |  | 48.692  | 0.0  | 311.9  |  | 58.000  |
> | ROCKET | 84.3  | 87.5  |  | 91.603  | 3519.3  | 25.6  |  | 38.524  | 0.9  | 0.8  |  | 52.000  |
> | HC2 | 2637.2  | 2049.4  |  | 97.710  | 65653.2  | 1856.6  |  | 66.796  | 57.1  | 24.3  |  | 70.000  |
> | Hydra | 111.4  | 110.4  |  | 98.473  | 48.2  | 28.5  |  | 67.052  | 0.5  | 0.5  |  | 68.000  |
> | MR+Hydra | 4.4  | 4.1  |  | 98.473  | 3761.1  | 119.7  |  | 65.744  | 4.6  | 4.5  |  | 60.000  |
> | Dlinear | 12.7  |  | 647,424K | 42.748  | 1032.5  |  | 45K | 18.996  | 47.0  |  | 1,962K | 66.000  |
> | LightTS | 45.7  |  | 324,209K | 54.198  | 1461.3  |  | 92K | 71.676  | 18.9  |  | 3,705K | 56.000  |
> | MTS-Mixer | 1216.6  |  | 20,142K | 66.412  | 328.3  |  | 151K | 72.424  | 17.6  |  | 2,147K | 68.000  |
> | TimesNet | 929.8  |  | 8,510K | 63.359  | 1845.4  |  | 5,560K | 64.024  | 62.6  |  | 2,517K | 68.000  |
> | ModernTCN | 81.1  |  | 4,060K | 64.122  | 34170.5  |  | 261,802K | 75.588  | 40.5  |  | 1,631,460K | 66.000  |
> | TSLANet |  | 5.9  | 909K | 81.679  |  | 30.3  | 128K | 10.012  |  | 3.5  | 636K | 74.000  |
> | TimeMixer++ | 207.6  |  | 30,701K | 61.069  | 2641.1  |  | 4,886K | 64.192  | 261.6  |  | 6,707K | 74.000  |
> | FEDformer | 366.7  |  | 26,051K | 63.359  | 4903.7  |  | 74K | 65.244  | 59.0  |  | 610K | 42.000  |
> | ETSformer | 837.7  |  | 47,382K | 70.992  | 2745.5  |  | 91K | 64.868  | 61.7  |  | 1,090K | 42.000  |
> | Crossformer | 242.0  |  | 3,913K | 58.015  | 1568.6  |  | 499K | 73.712  | 116.9  |  | 17,234K | 72.000  |
> | PatchTST | 26.8  |  | 55K | 57.252  | 2606.2  |  | 2,883K | 60.936  | 31.0  |  | 2,179K | 40.000  |
> | GPT4TS | 47.2  |  | 53,590K | 66.412  | 4802.2  |  | 44,498K | 61.396  | 37.1  |  | 75,858K | 66.000  |
> | iTransformer | 17.2  |  | 3,626K | 59.542  | 741.7  |  | 1,249K | 73.436  | 34.8  |  | 548K | 60.000  |
> | InterpGN | 310.9  |  | 3,430K | 82.443  | 585.1  |  | 577K | 67.180  | 119.5  |  | 9,133K | 76.000  |
> | TSCMamba | 57.1  |  | 2,173K | 88.550  | 274647.1  |  | 10,839K  | 68.244  | 85.1  |  | 127,384K | 50.000  |
> | MambaSL | 70.4  |  | 73K | 83.969  | 686.3  |  | 3,877K  | 66.304  | 34.2  |  | 11.503K | 70.000  |
>
> ###### * For TSLANet, training time was not measured previously, so only the test time is reported. Since it consists of 100 pre-training epochs and 50 fine tuning epochs, the training time is not very short.
>
>
> &nbsp;
>
> ---
> Once again, thank you again for your valuable feedback. The manuscript has been revised; and the source code, checkpoints, and experiment logs for the additional experiments will be updated in the near future.

---

> > ### Comment · Reviewer_wsQb · 2025-11-24
> > **Reply to (part 4/4)**
> >
> > Thank you for the authors’ response. The additional runtime analysis makes it clear that MambaSL indeed offers strong computational efficiency, which addresses my concerns on this aspect. However, I noticed that MambaSL is reported to be significantly faster than ROCKET, while on the 85 UCR datasets—using the default training and test splits—ROCKET typically requires only about 2–3 hours to produce all classification results.
> >
> > Given this relatively low computational cost for ROCKET, I would like to ask: how does MambaSL’s classification performance compare with ROCKET and MR+Hydra on the 85 UCR benchmark?

---

> > > ### Author Response · Authors · 2025-12-04
> > > **Reply to the Follow-up Questions (part 1/4)**
> > >
> > > Dear reviewer wsQb,
> > >
> > > Before moving on to the further discussion, we would first like to clarify two points where our previous response appears to have caused some misunderstanding.
> > >
> > > &nbsp;
> > >
> > > ---
> > > ### > About cross validation (30 stratified resamples)
> > >
> > > The cross validation protocol we mentioned, "on the average over 30 stratified resamples," was **not** the protocol we intended to use ourselves, but was introduced to **describe the evaluation protocol used in the UCR/UEA bakeoff and Hive-Cote2.0**.
> > > - UEA bakeoff paper: *"we average performance measures over the thirty resamples"*
> > > - Hive-Cote2.0 paper : *"For each dataset we present performance as an average over 30 resamples. Both archives provide a default split into train and test sets which we use for the first resample."*
> > >
> > > We are also aware that this protocol is somewhat unusual in deep learning research, and this is why, in our first comment, we wrote that "There is no data leakage, but the train/test split is replaced, which is unusual for deep-learning researchers."
> > > Since you provided feedback regarding the evaluation protocol, our intention was to explain, from the perspective of evaluation methodology, the gap between ML-based TSC research trends and deep-learning-based TSC research trends.
> > >
> > > The results under the InceptionTime setting were **also obtained based on the given original split (fold0)**, and for a more accurate comparison, we used only the fold0 results from the HC2 reference results (CIF, DrCIF, HC2).
> > >
> > > Regarding the number of epochs, as you pointed out, 1500 epochs in InceptionTime can indeed be considered somewhat excessive. Likewise, since MambaSL converges on most datasets with 100 epochs with 1e-3 learning rate, we kept our original setting and ran the experiments with the same configuration. The mention of the InceptionTime setting was solely for the purpose of clearly defining a reference point for comparison.
> > >
> > > &nbsp;
> > >
> > > ### > About 8:2 split
> > >
> > > We would also like to refine our wording here, as our previous expression appears to have caused some misunderstanding.
> > > We fully understand that your suggestion to try the 8:2 split was **not** intended as a criticism of reproducibility, but rather as a constructive alternative to enable more meaningful discussion. We are genuinely grateful for this.
> > >
> > > However, we encountered non-trivial difficulties when applying the 8:2 split to TSLANet, and we believe this reflects a **fundamental limitation of the UEA datasets**, which contain many datasets with very few samples.
> > >
> > > For example, in the Handwriting dataset, there exist classes with only two samples in the training set. Splitting such a dataset into 8:2 effectively makes **training infeasible**, and any resulting performance would be hard to interpret as a faithful reflection of the model’s capability. We suspect that this may be one of the reasons why, in the UCR/UEA bakeoff, they do not perform cross validation on the train dataset alone, but instead **re-split the entire dataset (train+test) 30 times for evaluation**.
> > >
> > > &nbsp;

---

> > > ### Author Response · Authors · 2025-12-04
> > > **Reply to the Follow-up Questions (part 4/4)**
> > >
> > > &nbsp;
> > >
> > > ### > Additional experiments on UCR datasets
> > >
> > > We now attach the results on the UCR datasets as you requested.
> > > For MambaSL, experiments were run on all 128 datasets, whereas for the non-DL models, experiments were conducted on 112 datasets, excluding those with missing values or variable-length sequences (the metadata in the .ts files lists 113 datasets including Fungi, but since most UCR works report results on the 112 datasets excluding Fungi, we followed that convention). The full results are provided in the supplementary material, and a summary is shown below. As can be seen, **unlike on UEA, non-DL models are generally stronger on UCR**.
> > >
> > > | model | # of datasets | avg. acc (128) | avg. acc (112) | # of win/draw | # of lose/draw |
> > > |:---:|---:|---:|---:|---:|---:|
> > > | **MambaSL** | 128 | 83.884 | 84.466 |  |  |
> > > | ROCKET | 112 |  | 85.581 | 56 | 68 |
> > > | HC2 | 112 |  | 86.506 | 52 | 75 |
> > > | Hydra | 112 |  | 85.742 | 58 | 65 |
> > > | MR+Hydra | 112 |  | 86.902 | 51 | 77 |
> > >
> > > What really stood out to us was that **a clear performance polarization across datasets persists between non-DL models and MambaSL (DL)**.
> > > The following table counts how many times each model ranks 1st, 2nd, 3rd, 4th, and 5th.
> > > As can be seen, except for 19 datasets, MambaSL is either 1st or 5th, showing highly polarized behavior. In other words, the performance gap among non-DL models is relatively small, so most of the time they occupy ranks 2–5 or 1–4 rather than producing such extremes.
> > >
> > > | model | # of rank 1 | # of rank 2 | # of rank 3 | # of rank 4 | # of rank 5 |
> > > |:---:|---:|---:|---:|---:|---:|
> > > | **MambaSL** | 45 | 6 | 6 | 7 | 48 |
> > > | ROCKET | 24 | 21 | 22 | 31 | 14 |
> > > | HC2 | 30 | 26 | 35 | 17 | 4 |
> > > | Hydra | 29 | 24 | 12 | 23 | 24 |
> > > | MR+Hydra | 46 | 28 | 17 | 13 | 8 |
> > >
> > > At the dataset level, MambaSL tends to perform better on the SIMULATED and SENSOR categories, whereas non-DL models show clear advantages on the ECG and IMAGE categories.
> > > Furthermore, even among dataset groups originating from the same source, we observe consistent patterns. For example, in the (PigAirwayPressure, PigArtPressure, PigCVP) dataset group, non-DL models consistently outperform MambaSL, whereas in the (SemgHandGenderCh2, SemgHandMovementCh2, SemgHandSubjectCh2) dataset group, MambaSL consistently outperforms all non-DL baselines.
> > >
> > > This again demonstrates, consistent with what we observed in `Figure 5,6 (UMAP scatterplot)`, that **dataset- or domain-specific inductive bias is extremely important in the TSC field**.
> > >
> > > We were able to derive these analyses thanks to your questions, and we are sincerely grateful for that.

---

> ### Comment · Reviewer_wsQb · 2025-11-24
> **Reply to  (part 1/4)**
>
> Thank you for the authors’ clarifications and responses. I would like to offer the following comments:
>
> First, regarding the authors’ statement that *“evaluation is typically based on the average over 30 stratified resamples”*: this strategy is indeed unsuitable for deep learning methods. The original UCR and UEA datasets already provide default training and test splits. For deep learning models, the authors could simply rely on these original splits for experimental evaluation and compare them fairly with non-deep learning baselines, without resorting to 30 stratified resamples, thereby significantly reducing training costs for deep learning methods.
>
> Second, the authors wrote: *“Our study clearly falls into the second category, focusing on multivariate time series classification and using TSLib to minimize gaps against prior work—even though this means inheriting its leakage issue.”*
> On this point, I want to emphasize that using the test set as the validation set for deep learning models is fundamentally incorrect and unacceptable in rigorous experimental evaluation. Although TSLib and ModernTCN have discussed and implicitly adopted this setting, mistakes in prior work should not be used as justification for reproducing the same flawed practice in new studies. Future work should build upon the strengths of existing research while correcting known weaknesses, rather than following prior errors simply because they exist.
>
> Therefore, regarding the authors’ decision to follow prior work in using the test set as the validation set for deep learning models, I do not consider this an appropriate or defensible experimental design.
>
>
> Regarding the authors’ response to **1.2. Answering questions related to univariate time series study [W1, Q2, Q3]**, the explanation remains unconvincing. First, InceptionTime does not explicitly emphasize univariate or multivariate settings because it does not model inter-variable relationships; moreover, the original InceptionTime paper evaluated the method solely on the 85 univariate UCR datasets due to not considering multivariate settings.
>
> In contrast, although the proposed MambaSL method repeatedly highlights UEA multivariate datasets in both the abstract and Introduction, the core contribution concerning how the proposed MambaSL models inter-variable relationships is not clearly articulated. For example, could the authors discuss in detail how their approach differs from graph-based methods [1] or the inter-variable relation modeling techniques described in [2]? More specifically, what unique innovations does the proposed method introduce for capturing inter-variable dependencies compared with these established approaches?
>
> Additionally, the authors could simply evaluate MambaSL on the default training and test splits of the UCR 85 univariate dataset and compare it with the MR-H model (the full runtime of MR-H is under two hour). The authors state that the computational burden of MambaSL  is minimal, especially given the authors’ repeated emphasis on MambaML’s efficiency and the modest size of the 85UCR dataset (fewer than 60,000 training samples). Under these conditions, the assertion that a week was insufficient to complete the UCR analysis appears unconvincing.
>
> [1] Fully-Connected Spatial-Temporal Graph for Multivariate Time Series Data. AAAI, 2024.
>
> [2] A comprehensive survey of deep learning for multivariate time series forecasting: A channel strategy perspective. arXiv, 2025.

---

> ### Author Response · Authors · 2025-12-04
> **Reply to the Follow-up Questions (part 2/4)**
>
> We now proceed to address the remaining points one by one.
>
> &nbsp;
>
> ---
>
> ### > How MambaSL models inter-variable relationships?
>
> We did not propose a separate structure that explicitly models inter-variable relationships. However, **the selective SSM in vanilla Mamba already inherently performs channel mixing** and thereby fulfills that role:
> - $\boldsymbol{B}$ can capture relationships among variables when projecting inputs into the state, and
> - $\boldsymbol{C}$ can again reflect inter-variable relationships when mapping the state back to the output, while
> - $\boldsymbol{\Delta}$ changes dynamically depending on the input, allowing time-varying structural relationships.
>
> Borrowing the terminology from the \[survey paper] you cited,
> Mamba can be seen as inherently incorporating **embedding fusion** and modeling inter-variable relationships with **asymmetry** and **dynamism**.
>
> However, inter-variable relationships are **not always beneficial** for every dataset. On the contrary, we wanted to emphasize that for some time series datasets, **not reflecting inter-variable relationships can be more effective**. This is precisely the motivation behind our H2 hypothesis and the key structural design direction of MambaSL.
>
> We **modularized $\boldsymbol{B}$, $\boldsymbol{C}$, and $\boldsymbol{\Delta}$** so that:
> - a selective SSM (time-variant, channel-mixing) and
> - an LTI SSM (time-invariant, channel-independent)
>
> can be chosen depending on dataset characteristics.
>
> As a result, we observed that for certain datasets, time-invariant step sizes or channel-independent input embeddings actually yielded better performance (e.g., EthanolConcentration, Handwriting, etc.).
> This aligns with observations from the forecasting literature, where \[PatchTST] demonstrated strong performance with a channel-independent model.
>
> In summary,
> - Mamba inherently has a structure capable of modeling inter-variable relationships, and
> - MambaSL (H2), conversely, provides a way to deactivate inter-variable relationships, enabling the selection of the most suitable inductive bias depending on the data.
>
> **> Reference**
>
> \[survey paper] *A comprehensive survey of deep learning for multivariate time series forecasting: A channel strategy perspective*. arXiv, 2025.
>
> \[PatchTST] *PatchTST: A Time Series is Worth 64 Words: Long-term Forecasting with Transformers*. ICLR, 2023.
>
> &nbsp;
>
> ### > Further discussion on modeling temporal properties for classification
>
> We fully recognize that feature engineering-based approaches (e.g., ROCKET) are powerful. However, in this work, we wanted to explore the possibility that **a well-learned latent space alone could be sufficient for effective classification**. As mentioned in the Introduction, we aimed to maximize the pure potential of Mamba itself, and to show that the structural advantages of MambaSL are effective across a broad range of time series classification datasets.
>
> This is possible because, compared to traditional recurrent models or Transformers, *Mamba is particularly good at memorizing long-term properties*. Thus, it is indeed plausible for Mamba to store only the information necessary for classification in the state across the entire sequence.
>
> One piece of evidence supporting this is the model ablation result where we use **only the latent vector at the last timestep** for classification (`Section 5.2` and `Appendix C.2`).
> If the Mamba block could not memorize long-term characteristics for a given dataset, it would be almost impossible to perform reasonable classification using only the last-timestep latent vector. However, the results do not support this concern; in fact, for datasets such as JapaneseVowels and DuckDuckGeese, using only the last timestep even outperforms all other pooling strategies.

---

> ### Author Response · Authors · 2025-12-04
> **Authors’ Sincere Gratitude to Reviewer wsQb**
>
> Dear reviewer wsQb,
>
> Once again, thank you for your thoughtful and constructive feedback.
> Although we did not have the chance to exchange more rounds of discussion, we hope that our final response was able to address at least some of your remaining questions.
> It was a pleasure to participate in this meaningful discussion.

---

### Author Response · Authors · 2025-12-04
**Summary of the Review Discussion by Authors**

We would like to sincerely thank the AC/SAC/PCs, who have devoted considerable effort to resolving the issues that inevitably arose during this review process, as well as all reviewers who carefully read our paper and provided valuable feedback.

Although we did not have much time to exchange many rounds of discussion, the three reviewers each provided deep and insightful feedback from different perspectives, which allowed us to significantly improve the clarity and overall quality of the paper.

&nbsp;

Below, we summarize three key points raised by each reviewer that helped highlight and strengthen the contributions of the study.

### > Reviewer **GbtJ**
- **Architectural Validity**: The reviewer evaluated that MambaSL’s "architecture choices are well-motivated and explained", and that the extension process from Mamba to MambaSL is presented in a "succinct and organized" manner.
- **Model/Dataset Visualization**: The reviewer highly appreciated the UMAP-based visualization and suggested extending it along the dataset axis. In response, we *newly added `Figure 6 (UMAP scatterplot along dataset axis)`*, which the reviewer characterized as "an important contribution to the TS community".
- **Extension to various types of time series**: The reviewer asked about the applicability and extensibility of MambaSL under various conditions such as irregular sampling, variable length, and recently released real-world benchmarks. We *added experimental results on ADFTD/FLAAP used in Medformer* to the main text and, based on those results and concrete examples, explained how MambaSL behaves and where its strengths and limitations lie. These additional discussions appear to have contributed to increasing confidence in MambaSL.

### > Reviewer **XLs9**
- **Thorough Experiments & Reproducibility**: The reviewer evaluated our experimental protocol and reporting as showing an "excellent commitment to reproducibility" and described our ablations as "thorough". The reviewer also referred to our re-evaluation of time-series-forecasting–origin models (e.g., Crossformer) as an "important finding".
- **Depth Ablation (Single-layer justification)**: The reviewer pointed out that, although our title and model name emphasize the single-layer design, we had not clearly justified this with comparisons to multi-layer variants. In response, we *newly conducted comparative experiments with 2- and 3-layer MambaSL and reflected the results* in the main text.
- **Novelty of H1–H4**: The reviewer commented that our four design choices are "a small change rather than a new architectural principle". While we agree with the characterization as "small changes", we emphasized in our reply that these are simple approaches that previous work has nonetheless overlooked, and that they in fact lead to meaningful improvements in Mamba’s TSC performance.

### > Reviewer **wsQb**
- **Clarification on Evaluation Protocol**: The reviewer raised concerns regarding the data leakage issue in $\mathtt{TSLib}$ on the UEA benchmark. We *added evaluation results for MambaSL under the InceptionTime setting* as the reviewer suggested. Although the average accuracy decreased somewhat, we confirmed that MambaSL remains competitive in terms of win–loss and Wilcoxon test results.
- **Explanation on capturing TS properties**: The reviewer asked for a more detailed explanation of how MambaSL models temporal patterns in multivariate time series. Our initial response addressed the questions regarding temporal patterns, after which the reviewer raised further questions about inter-variable relationships. In our follow-up, we clarified that selective SSM inherently performs channel mixing, and further emphasized that MambaSL (H2) provides a mechanism to deactivate inter-variable dynamics depending on dataset characteristics.
- **Computational Efficiency**: The reviewer asked about the advantages of MambaSL over other baselines in terms of memory usage and computational cost. We compared training time, test time, number of parameters, and accuracy across models, and the reviewer responded that this "makes it clear that MambaSL indeed offers strong computational efficiency".

&nbsp;

Overall, the diverse feedback from the three reviewers was invaluable in improving the clarity, technical soundness, and experimental reliability of the paper.
We are convinced that the paper has become significantly stronger, and we would once again like to express our sincere gratitude to all reviewers for their thoughtful comments.

&nbsp;

---

### Meta-Review · Area_Chair_9bqG · 2026-01-02

**Summary:**

Reviewers praised the paper for its strong performance on multivariate time series classification, clear and well-motivated architectural design and extensive reproducibility efforts. They noted that the work is principled, technically sound, and provides meaningful insights into time series modeling. The authors addressed all reviewer comments by adding new experiments, visualizations, and ablations, clarifying architectural choices, and expanding evaluations across datasets and so on. The revisions strengthened both empirical and conceptual contributions, demonstrating responsiveness to feedback and solidifying confidence in the MambaSL approach. Given the reviewers’ positive assessments and the authors’ thorough revisions, the paper meets the acceptance criteria and is recommended for acceptance.

**Reviewer Concerns:**

The authors submitted a rebuttal and addressed the reviewers’ main concerns. Key points addressed include clarifications of the MambaSL architecture, justification of the single-layer design through additional ablations, expanded evaluations across datasets, UMAP-based visualizations, and analysis of computational efficiency. These revisions improved the clarity, technical rigor, and reproducibility of the paper. All reviewers’ major suggestions regarding experimental thoroughness, architecture explanation, and model applicability were substantially incorporated, leaving no outstanding concerns.

**Reviewer Scores:**

**Reviewer GbtJ (original score 8 → 8)**

The reviewer had already expressed strong confidence in the architecture and clarity of MambaSL. Since the rebuttal primarily confirmed their understanding and added visualizations, it is unlikely their score would change, remaining at 8.

**Reviewer XLs9 (original score 6 → 6)**

The reviewer’s concerns about single-layer justification and novelty were addressed through additional ablations and clarifications. These responses strengthened confidence but did not substantially alter the overall assessment, so the score would likely remain 6.

**Reviewer wsQb (original score 4 → 6)**

The reviewer raised concerns about evaluation protocol, temporal properties, and test data leakage issue. The rebuttal addressed these with extended experiments, clarifications, significantly increasing confidence, which would likely raise the score from 4 to 6.

---

### Decision · Program_Chairs · 2026-01-26

Accept (Poster)